# Robust Test-time Video-Text Retrieval: Benchmarking and Adapting for Query Shifts

**Bingqing Zhang**[1,2]**, Zhuo Cao**[1]**, Heming Du**[1]**, Yang Li**[2]**, Xue Li**[1*]**, Jiajun Liu**[2,1*]**, Sen Wang**[1*]

[1]The University of Queensland, [2]CSIRO Data61, Australia

{bingqing.zhang, william.cao, heming.du, sen.wang}@uq.edu.au
xueli@eesc.uq.edu.au, {yang.li1, jiajun.liu}@csiro.au

## Abstract

Modern video-text retrieval (VTR) models excel on in-distribution benchmarks but are highly vulnerable to real-world *query shifts*, where the distribution of query data deviates from the training domain, leading to a sharp performance drop. Existing image-focused robustness solutions are inadequate to handle this vulnerability in video, as they fail to address the complex spatio-temporal dynamics inherent in these shifts. To systematically evaluate this vulnerability, we first introduce a comprehensive benchmark featuring 12 distinct types of video perturbations across five severity degrees. Analysis on this benchmark reveals that query shifts amplify the *hubness phenomenon*, where a few gallery items become dominant "hubs" that attract a disproportionate number of queries. To mitigate this, we then propose HAT-VTR (Hubness Alleviation for Test-time Video-Text Retrieval), as our baseline test-time adaptation framework designed to directly counteract hubness in VTR. It leverages two key components: a *Hubness Suppression Memory* to refine similarity scores, and *multi-granular losses* to enforce temporal feature consistency. Extensive experiments demonstrate that HAT-VTR substantially improves robustness, consistently outperforming prior methods across diverse query shift scenarios, and enhancing model reliability for real-world applications. Code is available at https://github.com/bingqingzhang/vtr_tta.git.

## 1 Introduction

While Video-Text Retrieval (VTR) models (Luo et al., 2022; Gorti et al., 2022) have achieved remarkable success, their performance hinges on a fragile assumption: that inference data are drawn from the same distribution as the training data. This assumption is frequently violated in real-world applications, leading to a phenomenon known as *query shift*, where the incoming data distribution deviates from the source. The problem is particularly acute for video input, as these distributional shifts introduce perturbations with a unique temporal dimension. For instance, real-world challenges like persistent fog or dynamic object occlusions (Fig. 1) introduce complex spatio-temporal domain shifts, corrupting not just static appearances but temporal consistency across frames and causing a sharp degradation in retrieval accuracy.

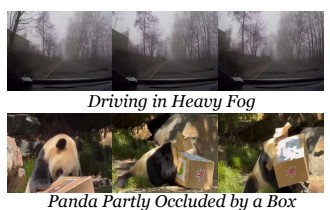

*Driving in Heavy Fog*

*Panda Partly Occluded by a Box*

Figure 1: Real-world videos illustrating diverse spatio-temporal complexities that challenge VTR models.

This vulnerability has spurred research into test-time robustness, yet efforts have so far been confined to the image-text domain. The first systematic study (Qiu et al., 2024) introduced a comprehensive image-text benchmark with controlled perturbations, revealing that even top-performing models are highly sensitive to distribution shifts. More recently, online Test-Time Adaptation (TTA) methods (Wang et al., 2021; Lee et al., 2024) have emerged to address this fragility. Notably, TCR (Li et al., 2025b) pioneered TTA for image-text query-shift retrieval by enforcing representation uniformity during inference. However, these pioneering works—both in benchmarking and adaptation—overlook the unique temporal challenges inherent to video. Their focus on static, frame-level

---

*Corresponding authors.

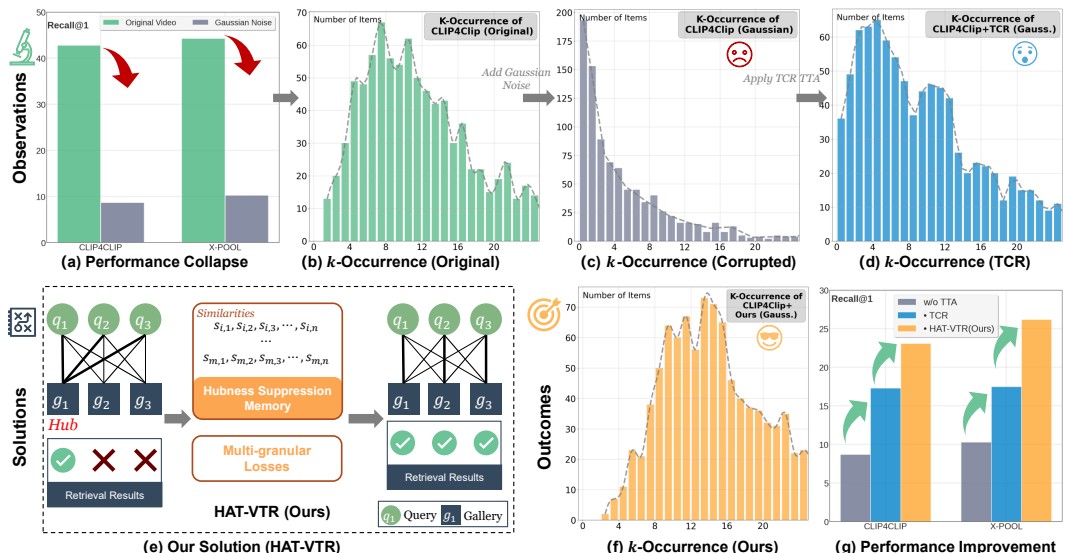

Figure 2: An overview of the motivation, solution, and performance of our proposed HAT-VTR method. **(a)** We first observe that the performance of representative video-to-text retrieval models collapses under Gaussian perturbations. **(b)** To diagnose this failure, we analyze the $k$-occurrence distribution (the number of times a gallery item is retrieved as the top-$15$ result), which is relatively balanced on original data. **(c)** When the query is corrupted, the distribution becomes heavy-tailed, highlighting a worsened hubness phenomenon where a few videos dominate retrieval rankings. **(d)** Applying the existing TTA method (TCR) partially mitigates the hubness problem. **(e)** To address this root cause, we propose HAT-VTR, a TTA method that uses a Hubness Suppression Memory and multi-granular losses to directly counteract hubness. **(f)** Our approach is highly effective, successfully restoring a balanced $k$-occurrence distribution. **(g)** Consequently, HAT-VTR significantly improves performance over corrupted baselines and prior art.

artifacts is insufficient for the dynamic nature of video perturbations. This makes test-time adaptation for VTR a unique challenge that requires spatio-temporal reasoning. However, to properly study this challenge, we need a benchmark designed for video dynamics, which is currently absent.

To fill this critical gap, we develop **MLVP** (**M**ulti-**L**evel **V**ideo **P**erturbations), an extended video perturbation benchmark that moves beyond the static image perturbations of prior work to probe the unique spatio-temporal and semantic vulnerabilities of VTR models. The MLVP benchmark encompasses perturbations across three hierarchical levels: (i) *low-level* perturbations that affect pixel values while maintaining temporal consistency (e.g., Gaussian noise), (ii) *mid-level* perturbations targeting object motion and spatial relationships (e.g., object occlusion), and (iii) *high-level* shifts that alter the core semantic or temporal structure (e.g., style transfer). In total, our benchmark comprises 12 distinct types of video perturbation approaches, each with five severity degrees, resulting in 60 controlled test scenarios. This systematic approach provides a principled foundation for analyzing VTR robustness and developing the next generation of resilient models.

Leveraging our benchmark, we uncover a striking vulnerability: the retrieval performance of VTR models collapses under video perturbations (Fig. 2(a)). We attribute this failure to an exacerbated *hubness phenomenon*, where a small subset of gallery items become "hubs" that disproportionately dominate nearest-neighbor rankings (Jian & Wang, 2023). Our analysis of the $k$-occurrence distributions provides clear evidence for this, as shown in Fig. 2(b-c), the distribution dramatically shifts from a relatively balanced state on clean data to a heavy-tailed one under perturbation. Crucially, we find that applying TCR partially alleviates this hubness (Fig. 2(d)); however, it is not equipped to handle this severe amplification directly, motivating the need for a targeted solution.

Motivated by our findings, we propose **HAT-VTR** (**H**ubness **A**lleviation for **T**est-time **V**ideo-**T**ext **R**etrieval), a straightforward yet effective framework designed to directly counteract this failure mode and establish a strong new baseline for robust test-time VTR. As conceptualized in Fig. 2(e), our framework enhances the TTA paradigm with two complementary innovations to systematically

mitigate hubness. First, a novel Hubness Suppression Memory (HSM) explicitly targets hubs at the similarity score level. Drawing inspiration from DSL (Cheng et al., 2021), this module maintains a dynamic history of retrieval patterns and performs real-time similarity refinement to demote overly popular gallery items, ensuring a more balanced neighbor distribution. Second, we adapt the core supervision signals of TCR (Li et al., 2025b) for the video domain by introducing multi-granular losses that leverage video's temporal hierarchy. Together, these components provide a direct and effective solution to the hubness problem. As visually demonstrated in Fig. 2(f-g), our approach successfully restores a balanced neighbor distribution and delivers robust retrieval performance that outperforms existing TTA methods. In summary, our contributions are threefold:

- To the best of our knowledge, this work introduces the first comprehensive multi-level video perturbation (MLVP) benchmark for evaluating test-time robustness of VTR, featuring a multi-level suite of spatio-temporal perturbations tailored for the video modality.

- We propose HAT-VTR, a straightforward yet effective framework that directly counteracts amplified hubness in test-time VTR. It introduces a Hubness Suppression Memory and multi-granular losses, establishing a strong new baseline for the field.

- Extensive experiments on different VTR TTA scenarios show that HAT-VTR consistently outperforms existing TTA methods under both query- and query-gallery-shift scenarios, enhancing model robustness and offering new insights into real-world VTR challenges.

## 2 RELATED WORK

**Video-Text Retrieval.** The dominant paradigm in VTR is the dual-encoder architecture (Radford et al., 2021; Luo et al., 2022), which learns a shared embedding space for videos and texts. Research in this area has focused on improving alignment strategies (Gorti et al., 2022), handling noisy correspondences in the training set (Huang et al., 2024), and mitigating the intrinsic *hubness phenomenon*—where a few items dominate retrieval results—through training objectives (Liu et al., 2020) or post-hoc score normalization (Cheng et al., 2021). However, these methods all operate under the standard i.i.d. assumption, presupposing that test data comes from the same clean distribution as the training data. Our work reveals that input corruptions at test time dramatically exacerbate the hubness problem and offers an online adaptation solution specifically for this failure mode.

**Test-Time Adaptation.** Test-Time Adaptation (TTA) aims to adapt a pre-trained model to a target domain using only unlabeled test data. The online setting, pioneered by TENT (Wang et al., 2021) through entropy minimization, adapts the model on a data stream without access to the source training data. This paradigm has since been extended to cross-modal retrieval, with methods like TCR (Li et al., 2025b) enforcing representation uniformity to handle query shifts. Nevertheless, these foundational works are designed for and evaluated on image-text tasks. They do not address the unique challenges of the video domain, such as the spatio-temporal nature of corruptions and the resulting amplification of the hubness phenomenon. Our work adheres to the strict online TTA setting, distinguishing it from paradigms like Unsupervised Domain Adaptation (UDA) (Hao & Zhang, 2023) or Test-Time Training (TTT) (Sun et al., 2020) that relax these constraints.

**Vision Corruption Benchmarks.** The systematic evaluation of model robustness was established by image-centric benchmarks like ImageNet-C (Hendrycks & Dietterich, 2019). While this has inspired efforts in the video domain, a comprehensive benchmark for VTR robustness against complex spatio-temporal corruptions has been lacking. The most related prior work (Schiappa et al., 2022) introduced a VTR-C benchmark, but its corruptions were primarily frame-wise extensions of image artifacts. In contrast, our benchmark is fundamentally different in two ways: 1) it introduces perturbations that explicitly target the dynamic, inter-frame properties of video, 2) it is the first designed not just for evaluating intrinsic robustness but for systematically comparing TTA methods in VTR.

**Data-driven Style Robustness.** Another line of research explicitly trains models on curated multi-style datasets (e.g., sketches, art) to achieve robustness against a *known* set of styles (Li et al., 2024b; Yanhao et al., 2025; Wu et al., 2025). This is fundamentally different from our online Test-TTA setting. Our method is trained only on clean data and adapts on-the-fly to unforeseen query shifts using only the unlabeled test stream, without requiring a pre-built style dataset.

Further literature discussions and their relation to our work are available in Appendix A.

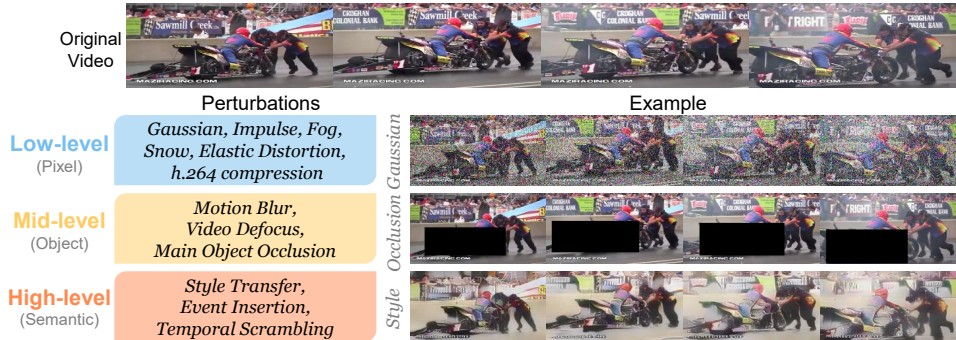

Figure 3: Overview of our proposed Multi-level Video Perturbations benchmark. We categorize 12 perturbations into a three-level hierarchy: *low-level* (pixel-based), *mid-level* (motion/object-aware), and *high-level* (semantic/temporal). Representative examples from each category are shown.

## 3 MLVP: MULTI-LEVEL VIDEO PERTURBATIONS BENCHMARK

As established in Sec. 2, existing robustness benchmarks are not well-suited for the video domain as they primarily apply static, image-level corruptions. To systematically investigate the vulnerabilities of VTR models to dynamic query shifts and provide a standard for evaluating adaptation methods, a benchmark tailored for video's spatio-temporal nature is essential. To this end, we extend the successful paradigm of systematic image-text robustness evaluation (Qiu et al., 2024) to the video domain, proposing our benchmark: MLVP (Multi-Level Video Perturbations). Our benchmark introduces 12 perturbation types across 5 severity degrees, creating 60 controlled scenarios. These scenarios are instantiated across the test sets of five standard VTR datasets (MSRVTT, ActivityNet, LSMDC, MSVD and DiDeMo), creating a comprehensive evaluation suite with over 8,500 unique perturbed videos. These perturbations are organized into a three-level hierarchy probing distinct model failures: from *low-level* pixel modifications and *mid-level* object/motion attributes to *high-level* alterations of the core semantic and temporal structure. (See Fig. 3.)

**Low-level video perturbations** modify pixel values while preserving temporal structure. Our suite simulates common degradations from hardware (*Gaussian* and *impulse* noise), weather (*fog*, *snow*), and digital processing (*elastic distortion*, *H.264 compression*). Critically, these perturbations are rendered with temporal consistency by applying the same realization (e.g., shared noise pattern) across frames of a video, distinguishing them from independent image corruptions.

**Mid-level perturbations** target object-centric and motion-based attributes to simulate more complex real-world degradations. To model camera artifacts, our *motion blur* and *video defocus* implementations use inter-frame motion vectors to apply a spatially varying blur, where the degradation is stronger in faster-moving regions. To simulate the obstruction of semantically critical elements, our *main object occlusion* employs a novel identification pipeline, providing a more challenging and realistic test than random occlusion. This pipeline first uses Qwen2.5-VL-7B (Bai et al., 2025a) to generate a video caption, then leverages key nouns from the caption as open-vocabulary queries for OWLv2 (Minderer et al., 2023) to locate and track the main object for occlusion.

**High-level video perturbations** alter the core semantic and temporal structure of a video to challenge its high-level understanding. Our *style transfer* perturbation tests a model's style invariance—its ability to recognize semantic content across diverse visual renderings. For efficiency, we follow the approach of Qiu et al. (2024) and employ AdaIN (Huang & Belongie, 2017), ensuring temporal consistency by applying a single style image and a fixed set of parameters across all frames. *Event insertion* challenges contextual understanding by using a retrieval model to select a semantically similar video snippet from a database and splice it into the original video. Finally, *temporal scrambling* simulates network streaming issues like packet loss and out-of-order delivery by trimming and reordering video chunks, disrupting the narrative flow and causal relationships.

Further details about datasets and MLVP benchmark are available in Appendix B.1 and E.

## 4 HAT-VTR: THE TEST-TIME ADAPTATION METHOD FOR VTR

### 4.1 NOTATIONS AND PROBLEM FORMULATION

In video-text retrieval (VTR), the setup consists of a query set $X^Q$ and a gallery set $X^G$. VTR encompasses two sub-tasks: video-to-text (*v2t*) and text-to-video (*t2v*) retrieval, where the modality of the query and gallery sets are swapped. A dual-encoder model, comprising a query encoder $f_{\theta^Q}$ and a gallery encoder $f_{\theta^G}$, maps these inputs into a shared embedding space. This is represented as:

$$Z^Q = \{f_{\theta^Q}(x) \mid x \in X^Q\}, \quad Z^G = \{f_{\theta^G}(x) \mid x \in X^G\}, \tag{1}$$

where $Z^Q$ and $Z^G$ are the sets of query and gallery embeddings. Retrieval is then based on a similarity matrix computed between these embeddings:

$$S^{Q,G} = g_\theta(Z^Q, Z^G). \tag{2}$$

The function $g_\theta$ varies with the alignment strategy, ranging from a parameter-free cosine similarity for coarse-grained alignment to learnable modules like cross-attention transformers for fine-grained alignment. Finally, the scores in $S^{Q,G}$ are used to rank gallery items for each query and return the top results. This process is typically asymmetric in practice: the gallery embeddings $Z^G$ are pre-computed offline, whereas query embeddings $Z^Q$ are computed online upon request.

The standard VTR paradigm operates on the assumption that the evaluation dataset, $\mathcal{D}_E$, shares the same distribution as the finetuning dataset, $\mathcal{D}_F$, i.e., $\mathcal{P}(\mathcal{D}_F) \sim \mathcal{P}(\mathcal{D}_E)$. Online Test-Time Adaptation (TTA) addresses the setting where this assumption is violated by a distribution shift. In the cross-modal TTA setting, the adaptation is formulated as a self-supervised query prediction task. For an online batch of query embeddings $Z^{Q_b} \in \mathbb{R}^{B \times D}$ and the full gallery embeddings $Z^G \in \mathbb{R}^{N_G \times D}$, the correspondence probabilities are modeled as:

$$\mathbf{p} = \text{Softmax}(Z^{Q_b}(Z^G)^T / \tau), \tag{3}$$

where $\tau$ is a temperature hyperparameter. A primary objective of TTA is to increase the model's prediction confidence on the target data by minimizing the softmax entropy $\eta(\cdot)$:

$$\min_{\theta \in \Theta_s} \mathcal{L}_{TTA}(\mathbf{p}) = \min_{\theta \in \Theta_s} \eta(\mathbf{p}), \tag{4}$$

where $\Theta_s$ are the adaptable source model parameters. Following Li et al. (2025b), this framework addresses two primary scenarios: **Query-Shift** (QS), where a query distribution shift ($\mathcal{P}(X_E^Q) \nsim \mathcal{P}(X_F^Q)$) requires adapting the query encoder; and more challenging **Query-Gallery-Shift** (QGS), where the gallery distribution also shifts. QGS includes cases such as (a) *Cross-dataset adaptation*, involving a $\mathcal{D}_F \to \mathcal{D}_E$ transfer under the shift $\mathcal{P}(\mathcal{D}_F) \nsim \mathcal{P}(\mathcal{D}_E)$; and (b) *Zero-shot adaptation*, involving a direct transfer from pretraining to evaluation ($P \to E$) with the shift $\mathcal{P}(\mathcal{D}_P) \nsim \mathcal{P}(\mathcal{D}_E)$.

### 4.2 HUBNESS SUPPRESSION MEMORY (HSM)

The Hubness Suppression Memory (HSM) is a dynamic module designed to counteract the amplified hubness phenomenon at test time. Inspired by DSL (Cheng et al., 2021), HSM's core mechanism is an adaptive, bilateral normalization of similarity scores, which leverages a memory bank to track recent query-gallery interaction patterns from the online data stream.

For the current batch of query embeddings $Z_t^{Q_b}$ at time step $t$, we first compute its similarity matrix $S_t = g_\theta(Z_t^{Q_b}, Z^G)$. The HSM leverages a memory bank, $\mathcal{M}_{t-1}$, which stores the $K-1$ most recent similarity matrices $\{S_{t-K+1}, \ldots, S_{t-1}\}$. Together with the current matrix $S_t$, these are used to form an aggregated similarity matrix $\bar{S} \in \mathbb{R}^{(B \cdot K) \times N_G}$:

$$\bar{S} = \text{Concat}(S_t, S_{t-1}, \ldots, S_{t-K+1}). \tag{5}$$

Based on this aggregated history, we compute two distinct weight matrices. First, a gallery-centric weight matrix, $W_{\text{gallery}} = \text{softmax}_{\text{col}}(\alpha \bar{S})$, captures the "popularity" of each gallery item across recent queries. Second, a query-centric matrix, $W_{\text{query}} = \text{softmax}_{\text{row}}(\beta \bar{S})$, captures the tendency of each query to concentrate on a few items. $\alpha$ and $\beta$ are temperature hyperparameters. The final hub-suppressed similarity matrix $\hat{S}$ is then calculated as a weighted combination of these components:

$$\hat{S} = m(\bar{S} \odot W_{\text{gallery}}) + (1 - m)(\bar{S} \odot W_{\text{query}}), \tag{6}$$

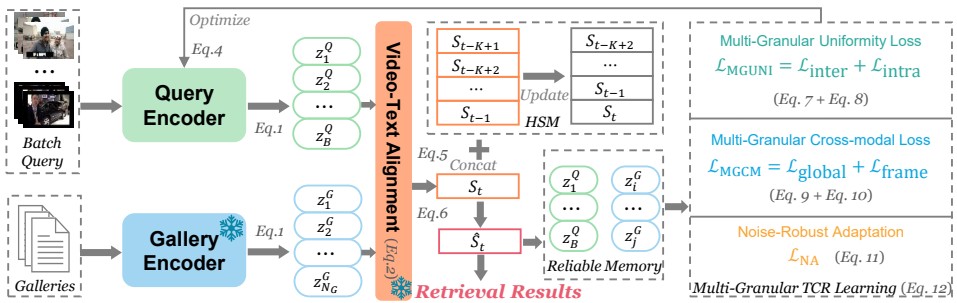

Figure 4: The pipeline of HAT-VTR. It operates via two parallel components: Hubness Suppression Memory (HSM) refines similarity scores to counteract hubness, while the query encoder is continuously updated using multi-granular losses to adapt to the target domain.

where $\odot$ denotes the element-wise product, and $m$ is a balancing parameter in $[0, 1]$. The refined score for the current batch, $\hat{S}_t$, is extracted from the first $B$ rows of $\hat{S}$. To maintain temporal relevance, the memory $\mathcal{M}$ is managed as a first-in, first-out (FIFO) queue of size $K$. This queue mechanism ensures the hubness statistics are always based on the most recent data, allowing for rapid adaptation. HSM is integrated into the TTA pipeline at two stages. First, for *Hubness-Aware Target Selection*, we use the refined scores $\hat{S}_t$ instead of raw, hub-biased similarities to select pseudo-positives for building the Reliable Memory (RM) in our adaptation loss (Sec. 4.3), stabilizing the learning process by preventing error accumulation. Second, for *Posterior Similarity Reranking*, we apply HSM to the adapted similarity scores to produce the final output. The bilateral re-weighting in Eq. 6 suppresses the spurious, low-consensus scores that hubs attract while preserving high-consensus matches, directly improving retrieval accuracy.

### 4.3 MULTI-GRANULAR TCR LEARNING

Our adaptation framework extends TCR's (Li et al., 2025b) learning principles by introducing multi-granular supervision for the video domain. We formulate our objectives for the *v2t* task without loss of generality; for the inverse *t2v* task, the uniformity loss simply omits its intra-video component. The entire process is stabilized by a **Reliable Memory** (RM), a memory of reliable query-gallery pairs selected by HSM, providing stable historical targets to prevent catastrophic forgetting.

**Multi-Granular Uniformity Loss** ($\mathcal{L}_{MGUNI}$)**.** To prevent representation collapse, we adapt TCR's uniformity principle for video structure. Our loss comprises two terms: an *inter-video* term scattering the global representation of each query in a batch ($Z_i^{Q_b}$) from the batch's mean $\bar{Z}^{Q_b}$,

$$\mathcal{L}_{\text{inter}} = \frac{1}{B} \sum_{i=1}^{B} \exp(-\|Z_i^{Q_b} - \bar{Z}^{Q_b}\|_2/t), \tag{7}$$

and an *intra-video* term scattering a video's frame-level features ($Z_{i,f}^{Q_b}$) from their per-video global representation (i.e. mean pooling in TTA) $Z_i^{Q_b}$ to preserve temporal diversity:

$$\mathcal{L}_{\text{intra}} = \frac{1}{B} \sum_{i=1}^{B} \left( \frac{1}{T} \sum_{f=1}^{T} \exp(-\|Z_{i,f}^{Q_b} - Z_i^{Q_b}\|_2/t) \right). \tag{8}$$

The total loss, $\mathcal{L}_{MGUNI} = \mathcal{L}_{\text{inter}} + \mathcal{L}_{\text{intra}}$, promotes multi-granular diversity.

**Multi-Granular Cross-Modal Loss** ($\mathcal{L}_{MGCM}$)**.** We evolve TCR's modality gap alignment into a multi-granular loss with two components. The *global alignment* loss aligns the modality gap of the current batch (between query batch's mean $\bar{Z}^{Q_b}$ and selected pseudo-positive galleries mean $\bar{Z}^{G_b}$) with a stable target gap, $\|\bar{Z}_{\mathcal{RM}}^Q - \bar{Z}_{\mathcal{RM}}^G\|_2$, computed from mean features in RM:

$$\mathcal{L}_{\text{global}} = (\|\bar{Z}^{Q_b} - \bar{Z}^{G_b}\|_2 - \|\bar{Z}_{\mathcal{RM}}^Q - \bar{Z}_{\mathcal{RM}}^G\|_2)^2. \tag{9}$$

Concurrently, the *frame-level alignment* loss aligns the cross-covariance of the batch's frame-level query features ($Z_f^{Q_b}$) and the corresponding gallery pseudo-positives ($Z^{G_b}$) with a target from RM:

$$\mathcal{L}_{\text{frame}} = \text{MSE}(\text{Cov}(Z_f^{Q_b}, Z^{G_b}), \text{Cov}(Z_{\mathcal{RM}}^Q, Z_{\mathcal{RM}}^G)). \tag{10}$$

Table 1: Comparisons *v2t* results on the MSRVTT-1kA with severity degree 5, regarding the Recall@1 (%) metric. The best results are in **bold**, and ours are highlighted.

| Query | Low-Level | | | | | | Mid-Level | | | High-Level | | | |
|---|---|---|---|---|---|---|---|---|---|---|---|---|---|
| Shift | Gauss. | Impul. | Fog | Snow | Elastic. | H264. | Motion | Defocus | Occlu. | Style | Event | Tempo. | Avg. |
| CLIP4Clip | 8.7 | 5.2 | 26.7 | 14.9 | 10.1 | 26.2 | 24.6 | 5.2 | 22.5 | 4.8 | 22.2 | 32.8 | 17.0 |
| • Tent | 5.0 | 2.7 | 27.8 | 16.1 | 10.8 | 26.2 | 25.6 | 4.6 | 22.4 | 4.2 | 22.9 | 33.1 | 16.8 |
| • READ | 9.5 | 6.7 | 26.7 | 13.8 | 9.8 | 25.9 | 22.6 | 5.8 | 22.2 | 5.8 | 22.1 | 32.7 | 17.0 |
| • SAR | 7.2 | 3.1 | 27.9 | 16.5 | 11.7 | 26.4 | 25.8 | 5.2 | 22.8 | 4.3 | 22.7 | 33.0 | 17.2 |
| • EATA | 16.2 | 0.7 | 30.0 | 17.3 | 18.5 | 27.5 | 28.7 | 6.4 | 24.5 | 2.1 | 23.2 | **34.0** | 19.1 |
| • TCR | 17.3 | 11.6 | 32.9 | 21.9 | 20.1 | 28.4 | 28.7 | 9.0 | 25.2 | 6.7 | 22.7 | 32.0 | 21.4 |
| • Ours | **23.1** | **13.5** | **38.1** | **29.6** | **29.6** | **30.6** | **32.6** | **15.7** | 30.1 | **12.1** | **26.3** | 33.1 | **26.2** |
| Xpool | 10.3 | 7.1 | 27.8 | 17.2 | 17.0 | 28.8 | 25.4 | 6.2 | 30.2 | 6.6 | 30.3 | 33.6 | 20.0 |
| • Tent | 7.4 | 3.7 | 29.0 | 19.1 | 18.2 | 29.5 | 27.0 | 6.7 | 31.0 | 5.9 | 30.5 | 34.1 | 20.2 |
| • READ | 11.5 | 8.9 | 27.7 | 17.0 | 16.3 | 28.3 | 23.6 | 6.3 | 29.8 | 7.6 | 30.3 | 33.7 | 20.1 |
| • SAR | 10.0 | 3.9 | 28.7 | 19.3 | 18.2 | 29.5 | 27.5 | 7.7 | 30.9 | 6.3 | 30.8 | 34.0 | 20.6 |
| • EATA | 18.2 | 0.9 | 33.9 | 21.8 | 23.1 | 30.9 | 30.6 | 9.3 | 30.6 | 3.5 | 31.4 | **35.0** | 22.4 |
| • TCR | 17.5 | 16.7 | 33.6 | 22.4 | 21.5 | 31.0 | 30.0 | 9.8 | 31.4 | 9.2 | 30.8 | 34.1 | 24.0 |
| • Ours | **26.2** | **22.3** | **41.4** | **30.9** | **33.7** | **35.6** | **35.3** | **17.8** | **35.5** | **14.4** | **35.2** | 34.7 | **30.3** |

Table 2: Comparisons on *v2t* R@1 on the ActivityNet dataset with the highest severity degree.

| Query | Low-Level | | | | | | Mid-Level | | | High-Level | | | |
|---|---|---|---|---|---|---|---|---|---|---|---|---|---|
| Shift | Gauss. | Impul. | Fog | Snow | Elastic. | H264. | Motion | Defocus | Occlu. | Style | Event | Tempo. | Avg. |
| CLIP4Clip | 4.90 | 5.13 | 19.58 | 10.19 | 7.36 | 33.54 | 14.81 | 2.95 | 13.02 | 3.66 | 9.80 | 24.02 | 12.41 |
| • Tent | 4.60 | 0.92 | 10.78 | 4.98 | 9.48 | 33.58 | 18.73 | 0.63 | 4.17 | 0.79 | 9.88 | 24.16 | 10.23 |
| • READ | 2.70 | 4.07 | 15.62 | 8.72 | 4.88 | 31.71 | 7.42 | 1.65 | 14.83 | 3.88 | 9.86 | 23.77 | 10.76 |
| • SAR | 8.87 | 1.26 | 18.97 | 12.53 | 11.06 | 33.94 | 18.45 | 1.36 | 6.24 | 1.32 | 10.01 | 24.30 | 12.36 |
| • EATA | 3.88 | 0.28 | 13.28 | 13.89 | 8.87 | 31.95 | 21.58 | 1.10 | 3.95 | 1.48 | 8.20 | **24.32** | 11.07 |
| • TCR | 5.45 | 11.61 | 27.17 | 17.51 | 18.67 | 18.22 | 11.73 | 7.46 | 7.44 | 4.27 | 7.20 | 17.61 | 12.86 |
| • Ours | **18.26** | **18.81** | **32.54** | **23.90** | **27.58** | **36.47** | **26.87** | **12.75** | **20.42** | **8.42** | **17.94** | 23.45 | **22.28** |
| Xpool | 6.06 | 5.08 | 18.81 | 10.13 | 8.24 | 29.27 | 13.00 | 3.15 | 16.76 | 4.21 | 22.60 | 23.35 | 13.39 |
| • Tent | 7.77 | 1.14 | 16.37 | 6.28 | 9.27 | 29.45 | 14.97 | 1.08 | 6.45 | 2.07 | 22.86 | 23.23 | 11.75 |
| • READ | 3.62 | 4.03 | 18.61 | 7.67 | 6.22 | 28.86 | 8.07 | 2.20 | 16.88 | 4.37 | 22.01 | 23.10 | 12.14 |
| • SAR | 8.48 | 1.53 | 19.52 | 10.45 | 10.84 | 29.43 | 14.85 | 1.99 | 14.64 | 2.79 | 22.84 | 23.35 | 13.39 |
| • EATA | 7.75 | 0.39 | 10.09 | 3.68 | 10.72 | 28.66 | 16.70 | 0.43 | 6.10 | 1.50 | 22.55 | 22.98 | 10.96 |
| • TCR | 9.21 | 9.92 | 23.65 | 16.19 | 14.54 | 27.21 | 15.76 | 6.10 | 10.88 | 4.05 | 18.79 | 17.65 | 14.50 |
| • Ours | **14.60** | **14.64** | **28.41** | **20.64** | **23.04** | **31.99** | **22.35** | **10.66** | **19.02** | **7.61** | **27.80** | 23.39 | **20.35** |

The total loss, $\mathcal{L}_{MGCM} = \mathcal{L}_{\text{global}} + \mathcal{L}_{\text{frame}}$, ensures alignment at both coarse and fine-grained levels.

**Noise-Robust Adaptation and Total Loss.** Finally, we retain TCR's core noise-robust entropy minimization, which is a weighted entropy over the batch's correspondence probabilities $\mathbf{p}$:

$$\mathcal{L}_{\text{NA}} = \frac{1}{\sum_i \mathbb{I}_{\{S(\mathbf{p}_i)>0\}}} \sum_{i=1}^{B} S(\mathbf{p}_i)\eta(\mathbf{p}_i), \quad \text{where } S(\mathbf{p}_i) = \max(1 - \eta(\mathbf{p}_i)/E_m, 0). \quad (11)$$

The self-adaptive weight $S(\mathbf{p}_i)$ filters out unreliable samples by assigning zero weight to any query whose prediction entropy $\eta(\mathbf{p}_i)$ exceeds a threshold $E_m$ derived from the Reliable Memory. Our final adaptation objective is a sum of all components:

$$\mathcal{L}_{\text{total}} = \mathcal{L}_{MGUNI} + \mathcal{L}_{MGCM} + \mathcal{L}_{NA}. \quad (12)$$

**Pipeline.** Finally, the HAT-VTR pipeline is depicted in Fig. 4. For each incoming query batch, an initial similarity matrix $S_t$ is computed against gallery features. This matrix serves two parallel purposes: for model adaptation, it is used to compute $\mathcal{L}_{\text{total}}$ (Eq. 12) which updates the query encoder; for retrieval, it is concurrently refined by the HSM into a hub-alleviated matrix $\hat{S}_t$ for the final ranking. This dual-path mechanism allows the framework to adapt representations while directly mitigating hubness in the similarity space, leading to robust online test-time video-text retrieval.

## 5 EXPERIMENTS

### 5.1 IMPLEMENTATION DETAILS AND EXPERIMENT SETTINGS

**Models, Datasets, and Baselines.** Our experiments are based on two representative VTR models, CLIP4Clip (Luo et al., 2022) and X-Pool (Gorti et al., 2022), covering coarse- and fine-grained

Table 3: Comparisons on *t2v* Recall@1 (%) on the MSRVTT-1kA dataset under text perturbations.

| Query Shift | Character-Level | | | | | Word-Level | | | | | Sentence-Level | | | | | Avg. |
|---|---|---|---|---|---|---|---|---|---|---|---|---|---|---|---|---|
| | OCR | CI | CR | CS | CD | SR | WI | WS | WD | IP | Backtrans. | Formal | Casual | Passive | Active | |
| CLIP4Clip | 21.5 | 12.8 | 12.2 | 15.7 | 11.2 | 38.9 | 39.0 | 39.7 | 39.5 | 39.7 | 38.5 | 40.9 | 39.7 | 40.6 | 41.8 | 31.4 |
| • Tent | 21.5 | 12.6 | 11.9 | 15.9 | 11.1 | 38.6 | 39.0 | 39.9 | 39.7 | 39.5 | 38.3 | 41.4 | 40.1 | 41.0 | 41.6 | 31.5 |
| • READ | 21.5 | 12.9 | 12.4 | 15.8 | 11.0 | 38.8 | 39.1 | 39.7 | 39.2 | 39.5 | 38.6 | 41.0 | 39.9 | 40.4 | 41.6 | 31.4 |
| • SAR | 21.4 | 12.8 | 12.1 | 15.7 | 11.1 | 38.6 | 38.9 | 39.9 | 39.8 | 39.5 | 38.3 | 41.3 | 40.2 | 40.0 | 41.7 | 31.4 |
| • EATA | 21.6 | 12.5 | 12.2 | 15.1 | 10.5 | 38.6 | 39.3 | 39.7 | 40.2 | 39.8 | 38.4 | 41.2 | 39.6 | 40.2 | 41.5 | 31.4 |
| • TCR | 21.7 | 12.8 | 13.0 | 14.9 | 11.0 | 39.1 | 39.4 | 39.4 | 39.8 | 39.8 | 37.8 | 40.9 | 39.8 | 40.1 | 41.7 | 31.4 |
| • Ours | **24.5** | **13.8** | **14.1** | **16.7** | **12.8** | **40.7** | **41.6** | **41.2** | **42.9** | **42.9** | **39.8** | **43.6** | **42.3** | **42.4** | **43.7** | **33.5** |
| Xpool | 25.0 | 12.5 | 13.2 | 16.9 | 12.1 | 43.4 | **44.1** | 42.3 | 45.3 | 46.2 | 43.1 | 46.6 | 45.3 | 44.9 | 47.0 | 35.2 |
| • Tent | 25.5 | 12.4 | 13.1 | 17.0 | 12.3 | 43.2 | 43.8 | 42.2 | 45.2 | 46.2 | 42.9 | 46.7 | 44.9 | 44.6 | 46.7 | 35.1 |
| • READ | 25.1 | 12.4 | 13.3 | 17.0 | 12.0 | 43.1 | 43.9 | 42.2 | 45.7 | 46.3 | 43.1 | 46.6 | 45.5 | 45.1 | 47.2 | 35.2 |
| • SAR | 25.6 | 12.7 | 13.2 | 17.1 | 12.2 | 43.4 | 43.7 | 42.3 | 45.3 | 46.2 | **43.3** | 46.7 | 45.2 | 44.7 | 47.0 | 35.2 |
| • EATA | 25.2 | 12.8 | 13.2 | 17.1 | 12.2 | 42.8 | **44.1** | 42.1 | 45.3 | 45.4 | 42.1 | 45.7 | 44.9 | 45.0 | 46.2 | 34.9 |
| • TCR | 25.7 | 12.8 | 13.2 | 16.4 | 12.2 | 43.6 | 43.9 | 41.5 | 45.2 | 46.0 | 42.7 | 46.1 | 44.5 | 44.8 | 46.8 | 35.0 |
| • Ours | **26.9** | **14.8** | **14.4** | **18.5** | **14.7** | **44.8** | 43.8 | **44.4** | **46.6** | **47.3** | 42.6 | **48.7** | **46.1** | **46.1** | **48.1** | **36.5** |

alignment. We evaluate on five standard benchmarks, reporting results on MSRVTT-1kA (Xu et al., 2016) (Video-Text Dataset) and ActivityNet (Fabian & Niebles, 2015) (Video-Paragraph Dataset) in the main paper (see Appendix D for full results). We compare HAT-VTR against five TTA baselines: TENT (Wang et al., 2021), READ (Yang et al., 2024), SAR (Niu et al., 2023), EATA (Niu et al., 2022) and the most relevant method TCR (Li et al., 2025b).

**TTA Scenarios and Implementation.** We evaluate both video-to-text (*v2t*) and text-to-video (*t2v*) tasks under two primary domain shift scenarios. The first is **Query-Shift (QS)**, where only queries are corrupted using the 12 video perturbations from our benchmark for *v2t* and 15 text perturbations (Qiu et al., 2024) for *t2v*. The second, more challenging scenario is **Query-Gallery-Shift (QGS)**, which includes both cross-dataset and zero-shot adaptation settings. Our framework is built upon the X-Pool codebase, and all baselines are adapted from their official repositories for fairness. Following standard TTA practice (Li et al., 2025b; Wang et al., 2021), we use the AdamW optimizer Loshchilov & Hutter (2017) to adapt only the Layer Normalization (LN) parameters of the query encoder. All experiments run on a single NVIDIA RTX 4090 GPU. We report the standard Recall@K (R@K) metric and use a batch size of 16 for all online inference. Key hyperparameters ($\tau = 0.02, t = 10$) follow TCR (Li et al., 2025b) to ensure a fair comparison.

## 5.2 COMPARISON RESULTS ON QUERY-SHIFT

We first evaluate HAT-VTR under QS scenarios, where only the queries are corrupted at test time. This setup mimics common real-world problems, such as using a noisy video or a text query with typos to search a clean database.

As shown in Tab. 1, 2, the performance of baseline VTR models collapses under our MLVP benchmark. Most existing TTA methods, which are not designed for video's complexities, struggle to adapt and offer limited benefits. While TCR shows modest recovery, our HAT-VTR demonstrates consistently superior robustness. By directly targeting the amplified hubness, it substantially outper-

Table 4: Comparisons on *Cross-dataset Adaptation* of QGS.

| QGS Cross Dataset | MSRVTT→ActivityNet | | | | ActivityNet→MSRVTT | | | |
|---|---|---|---|---|---|---|---|---|
| | *v2t* | | *t2v* | | *v2t* | | *t2v* | |
| *Metrics* | **R@1↑** | **R@5↑** | **R@1↑** | **R@5↑** | **R@1↑** | **R@5↑** | **R@1↑** | **R@5↑** |
| CLIP4Clip | 32.64 | 60.28 | 28.70 | 57.58 | 35.50 | 60.20 | 35.00 | 57.50 |
| • Tent | 32.80 | 60.55 | 28.19 | 57.07 | 35.40 | 60.50 | 34.90 | 58.30 |
| • READ | 26.38 | 52.45 | 27.98 | 57.25 | 35.20 | 59.80 | 34.90 | 57.80 |
| • SAR | 32.89 | 60.57 | 28.51 | 57.01 | 35.20 | 60.60 | 34.80 | 58.10 |
| • EATA | 31.75 | 58.92 | 27.70 | 55.26 | 36.40 | 60.80 | 36.30 | 59.30 |
| • TCR | 19.28 | 40.37 | 26.30 | 52.49 | 35.20 | 60.00 | 35.00 | 58.40 |
| • Ours | **36.10** | **64.21** | **36.53** | **65.43** | **38.00** | **64.70** | **38.60** | **62.60** |
| Xpool | 29.79 | 57.76 | 30.61 | 57.92 | 35.80 | 62.80 | 38.10 | 61.80 |
| • Tent | 29.92 | 57.94 | 30.53 | 57.94 | 36.30 | 62.20 | 38.00 | 62.10 |
| • READ | 29.14 | 57.31 | 30.14 | 56.46 | 36.20 | 62.60 | 37.80 | 61.40 |
| • SAR | 29.98 | 57.8 | 30.69 | 57.96 | 36.30 | 62.70 | 38.00 | 62.00 |
| • EATA | 28.11 | 56.21 | 29.98 | 57.64 | 36.60 | 62.80 | 38.60 | 61.90 |
| • TCR | 29.45 | 56.17 | 29.81 | 57.25 | 37.60 | 62.90 | 38.50 | 62.20 |
| • Ours | **33.62** | **62.21** | **34.13** | **61.85** | **40.30** | **64.80** | **39.50** | **64.50** |

forms all competitors on both ActivityNet and MSRVTT-1kA datasets, setting a new strong baseline for robust *v2t* retrieval.

For the reverse *t2v* task (Tab. 3), we observe a similar trend where HAT-VTR again achieves the best average performance against text corruptions. Overall, these results validate our core strategy of targeting the hubness phenomenon. However, We also note that our method's advantage diminishes

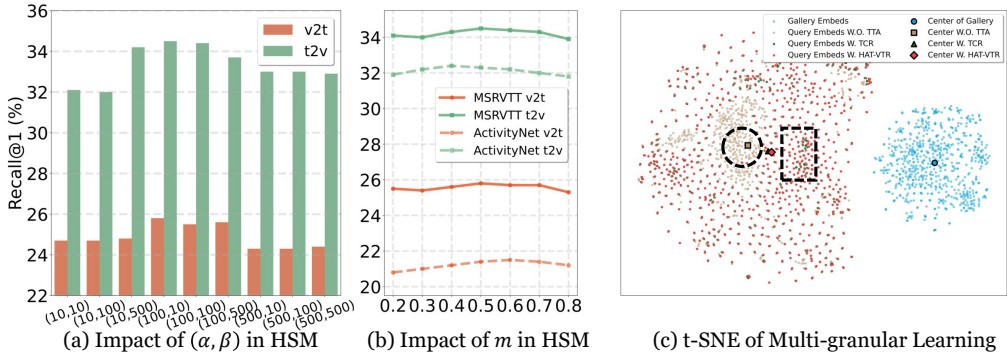

(a) Impact of $(\alpha, \beta)$ in HSM  (b) Impact of $m$ in HSM  (c) t-SNE of Multi-granular Learning

Figure 5: Ablation studies on HSM's hyperparameters and t-SNE visualization of HAT-VTR.

in specific cases (e.g., Temporal Scrambling for *v2t* or BackTrans for *t2v*) where, as analyzed in Sec. F.1, the hubness problem itself is less severe. On the other hand, this highlights that our MLVP benchmark is comprehensive and challenging enough to uncover these specific model failure modes.

## 5.3 COMPARISON RESULTS ON QUERY-GALLERY-SHIFT

We further evaluate HAT-VTR in the more challenging QGS scenarios, where a model fine-tuned on one dataset is adapted to a completely different dataset at test time. These include *cross-dataset adaptation*, where a model fine-tuned on one dataset is transferred to another (Tab. 4), and the even more difficult *zero-shot adaptation*, where a pre-trained CLIP model adapts directly to a retrieval task without any fine-tuning (Tab. 5). In both scenarios, the large query and gallery domain gaps pose a major chal-

Table 5: Results of *Zero-shot Adaptation*.

| QGS | MSRVTT | | | | ActivityNet | | | |
|---|---|---|---|---|---|---|---|---|
| Zero-shot | v2t | | t2v | | v2t | | t2v | |
| *Metrics* | **R@1**↑ | **R@5**↑ | **R@1**↑ | **R@5**↑ | **R@1**↑ | **R@5**↑ | **R@1**↑ | **R@5**↑ |
| CLIP | 26.50 | 51.80 | 30.10 | 53.40 | 17.84 | 41.18 | 21.17 | 46.35 |
| • Tent | 26.80 | 52.20 | 30.30 | 53.20 | 16.78 | 39.19 | 21.72 | 46.92 |
| • READ | 25.70 | 49.30 | 29.80 | 53.40 | 9.27 | 24.36 | 15.44 | 36.47 |
| • SAR | 26.70 | 52.50 | 30.20 | 53.30 | 18.43 | 41.65 | 21.80 | 46.86 |
| • EATA | 27.20 | 53.50 | 30.70 | 53.10 | 16.80 | 36.77 | 18.32 | 42.57 |
| • TCR | 27.90 | 54.80 | 30.50 | 53.90 | 18.65 | 41.65 | 22.11 | 48.30 |
| • Ours | **35.40** | **61.40** | **35.20** | **58.10** | **28.92** | **53.91** | **29.18** | **57.23** |

lenge that most TTA methods fail to overcome, providing little to no improvement and sometimes even hurting performance. In contrast, our method consistently adapts to the new domains effectively, achieving clear and significant performance gains across all transfer settings and on both retrieval tasks (*v2t* and *t2v*). This demonstrates that our hubness-mitigating approach is a powerful and generalizable solution for handling severe domain shifts.

## 5.4 ABLATION STUDY

We conduct ablation studies to analyze the contribution of HAT-VTR's core components. We first analyze our proposed HSM module. As shown in Tab. 6, we test how integrating HSM at two key stages—*Target Selection* and *Posterior Reranking*—affects performance. Using HSM at either stage alone improves results over the baseline, with reranking providing a larger boost. Applying HSM at both stages yields the best performance, confirming that the two mechanisms are complementary and effective. Furthermore, Fig. 5 (a)(b) shows the performance across different HSM's hyperparameters on both the MSRVTT and ActivityNet datasets. The con-

Table 6: Ablation study of the HSM integration at *Target Selection* and *Posterior Reranking*.

| Target | Rerank | v2t | t2v | Avg. |
|---|---|---|---|---|
| | | 23.3 | 32.6 | 28.0 |
| ✓ | | 23.6 | 32.8 | 28.2 |
| | ✓ | 25.5 | 34.3 | 29.9 |
| ✓ | ✓ | 25.8 | 34.5 | 30.1 |

sistent trends observed across both benchmarks validate the stability of our hyperparameter choices. Based on this analysis, we set $(\alpha, \beta, m)$ to $(100, 10, 0.5)$.

Next, we study the impact of each component in our multi-granular adaptation loss. Tab. 7 presents the results of combining different loss terms. We observe that each component contributes positively to the final performance. Specifically, both the multi-granular uniformity losses ($\mathcal{L}_{\text{inter}}$, $\mathcal{L}_{\text{intra}}$) and the cross-modal alignment losses ($\mathcal{L}_{\text{global}}$, $\mathcal{L}_{\text{frame}}$) are beneficial. The noise-robust entropy term ($\mathcal{L}_{\text{NA}}$)

Table 8: Runtime comparison (ms per query) across methods. All measurements are on an RTX 4090 GPU with a batch size of 16.

| Method | CLIP4Clip | Tent | READ | SAR | EATA | TCR | HAT-VTR |
|---|---|---|---|---|---|---|---|
| Runtime(ms) | 2.25 | 26.26 | 26.78 | 53.41 | 26.58 | 26.37 | 32.27 |

Table 9: Component-wise runtime breakdown (ms per query) of HAT-VTR.

| Process | forward | backward | rm | hsm | loss_calculation | total |
|---|---|---|---|---|---|---|
| Runtime(ms) | 2.23 | 21.2 | 2.45 | 4.2 | 2.19 | 32.27 |
| Percentage | 6.9% | 65.7% | 7.6% | 13.0% | 6.8% | 100% |

also provides a clear improvement. The best results are achieved when all components are used together, validating the design of our adaptation loss.

**Visualization** To understand the effective nature of multi-granular TCR learning, we visualize the corrupted query embedding space using t-SNE in Fig. 5(c). Without adaptation, the query embeddings suffer from representation collapse and cluster tightly (dashed circle). TCR alleviates this by enforcing uniformity to spread the embeddings, but some local clustering remains (dashed rectangle). Our HAT-VTR achieves an even more uniform distribution by using fine-grained information. Crucially, the center of its embeddings also shifts significantly closer to the gallery center, indicating a better query-gallery alignment that explains its superior retrieval performance.

## 5.5 EFFICIENCY ANALYSIS

We analyze the computational overhead of HAT-VTR to demonstrate its practical applicability. As shown in Table 8, our HAT-VTR (32.27ms per query) remains highly competitive with other TTA methods like TCR (26.37ms) and EATA (26.58ms) when run on an NVIDIA RTX 4090 (batch size 16). This is a minimal and justifiable cost considering the substantial robustness gains observed across all query-shift and query-gallery-shift scenarios.

To pinpoint this overhead, Table 9 provides a component-wise breakdown. The gradient computation for the backward pass, common to most TTA methods, a standard component common to most TTA methods, is the primary time consumer (65.7%). Critically, our core contribution, the Hubness Suppression Memory (HSM) module, accounts for only 13.0% of the total runtime (4.2ms). This analysis confirms that HAT-VTR's significant robustness gains are achieved with a minimal and justifiable computational cost, primarily through an efficient hubness suppression mechanism.

Additional implementation details are available in Appendix B, ablation studies in C, comparative results under QS and QGS settings with more datasets in D, and limitations and future work in F.

Table 7: Ablation Study of Adaptation Loss.

| $\mathcal{L}_{inter}$ | $\mathcal{L}_{intra}$ | $\mathcal{L}_{global}$ | $\mathcal{L}_{frame}$ | $\mathcal{L}_{NA}$ | v2t | t2v | Avg. |
|---|---|---|---|---|---|---|---|
| | | | | | 22.6 | 34.1 | 28.3 |
| ✓ | | | | | 23.9 | 34.1 | 29.0 |
| ✓ | ✓ | | | | 24.3 | 34.3 | 29.3 |
| | | ✓ | | | 23.5 | 34.1 | 28.8 |
| | | ✓ | ✓ | | 23.9 | 34.2 | 29.0 |
| | | | | ✓ | 24.1 | 34.2 | 29.2 |
| ✓ | ✓ | | | ✓ | 25.3 | 34.4 | 29.8 |
| | | ✓ | ✓ | ✓ | 25.0 | 34.3 | 29.6 |
| ✓ | ✓ | ✓ | ✓ | ✓ | 25.8 | 34.5 | 30.1 |

## 6 CONCLUSION

In this work, we address the vulnerability of VTR models to real-world, spatio-temporal *query shifts*. We introduce the MLVP benchmark to diagnose this failure and uncover an amplified *hubness phenomenon* as the primary cause. To mitigate this, we propose HAT-VTR, a test-time adaptation framework that directly counteracts hubness using a Hubness Suppression Memory and multi-granular losses. Extensive experiments show HAT-VTR substantially improves robustness, consistently outperforming prior methods across diverse test scenarios. Our work thus contributes both a principled benchmark for systematic evaluation and a strong, hubness-aware solution, paving the way for VTR systems that are significantly more robust and reliable in real-world applications.

ACKNOWLEDGEMENTS

This work is supported by Australian Research Council (ARC) Discovery Project DP230101753, CSIRO's Science Leader Project R-91559, and Responsible AI Research Centre Research Theme 2.

REPRODUCIBILITY STATEMENT

We are committed to the reproducibility of our research. We provide the source code, environment setup instructions, and a step-by-step guide to reproduce our results in our Github repository. Further details on our hyperparameter choices and the conceptual design of our benchmark can be found in the appendices (Appendix B and E).

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

APPENDIX CONTENTS

# A  LITERATURE REVIEW

## A.1  VIDEO-TEXT RETRIEVAL

Video-Text Retrieval (VTR) is a fundamental cross-modal task that aims to match visual content with natural language descriptions. It encompasses two reciprocal subtasks: video-to-text retrieval (*v2t*), where a video query retrieves the best matching textual description and text-to-video retrieval (*t2v*), where a text query is used to find the most relevant video from a large gallery. The dominant paradigm for this task has evolved significantly. Early approaches often employed single-stream,

unified models (Gabeur et al., 2020; Patrick et al., 2021; Bain et al., 2021) that fed concatenated video and text features into a shared transformer for joint reasoning, but these were often computationally intensive. Propelled by the success of CLIP (Radford et al., 2021) and BLIP (Li et al., 2022), the field has largely converged on dual-encoder architectures (Luo et al., 2022; Gorti et al., 2022; Ma et al., 2022; Deng et al., 2023; Wang et al., 2023b; Shen et al., 2025; Bai et al., 2025b; Zhang et al., 2025b). These models learn a shared embedding space by independently encoding the video and text modalities and then aligning their representations using a contrastive learning objective. This process, typically conducted via large-scale pre-training on web data followed by downstream fine-tuning, is highly efficient for retrieval as gallery embeddings can be pre-computed. However, the success of these models hinges on a critical assumption: that test data is clean and shares the same distribution as the training data, overlooking their vulnerability to real-world domain shifts.

**Alignment Strategies in VTR.** To compute similarity within the shared embedding space, VTR models employ alignment strategies at different granularities. *Coarse-grained* alignments (Luo et al., 2022; Xue et al., 2022b; Deng et al., 2023) are more efficient approachs, where frame-level features are pooled into a single global vector to represent the entire video. This global video representation is then matched against the global text representation. While highly efficient for large-scale retrieval, these methods can miss nuanced temporal details. Conversely, *fine-grained* alignments (Gorti et al., 2022; Ma et al., 2022; Wang et al., 2023b; Liu et al., 2022; Li et al., 2023; Zhao et al., 2022; Zhang et al., 2025a) seek to capture more detailed interactions by comparing local features, such as frame-word or clip-word pairs, often using cross-attention mechanisms before aggregating the local similarity scores. Although these strategies offer a more detailed comparison, they define the mechanics of similarity computation under the ideal condition that the features themselves are robust and the training pairs are correctly matched.

**Noisy Correspondence in VTR.** A distinct line of research focuses on enhancing model robustness against noisy correspondences (NC) within the *training set* (Huang et al., 2024; Liu et al., 2024; Ma et al., 2024a; Lai et al., 2025; Dang et al., 2025). The web-crawled datasets (Miech et al., 2019; Xue et al., 2022a; Bain et al., 2021) used for pre-training often contain mismatched video-text pairs (i.e., label noise). NC methods aim to mitigate the impact of this noise during the training phase, for instance, by identifying and down-weighting corrupted samples or by using robust loss functions, thereby improving generalization to a clean test set. This focus is orthogonal to our work. While NC methods address label noise during *training*, we tackle the challenge of adapting a pre-trained model to handle input corruptions at *test time*, a scenario where the model is already trained on clean data but deployed in a shifted domain.

**Hubness in VTR.** A key challenge inherent to high-dimensional nearest neighbor search is the *hubness phenomenon* (Radovanovic et al., 2010; Jian & Wang, 2023), where a few gallery items—the "hubs"—become the nearest neighbors for a disproportionately large number of queries, thereby harming retrieval accuracy. Existing solutions aim to mitigate this intrinsic bias through two primary approaches. The first is training-time regularization, which modifies the contrastive loss function to encourage a more uniform or isotropic embedding space, thus preventing hubs from forming (Liu et al., 2020; Liu & Ye, 2019; Lin et al., 2025; Cheng et al., 2021). The second approach involves inference-time post-processing, which adjusts similarity scores after retrieval. Methods like CLIP-ViP with DSL (Xue et al., 2022b) and query-bank normalization (Bogolin et al., 2022; Wang et al., 2023a; Xue et al., 2022b) re-rank results by analyzing neighborhood statistics to demote over-popular items. While effective in standard settings, these methods were not designed to handle the drastic distributional changes caused by test-time corruptions.

A critical limitation of all prior work—spanning alignment strategies, noisy correspondence, and hubness mitigation—is its focus on the standard i.i.d. setting. These approaches assume that test data shares the same clean distribution as the training data. While Test-Time Adaptation (TTA) has emerged as a promising solution for domain shifts, its application in cross-modal retrieval (Li et al., 2025b) has so far been limited to static images and has not addressed the hubness problem. In contrast, we are the first to reveal that real-world video corruptions dramatically exacerbate the hubness phenomenon and propose a TTA framework specifically designed to counteract this critical failure mode in VTR.

## A.2 TEST-TIME ADAPTATION

Test-Time Adaptation (TTA) aims to adapt a pre-trained source model to a target domain using only unlabeled test data, enhancing robustness against distribution shifts encountered during inference. The dominant paradigm, named online or fully test-time adaptation, operates on a stream of test data without access to the source training set. A pioneering work in this area, TENT (Wang et al., 2021), introduced entropy minimization as a self-supervised objective, updating batch normalization parameters to increase the model's prediction confidence on target data. Building on this principle, subsequent research has focused on improving the stability and efficiency of adaptation. For instance, some methods employ active sample selection to update the model only on reliable, low-entropy samples, thereby preventing error accumulation and reducing computational overhead (Niu et al., 2022). Others address the inherent instability of TTA in challenging real-world scenarios—such as mixed domains or small batch sizes—by proposing techniques like sharpness-aware minimization to find flatter minima that are more robust to noisy updates (Niu et al., 2023). The limitations of entropy as a sole confidence metric have also been explored, leading to novel self-supervision signals, such as using object-destructive transformations to better disentangle features and guide adaptation (Lee et al., 2024).

While most TTA research has focused on unimodal classification, recent efforts have started to extend these ideas to multi-modal and retrieval tasks. READ (Yang et al., 2024) was the first to tackle TTA for multi-modal scenarios, identifying a unique *reliability bias* across modalities and proposing an adaptive fusion mechanism to counter it. More directly related to our work, TCR (Li et al., 2025b) pioneered TTA for cross-modal retrieval by addressing the *query shift* problem. It enforces representation uniformity during inference to stabilize the shared embedding space. However, these foundational works primarily address static image-text retrieval and focus on specific issues like modality alignment or query distribution, without investigating the distinct failure modes, such as exacerbated hubness, that emerge in the video domain under spatio-temporal corruptions.

**Relation to Other Adaptation Paradigms.** It is crucial to distinguish our online TTA setting from other adaptation paradigms. Unsupervised Domain Adaptation (UDA) (Liu et al., 2021; Li et al., 2024a; Chen et al., 2021; Hao & Zhang, 2023), for example, also aims to adapt models to an unlabeled target domain but typically assumes offline access to the entire target dataset, allowing for global distribution alignment. This setting is less practical for real-time applications where data arrives as a stream. Another line of research involves non-standard TTA settings that relax the strict online assumption. For instance, Test-Time Training (TTT) (Sun et al., 2020; Gandelsman et al., 2022) and its variants require modifying the pre-training phase to include an auxiliary self-supervised task, which is then leveraged for adaptation at test time. Other approaches rely on external memory banks or retrieval mechanisms (Zancato et al., 2023; Ma et al., 2024b) to source relevant samples for adaptation. In contrast, our work operates under the challenging yet practical online TTA setting, where the model must adapt on-the-fly without any modifications to the original training pipeline or reliance on external data sources, a scenario that closely mirrors real-world deployment.

## A.3 VISION CORRUPTION BENCHMARKS

The systematic evaluation of model robustness is built upon a strong foundation of corruption benchmarks, though this field has historically been dominated by image-level analysis. The seminal work on ImageNet-C (Hendrycks & Dietterich, 2019) established a standard for evaluating image classifier resilience by introducing a comprehensive suite of 15 algorithmically generated corruptions (e.g., noise, blur, weather) at varying severity levels. This principled approach was subsequently extended to other vision tasks like object detection with benchmarks such as COCO-C and Cityscapes-C (Michaelis et al., 2019), and was further adapted to probe the texture versus shape bias of CNNs (Geirhos et al., 2018). Later, with the rise of Large Multimodal Models (LMMs) (OpenAI, 2023), benchmarks like R-Bench (Li et al., 2025a), MMCBench (Zhang et al., 2024), and the multi-modal robustness benchmark by Qiu et al. (2024) have emerged to assess their resilience. A unifying characteristic of these influential works is their focus on static images. Perturbations are typically applied on a frame-by-frame basis, neglecting the temporal dimension and inter-frame dependencies that are fundamental to video.

Recently, research has begun to address this gap by extending robustness analysis to the video domain. One line of work introduced TemRobBench to specifically evaluate LMMs against temporal inconsistencies, revealing that models often disregard motion dynamics (Liang et al., 2025).

The work most related to ours introduced MSRVTT-P and YouCook2-P, the first large-scale benchmarks for evaluating VTR robustness against both visual and textual perturbations (Schiappa et al., 2022). However, our work differs in several crucial aspects.

- First, their visual perturbations are largely extensions of image-based corruptions applied frame-wise, leading to the inclusion of artifacts like *JPEG compression* that do not holistically capture the dynamic nature of video degradation. In contrast, our benchmark is designed to model perturbations that affect inter-frame relationships, such as object motion and semantic consistency.

- Second, their primary goal is to evaluate the intrinsic robustness of various VTR models, whereas our benchmark is specifically designed to facilitate the study and comparison of test-time adaptation methods under these challenging conditions.

In addition, the official code and data for this earlier work are no longer accessible, precluding direct comparison and further research on its foundation. Therefore, to create a reproducible and more ecologically valid standard, we extend the principles established by image-centric benchmarks (Qiu et al., 2024) to the video domain, proposing a new suite of 12 perturbations that explicitly account for spatio-temporal complexities.

## B    MORE IMPLEMENTATION DETAILS

### B.1    MORE DETAILS ON DATASETS

We conduct our experiments on five standard video-text retrieval benchmarks: MSRVTT, ActivityNet, LSMDC, MSVD and DiDeMo. Due to space constraints in the main paper, we only report the results for the first two datasets. Below we provide further details on all five datasets and our specific experimental setup.

**MSRVTT** The MSR-VTT (Microsoft Research Video-to-Text) dataset (Xu et al., 2016) is a large-scale benchmark for video-text retrieval, consisting of 10,000 YouTube clips and a total of 200,000 natural language captions. Each clip, approximately 10–32 seconds in duration, covers a wide range of real-world scenarios. For our experiments, we adhere to the most widely adopted evaluation protocol and use the MSRVTT-1kA test split, which contains 1,000 video-text pairs for testing (Yu et al., 2018).

**ActivityNet** The ActivityNet dataset Fabian & Niebles (2015) is a large-scale benchmark designed for high-level video understanding, containing around 20,000 YouTube videos. It is particularly used for the task of video-paragraph retrieval. Following standard practices Gabeur et al. (2020); Luo et al. (2022), all individual sentence descriptions for a given video are concatenated into a single paragraph. This setup allows for evaluation at the video-paragraph level. We report our results on the official 'val1' split which contains 4,917 video-paragraph pairs.

**LSMDC** The LSMDC (Large Scale Movie Description Challenge) dataset (Rohrbach et al., 2017) is a benchmark composed of 118,081 video clips extracted from 202 different movies. Each clip ranges from 2 to 30 seconds. The cinematic and narrative complexity of the content makes it a challenging dataset for video-language research. For evaluation, we align with the data processing of prior works (Gorti et al., 2022) and report results on the official test set of 999 video clips.

**MSVD** The MSVD (Microsoft Research Video Description) dataset Chen & Dolan (2011) is a widely-used benchmark containing 1,970 short video clips sourced from YouTube, covering a broad set of open-domain, everyday activities. While the dataset provides rich annotations of approximately 40 English sentences per video, we form the video-text evaluation pairs by selecting the first official caption for each video. Our evaluation is conducted on the standard test partition, which consists of 670 such pairs.

**DiDeMo** The DiDeMo dataset (Hendricks et al., 2018) contains 10K long-form videos from Flickr. For each video, approximately 4 short sentences are annotated in temporal order. We follow existing

works to concatenate these short sentences and evaluate 'paragraph-to-video' retrieval on this benchmark. Our evaluation is conducted on the official test split, which consists of 1,004 video-paragraph pairs (concatenated from 4,021 short captions).

## B.2 MORE DETAILS ON TEST-TIME DOMAIN SHIFT SCENARIOS

In our work, we evaluate model robustness under two primary test-time query domain shift scenarios, which are designed to simulate common real-world challenges.

**Query-Shift (QS)** This is the most fundamental and common scenario, where only the query distribution deviates from the training domain. This setup is designed to measure a VTR model's robustness against the diverse and often imperfect inputs from online users.

- In the *video-to-text (v2t)* task, query shifts can arise from user-provided videos that vary widely in quality or are affected by real-world perturbations such as adverse weather or compression artifacts. We simulate this using the 12 video perturbations from our MLVP benchmark.
- In the *text-to-video (t2v)* task, shifts can originate from user search queries containing grammatical errors, typos, or different stylistic expressions (e.g., formal vs. casual tone, active vs. passive voice). Following TCR (Li et al., 2025b), we simulate this using 15 standard text perturbations from Qiu et al. (2024).

**Query-Gallery-Shift (QGS)** This is a more challenging scenario where the distributions of both the query and the gallery data shift simultaneously. This setup models situations where a pre-existing system is deployed to an entirely new environment. We investigate two distinct and practical settings under QGS:

- *Cross-Dataset Adaptation.* This setting simulates the deployment of a model that was fine-tuned on a specific dataset (e.g., Dataset A) to a new application domain that is related but different. To emulate this, we evaluate the model's ability to transfer from one benchmark dataset to another at test time.
- *Zero-Shot Adaptation.* This represents an even more difficult scenario where a model must be deployed without any task-specific fine-tuning. It corresponds to a real-world case where only a general pre-trained model (e.g., CLIP) is available to serve a new collection of data where user query patterns are unknown. We simulate this by directly adapting the pre-trained model on a new downstream retrieval task without it having been fine-tuned on any related data.

## B.3 MORE EXPERIMENT DETAILS

Our framework is implemented on top of the official codebase of X-Pool [1]. The implementations of TTA baselines are adapted from the official repositories of TCR and EATA for fair comparison. During inference, all methods process queries online with a fixed batchsize of 16. Retrieval results are computed and recorded immediately after each batch in a single evaluation pass. For the learning rate, we set it to $3 \times 10^{-5}$ for TENT, READ, and SAR in all scenarios. For EATA and TCR, we use a learning rate of $3 \times 10^{-4}$. For our HAT-VTR, the learning rate is set to $3 \times 10^{-4}$ for the *v2t* task. For the *t2v* task, we use a slightly lower learning rate of $3 \times 10^{-5}$ due to the adjustment of the $\mathcal{L}_{MGUNI}$ objective. For the hyperparameters in our HSM module, we set the temperature values $(\alpha, \beta)$ to $(100, 10)$ and the balancing term $m$ to $0.5$.

**Ablation and Visualization Settings.** Without loss of generality, all ablation studies and visualizations presented in the main paper were conducted on the MSRVTT-1kA dataset under the Query-Shift (QS) scenario, using CLIP4Clip as the base model . To ensure a comprehensive yet efficient evaluation, we selected a representative subset of perturbations. For video perturbations (*v2t*), we report the average results across six types: Gaussian, H.264 Compression, Motion Blur, Main Object Occlusion, Style Transfer, and Event Insertion. Similarly, for text perturbations (*t2v*), we selected six types: OCR, CD, SR, WI, Formal, and Active. This selection was designed to cover perturbations from different hierarchical levels while accelerating the experimental process.

---

[1]https://github.com/layer6ai-labs/xpool

### B.4 DETAILS ON TEXT PERTURBATIONS

For the text-to-video (*t2v*) query-shift experiments, we adopted the comprehensive suite of 15 text perturbations proposed by Qiu et al. (2024). (See Tab. 10) These perturbations are categorized into three hierarchical levels: character-level, word-level, and sentence-level. The original benchmark defines severity degrees from 1 to 7 for character- and word-level perturbations, while sentence-level perturbations have a single degree. In our experiments, we used a fixed severity degree for each level to ensure consistency: severity 7 for character-level, severity 2 for word-level, and the default degree 1 for sentence-level perturbations.

These methods, detailed in Qiu et al. (2024), simulate a wide range of common errors and stylistic variations in user-generated text queries. For instance, the sentence-level perturbation 'Back Translation' (Backtrans.) involves translating a sentence into another language (e.g., German) and then translating it back to the original language (English) via Ng et al. (2019), a process which often introduces grammatical or stylistic variations while preserving the core meaning.

Table 10: Overview of the 15 text perturbation methods from Qiu et al. (2024), illustrated with an example from the MSRVTT-1kA dataset where perturbed segments are highlighted.

| Perturbation Method | Abbr. | Example |
|---|---|---|
| **Original** | | |
| Clean Text | – | a person is connecting something to system |
| **Character Level** | | |
| Optical Character Recognition | OCR | a person is connecting something t0 system |
| Character Insertion | CI | a Qperso(n is connecting something to syˆsteum |
| Character Replacement | CR | a peGsen is connecting something to qysoem |
| Character Swap | CS | a person is ocnnectngi soemthngi to system |
| Character Deletion | CD | a person is connec[t]i[n]g something to [s]yste[m] |
| **Word Level** | | |
| Synonym Replacement | SR | a somebody is connecting something to system |
| Word Insertion | WI | a person is group a connecting something to system |
| Word Swap | WS | a person is connecting to something system |
| Word Deletion | WD | a person is connecting [something] to system |
| Insert Punctuation | IP | ! a person is connecting something to system |
| **Sentence Level** | | |
| Back Translation | Bracktrans. | a person connects something to the system; |
| Formal Style | Formal | A person is connecting something to a system. |
| Casual Style | Casual | a person is connecting something to system |
| Passive Voice | Passive | something is connected by a person to system |
| Active Voice | Active | a person connects something to system |

## C MORE ABLATION STUDY RESULTS

In this part, we provide a more detailed analysis of our model's hyperparameters and offer additional qualitative results to further validate the effectiveness and efficiency of our proposed HAT-VTR framework.

### C.1 MORE HYPERPARAMETERS ANALYSIS

We conduct a thorough hyperparameter sensitivity analysis in Fig. 6 to determine the optimal settings for our method. Our investigation into the temperature parameter $t$ from Eq. 7 8 reveals that

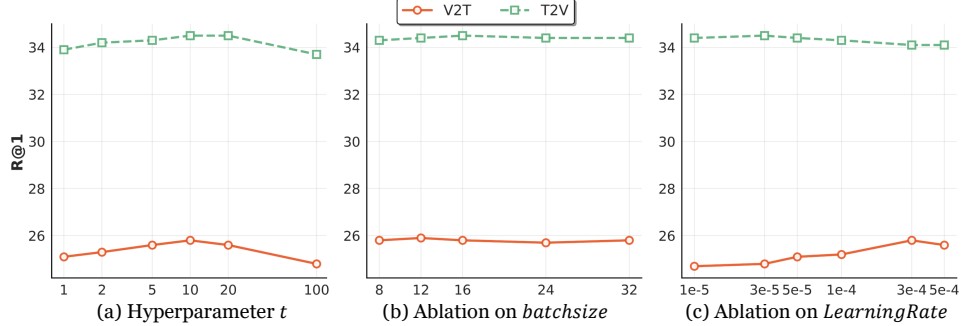

Figure 6: Ablation study on parameter $t$ from Eq. 7 8; *batchsize* and *LearningRate* of our method.

performance peaks at $t = 10$, which we adopt for our experiments. The model demonstrates remarkable stability across various batch sizes $B$. To ensure a fair comparison, we fix the batch size to 16 for all TTA methods. Consequently, we set the learning rate to $3 \times 10^{-4}$ for *v2t* and $3 \times 10^{-5}$ for *t2v*, as these values yield the best performance for each respective direction.

Fig. 7 illustrates the impact of other key hyperparameters,showing results across both MSRVTT and ActivityNet to demonstrate stability.Fig. 7 (a) analyzes $\tau$ Eq. 3), with optimal performance observed around $r = 0.02$, which we use in our experiments. Fig. 7(b) and (c) study the memory bank sizes for the Hubness Suppression Memory (HSM) and the Reliable Memory (RM). For the HSM size $K$ , performance is highly stable across the tested range of 50 to 300 on both datasets. This contrasts with the previous figure and confirms that the model is not sensitive to this choice. Considering the computational overhead, we set $K = 100$, which provides an excellent balance between performance and efficiency. For the RM , performance also remain stable across the tested range from 8 to 32, demonstrating robustness to this hyperparameter. We select a size of 16 for all experiments. These results highlight that our model achieves stable performance with reasonable parameter settings while being practical for deployment.

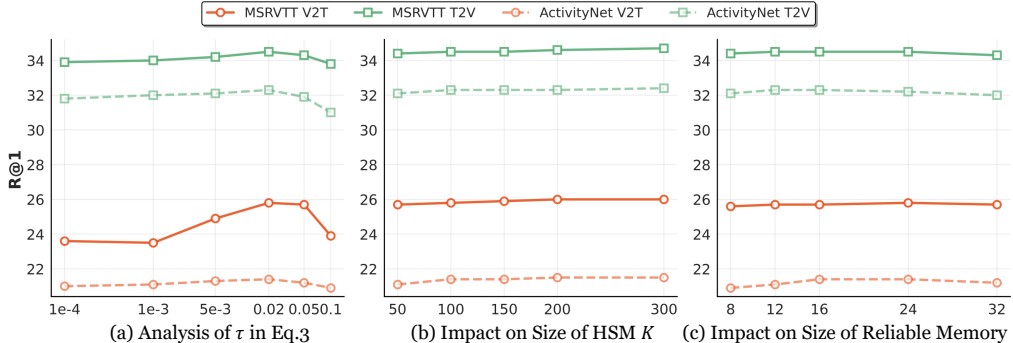

Figure 7: Ablation study of $\tau$, the memory bank size from HSM and Reliable Memory.

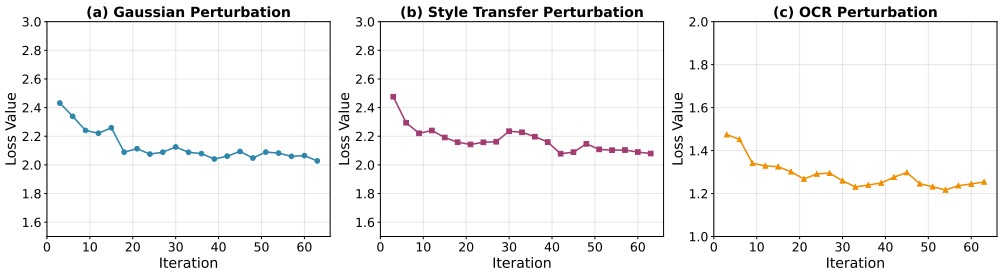

Figure 8: Loss Convergence of HAT-VTR under Different Perturbations.

**Loss Convergence**    We validate our adaptation stability in Figure 8, which plots the total adaptation loss convergence in MSRVTT dataset. The loss is shown over test-time iterations under three diverse perturbations: (a) Gaussian, (b) Style Transfer, and (c) OCR. In all scenarios, the loss rapidly converges to a stable value, demonstrating the robustness and efficiency of our test-time adaptation framework.

Table 11: Stability analysis of HAT-VTR across various random seeds.

| seed | MSRVTT | | ActivityNet | |
|---|---|---|---|---|
| | v2t | t2v | v2t | t2v |
| 42 | 25.8 | 34.5 | 21.4 | 32.3 |
| 0 | 25.8 | 34.6 | 21.2 | 32.5 |
| 100 | 25.9 | 34.7 | 21.3 | 32.3 |
| 200 | 25.5 | 34.4 | 21.4 | 32.4 |
| 512 | 25.6 | 34.4 | 21.4 | 32.3 |

**Seed Analysis**    To verify that our method's performance is stable and not sensitive to random initialization, we conduct experiments using five different random seeds (0, 42, 100, 200, and 512). Our main experiments follow the default seed 42 from the X-Pool codebase, which ensures our results are fully reproducible. We report the average R@1 on the same ablation subsets of MSRVTT and ActivityNet used in the main paper. As shown in Table 11, the performance metrics for both v2t and t2v tasks across both datasets show negligible variance. For example, the MSRVTT v2t score only varies between 25.5% and 25.9%, and the ActivityNet t2v score ranges from 32.3% to 32.5%. This high consistency strongly demonstrates the stability and reliability of our HAT-VTR framework.

Table 12: Sensitivity analysis for HSM's $\alpha$ and $\beta$ hyperparameters across MSRVTT and ActivityNet.

| $\alpha, \beta$ | MSRVTT | | ActivityNet | |
|---|---|---|---|---|
| | v2t | t2v | v2t | t2v |
| 10, 10 | 24.7 | 32.1 | 20.2 | 31.7 |
| 10, 100 | 24.7 | 32.0 | 20.5 | 31.9 |
| 10, 500 | 24.8 | 34.2 | 20.6 | 31.9 |
| **100,10** | 25.8 | 34.5 | 21.4 | 32.3 |
| 100, 100 | 25.5 | 34.4 | 21.4 | 32.4 |
| 100, 500 | 25.6 | 33.7 | 21.3 | 32.1 |
| 500,10 | 24.3 | 33.0 | 20.8 | 31.8 |
| 500, 100 | 24.3 | 33.0 | 20.9 | 31.5 |
| 500,500 | 24.4 | 32.9 | 20.1 | 31.5 |

**Sensitivity Analysis of $\alpha, \beta$**    Table 12 provides the detailed ablation data for the HSM hyperparameters $\alpha$ and $\beta$, supplementing the visualization in Fig. 5(a). The analysis is conducted on both MSRVTT and ActivityNet. The results show that our chosen setting, $(\alpha, \beta) = (100, 10)$, consistently yields the best performance for both v2t and t2v tasks across both datasets. This comprehensive validation confirms the stability and robustness of this hyperparameter choice.

## C.2    Visualization of Hubness Phenomenon and Effectiveness of HSM

To provide a qualitative understanding of our method's ability to mitigate hubness, we visualize the video-text similarity matrices on the MSRVTT-1kA dataset in Fig. 9. The visualization clearly shows that under Gaussian noise, the original similarity matrix (b) suffers from significant off-diagonal noise, where incorrect pairs exhibit high similarity scores—a direct manifestation of the hubness problem. While the baseline TTA method, TCR, partially reduces this noise (c), our HAT-VTR method (d) achieves a substantially more effective suppression of these spurious similarities. This significant improvement is primarily driven by our novel Hubness Suppression Memory (HSM), which greatly suppresses the hubness phenomenon by directly targeting and refining the similarity

scores. The resulting matrix features a much cleaner diagonal pattern, indicating that our approach successfully reinforces the ground-truth video-text correspondences while effectively reducing the hubness effect that plagues baseline models.

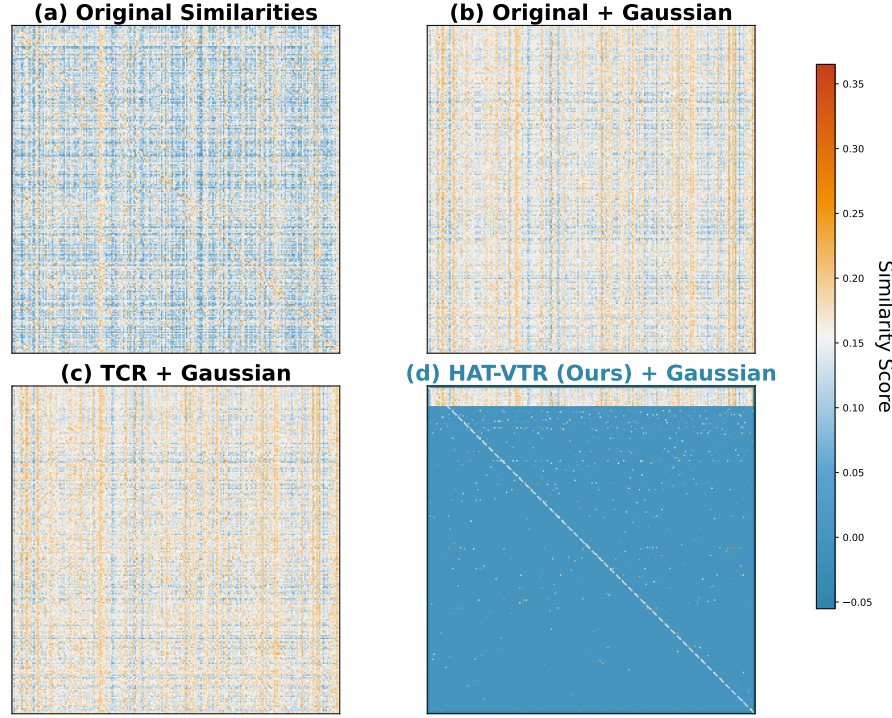

Figure 9: Visualization of similarity matrices for video-text retrieval on MSRVTT-1kA dataset. We show the first 250 *v2t* samples across four methods: (a) Original similarities, (b) Original + Gaussian noise, (c) TCR + Gaussian noise, and (d) Our HAT-VTR method. The diagonal represents ground truth video-text correspondences. Our HAT-VTR method demonstrates significantly reduced hubness effect with cleaner diagonal patterns and suppressed off-diagonal noise compared to baseline methods.

Fig. 10 further validates the superiority of our framework by analyzing the accuracy of the Reliable Memory (RM) during the dynamic test-time adaptation process. Across three distinct and challenging perturbation scenarios—Gaussian Noise, Motion Blur, and Style Transfer—our HAT-VTR consistently maintains a higher RM accuracy compared to the TCR baseline. This demonstrates that our Hubness-Aware Target Selection mechanism is more effective at identifying and storing truly reliable query-gallery pairs, even under severe distribution shifts. This robust memory update process is crucial for preventing catastrophic forgetting and ensuring stable adaptation, which in turn leads to superior retrieval performance.

### C.3 FURTHER ANALYSIS OF HUBNESS EFFECT ON QUERY SHIFT

To quantitatively assess hubness, we utilize distribution-based metrics (skewness(skew) (Radovanovic et al., 2010), truncated skewness(trunc) (Tomašev, 2014), Atkinson index(akinson) (Fischer & Lundtofte, 2020), Robin Hood index(robin) (Feldbauer et al., 2018)) and occurrence-based metrics (antihub(anti) (Radovanović et al., 2014) and hub occurrence(hub) (Radovanovic et al., 2010)), following prior work (Lin et al., 2025)

Table 13 presents a detailed comparison of these metrics. The results confirm our hypothesis: query shifts (e.g., Gaussian noise) dramatically exacerbate the hubness phenomenon, leading to high skewness and hub occurrence in the baseline model. While TCR offers partial mitigation, our HAT-VTR framework consistently and significantly reduces hubness across all tested video and text perturbations, restoring a much more balanced retrieval distribution.

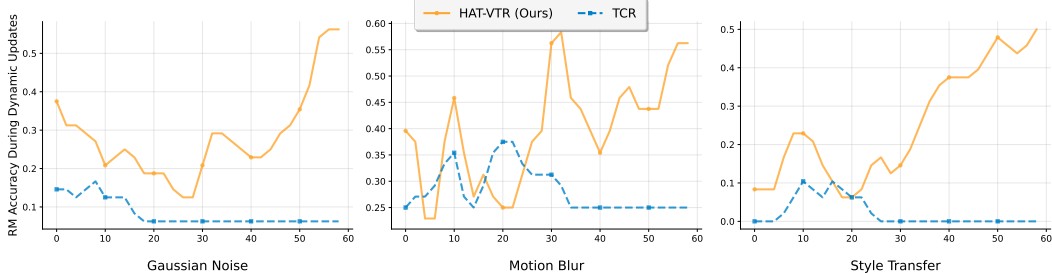

Figure 10: Reliable Memory (RM) accuracy during dynamic updates across different perturbation scenarios on MSRVTT-1kA. The x-axis represents the number of RM update steps during test-time, and the y-axis shows the RM accuracy between video-text pairs. Our HAT-VTR consistently outperforms TCR across all perturbation scenarios, demonstrating superior robustness in maintaining reliable memory accuracy during dynamic updates.

Table 13: Comparaison of Hubeness Metrics on Different Perturbations.

| Metrics | gaussian | | | motion_blur | | | temporal_scrambling | | | ocr | | | backtrans | | |
|---|---|---|---|---|---|---|---|---|---|---|---|---|---|---|---|
| | CLIP4Clip | TCR | Ours | CLIP4Clip | TCR | Ours | CLIP4Clip | TCR | Ours | CLIP4Clip | TCR | Ours | CLIP4Clip | TCR | Ours |
| skew ↓ | 9.09 | 5.07 | **0.97** | 16.27 | 5.25 | **0.86** | 2.19 | 1.8 | **1.15** | 1.95 | 1.84 | **0.39** | 1.36 | 1.19 | **1.12** |
| trunc ↓ | 5.05 | 2.12 | **0.63** | 2.22 | 1.55 | **0.6** | 1.11 | 1.18 | **0.53** | 1.19 | 1.04 | **0.34** | 0.72 | 0.8 | **0.5** |
| atkinson ↓ | 0.6 | 0.27 | **0.05** | 0.23 | 0.15 | **0.05** | 0.08 | 0.1 | **0.06** | 0.12 | 0.11 | **0.05** | 0.08 | 0.07 | **0.06** |
| robin ↓ | 0.64 | 0.41 | **0.18** | 0.38 | 0.31 | **0.18** | 0.38 | 0.31 | **0.18** | 0.27 | 0.26 | **0.18** | 0.22 | 0.21 | **0.19** |
| anti ↓ | 0.19 | 0.04 | **0** | 0.02 | **0** | **0** | 0.02 | 0.004 | **0** | **0** | **0** | **0** | **0** | **0** | **0** |
| hub ↓ | 10.25 | 6.58 | **0.94** | 6.11 | 4.54 | **1.07** | 2.78 | 3.32 | **1.24** | 3.74 | 3.16 | **0.59** | 2.63 | 1.88 | **1.15** |

This superior hubness suppression translates directly to performance gains. As shown in Table 14, our TTA-based optimization outperforms prior training-based hubness suppression methods like NeighborRetr (Lin et al., 2025) and QBNorm (Bogolin et al., 2022) on both CLIP4Clip and Xpool. This demonstrates that an adaptive, test-time solution like HAT-VTR is essential for achieving robust retrieval under query shifts.

## C.4  MORE COMPUTATIONAL EFFICIENCY ANALYSIS

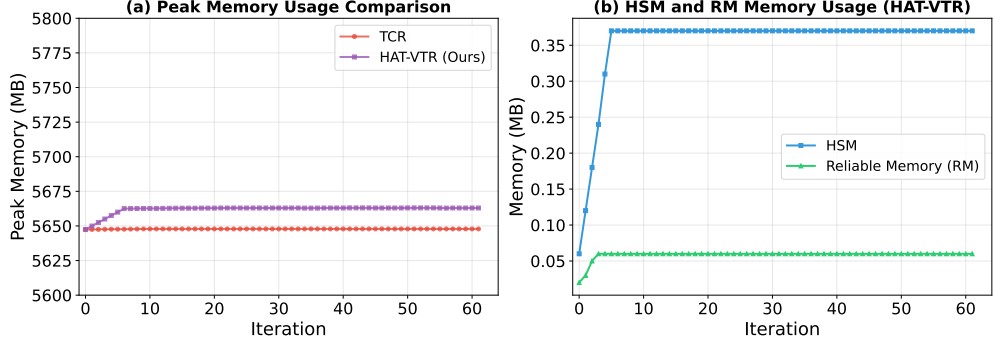

Figure 11: GPU Memory Usage during Test-time Adaptation on MSRVTT-1kA. (a) Peak memory usage comparison between HAT-VTR and TCR. (b) Memory footprint of the HSM and Reliable Memory (RM) components in HAT-VTR, demonstrating their low and stable consumption.

The memory overhead of HAT-VTR primarily stems from the HSM module, which maintains a queue of recent similarity matrices. As shown in Fig. 11(a), the peak memory usage of HAT-VTR is only negligibly higher (approx. 17MB) than the TCR baseline. Fig. 11(b) further details the memory consumption of our new components, showing the HSM and Reliable Memory (RM) are highly efficient, consuming less than 0.4MB and 0.1MB, respectively. The memory usage scales linearly with the queue size rather than the total dataset size, ensuring that our method remains practical for large-scale retrieval scenarios.

Table 14: Video Perturbations with Different Hubness-suppression Methods.

| v2t | type | gaussian | h264_compression | motion_blur | main_object_occlusion | style_transfer | event_insertion | Avg. |
|---|---|---|---|---|---|---|---|---|
| CLIP4Clip_ori | - | 8.7 | 26.2 | 24.6 | 22.5 | 4.8 | 22.2 | 18.2 |
| NeighborRetr | Training-based | 18.5 | 28.1 | 26.4 | 29.9 | 10.9 | 26.5 | 23.4 |
| HSM+QBNorm | Training-free | 22.5 | 28.6 | 31.2 | 29.3 | 11.8 | 25.6 | 24.8 |
| CLIP4Clip+ours | TTA-Optimization | 23.1 | 30.6 | 32.6 | 30.1 | 12.1 | 26.3 | 25.8 |
| Xpool+ours | TTA-Optimization | **26.2** | **35.6** | **35.3** | **35.5** | **14.4** | **35.2** | **30.4** |

| t2v | type | ocr | char_delete | synonym_replace | word_insert | formal | active | Avg. |
|---|---|---|---|---|---|---|---|---|
| CLIP4Clip_ori | - | 21.5 | 11.2 | 38.9 | 39.0 | 40.9 | 41.8 | 32.2 |
| NeighborRetr | Training-based | 23.8 | 12.8 | 40.2 | 40.8 | 45.4 | 46.2 | 34.9 |
| HSM+QBNorm | Training-free | 23.6 | 12.1 | 40.4 | 39.8 | 42.7 | 42.8 | 33.6 |
| CLIP4Clip+ours | TTA-Optimization | 24.5 | 12.8 | 40.7 | 41.6 | 43.6 | 43.7 | 34.5 |
| Xpool+ours | TTA-Optimization | **26.9** | **14.7** | **44.8** | **43.8** | **48.7** | **48.1** | **37.8** |

Table 15: Wall-clock time (in seconds) for different TTA methods to process the entire MSRVTT-1kA test set on NVIDIA RTX 4090 and RTX A6000 GPUs.

| | RTX4090 | | RTX A6000 | |
|---|---|---|---|---|
| | v2t | t2v | v2t | t2v |
| Tent | 36.27s | 14.27s | 62.05s | 24.11s |
| READ | 36.73s | 14.85s | 62.72s | 24.51s |
| SAR | 63.09s | 16.54s | 113.27s | 28.61s |
| EATA | 36.58s | 14.58s | 62.6s | 24.44s |
| TCR | 36.38s | 14.52s | 62.34s | 24.36s |
| ours | 37.06s | 14.77s | 65.41s | 24.76s |

We further report the total wall-clock time to process the entire MSRVTT-1kA test set in Table 15. The experiments are conducted on both NVIDIA RTX 4090 and RTX A6000 GPUs. Our method (HAT-VTR) demonstrates comparable efficiency to other TTA baselines like TCR and EATA. For instance, on the RTX 4090, our method incurs only a minor computational overhead (37.06s vs. 36.38s for TCR in the v2t task), confirming that the substantial robustness gains come at a very low cost in terms of inference speed.

These analyses confirm that HAT-VTR achieves substantial performance improvements with minimal computational and memory overhead, making it a practical solution for real-world video-text retrieval applications.

# D   MORE COMPARISON RESULTS

## D.1   GENERALIZABILITY ACROSS FOUNDATION MODELS

To validate the generalizability of our framework, we apply TTA to four different foundation models, with results shown in Tab. 16. We conduct experiments on MSRVTT-1Ka (v2t) using CLIP-ViT-B/32, the stronger CLIP-ViT-B/16, the vision-language model BLIP-ViT-B/16, and the recent universal embedding model, LanguageBind Zhu et al. (2024). The results clearly demonstrate that HAT-VTR consistently and substantially outperforms all baselines across all architectures. Notably, our method improves the average R@1 score over the strongest baseline (TCR) from 14.4% to 21.3% on CLIP-ViT-B/32, from 16.5% to 23.6% on BLIP-ViT-B/16, and from 22.6% to 27.7% on LanguageBind. This confirms that amplified hubness is a common failure mode and that our hubness-aware adaptation is a generalizable solution for enhancing the robustness of diverse VTR models.

## D.2   MIXED VIDEO PERTURBATIONS IN MLVP

While our MLVP benchmark systematically evaluates robustness under controlled single-type perturbations with fixed severity degrees, real-world scenarios present a more complex challenge where multiple perturbations of varying severities occur simultaneously. To assess the practical robustness of HAT-VTR under such realistic conditions, we conduct additional experiments with mixed video perturbations that better reflect the complexity of real-world deployment scenarios.

Tab. 17 presents the results under mixed severity degrees, where videos within the same batch are corrupted with the same perturbation type but with different severity levels ranging from 1 to 5. This

Table 16: Comparisons on *v2t* R@1 on the MSRVTT Dataset with Different Foundation Models.

| Query | Low-Level | | | | | | Mid-Level | | | High-Level | | | |
|---|---|---|---|---|---|---|---|---|---|---|---|---|---|
| Shift | Gauss. | Impul. | Fog | Snow | Elastic. | H264. | Motion | Defocus | Occlu. | Style | Event | Tempo. | Avg. |
| CLIP-ViT-B/32 | 7.1 | 4.9 | 16.1 | 10.6 | 7.9 | 14.2 | 11.8 | 3.6 | 8.7 | 4.3 | 16.7 | 22.8 | 10.7 |
| • Tent | 5.5 | 4.3 | 15.3 | 9.3 | 7.4 | 14.4 | 14.5 | 3.6 | 8.5 | 3.2 | 17.4 | 22.8 | 10.5 |
| • READ | 7.4 | 5.0 | 17.0 | 11.1 | 7.8 | 14.7 | 9.9 | 3.4 | 8.7 | 4.4 | 15.5 | 22.1 | 10.6 |
| • SAR | 6.4 | 4.7 | 16.0 | 9.9 | 8.3 | 14.6 | 14.2 | 3.9 | 8.8 | 3.6 | 17.4 | 22.9 | 10.9 |
| • EATA | 4.1 | 2.6 | 17.1 | 3.2 | 1.0 | 15.1 | 15.6 | 2.7 | 10.1 | 1.8 | 18.5 | 23.1 | 9.6 |
| • TCR | 12.0 | 9.3 | 21.8 | 13.3 | 14.2 | 17.1 | 17.5 | 6.6 | 12.5 | 6.3 | 18.7 | 23.8 | 14.4 |
| • Ours | **18.4** | **15.9** | **29.7** | **22.7** | **24.2** | **25.0** | **26.1** | **13.1** | **22.6** | **7.6** | **23.4** | **27.4** | **21.3** |
| CLIP-ViT-B/16 | 5.5 | 6.8 | 19.9 | 15.4 | 3.7 | 15.6 | 14.0 | 4.0 | 12.5 | 3.2 | 17.8 | 25.8 | 12.0 |
| • Tent | 3.1 | 4.9 | 19.9 | 15.3 | 2.9 | 14.3 | 15.2 | 3.2 | 11.4 | 2.7 | 18.1 | 26.0 | 11.4 |
| • READ | 6.2 | 7.3 | 19.8 | 15.7 | 4.2 | 15.5 | 12.9 | 4.0 | 12.2 | 4.0 | 15.9 | 24.6 | 11.9 |
| • SAR | 4.7 | 6.2 | 20.2 | 15.6 | 3.3 | 14.5 | 15.1 | 3.6 | 12.4 | 2.8 | 18.0 | 25.8 | 11.9 |
| • EATA | 4.9 | 3.7 | 21.1 | 15.5 | 1.0 | 15.7 | 16.7 | 1.5 | 13.6 | 1.1 | 18.3 | 26.7 | 11.7 |
| • TCR | 11.6 | 11.0 | 26.5 | 19.0 | 12.5 | 18.1 | 19.0 | 8.1 | 14.2 | 5.0 | 17.3 | 24.8 | 15.6 |
| • Ours | **17.4** | **18.0** | **32.6** | **26.1** | **19.3** | **26.6** | **26.4** | **15.4** | **26.4** | **9.4** | **25.6** | **30.6** | **22.8** |
| BLIP-ViT-B/16 | 6.7 | 7.4 | 20.6 | 16.3 | 4.5 | 17.2 | 15.1 | 4.7 | 12.9 | 4.5 | 18.4 | 26.4 | 12.9 |
| • Tent | 3.6 | 5.2 | 20.3 | 16.2 | 3.7 | 15.6 | 16.1 | 4.1 | 12.8 | 4.1 | 18.8 | 26.4 | 12.2 |
| • READ | 6.9 | 7.8 | 20.1 | 16.6 | 5.0 | 17.0 | 15.1 | 4.8 | 12.8 | 4.6 | 17.9 | 25.8 | 12.9 |
| • SAR | 5.3 | 7.0 | 21.1 | 16.6 | 4.1 | 16.6 | 16.3 | 4.5 | 13.1 | 3.9 | 19.2 | 26.8 | 12.9 |
| • EATA | 5.1 | 3.9 | 23.2 | 16.2 | 1.5 | 16.2 | 17.6 | 2.2 | 14.3 | 1.4 | 18.4 | 26.9 | 12.2 |
| • TCR | 11.9 | 12.5 | 28.1 | 19.7 | 13.0 | 18.8 | 20.2 | 8.5 | 15.5 | 5.3 | 18.8 | 25.2 | 16.5 |
| • Ours | **17.9** | **19.2** | **34.0** | **26.5** | **20.2** | **27.6** | **26.7** | **15.3** | **28.2** | **10.0** | **26.1** | **31.5** | **23.6** |
| LanguageBind | 14.4 | 15.0 | 32.9 | 23.1 | 11.2 | 27.6 | 27.3 | 5.5 | 27.5 | 7.1 | 23.2 | 32.6 | 20.6 |
| • Tent | 12.2 | 11.1 | 32.0 | 22.3 | 8.5 | 27.5 | 27.6 | 3.3 | 28.0 | 3.7 | 24.2 | 32.6 | 19.4 |
| • READ | 15.3 | 15.8 | 33.4 | 23.0 | 11.8 | 27.8 | 27.4 | 6.5 | 26.8 | 7.7 | 23.2 | 33.0 | 21.0 |
| • SAR | 14.1 | 14.1 | 33.1 | 23.7 | 9.7 | 27.7 | 27.4 | 4.2 | 28.0 | 4.9 | 24.1 | 32.9 | 20.3 |
| • EATA | 16.5 | 7.1 | 35.2 | 16.5 | 4.5 | 29.4 | 27.6 | 0.4 | 30.8 | 4.3 | 22.5 | 32.9 | 19.0 |
| • TCR | 18.4 | 19.5 | 32.3 | 27.1 | 15.5 | 29.1 | 28.5 | 8.0 | 28.0 | 9.9 | 22.5 | 31.8 | 22.6 |
| • Ours | **24.7** | **23.9** | **39.3** | **31.9** | **24.8** | **32.5** | **32.3** | **15.1** | **33.0** | **12.5** | **28.9** | **33.3** | **27.7** |

setting simulates scenarios where environmental conditions vary in intensity across different video samples. HAT-VTR demonstrates consistent superiority across all perturbation categories, achieving an average improvement of 3.5% R@1 points over TCR on CLIP4Clip.

Table 17: Comparisons *v2t* results on the MSRVTT-1kA with mixed severity degrees.

| Query | Low-Level | | | | | | Mid-Level | | | High-Level | | | |
|---|---|---|---|---|---|---|---|---|---|---|---|---|---|
| Shift | Gauss. | Impul. | Fog | Snow | Elastic. | H264. | Motion | Defocus | Occlu. | Style | Event | Tempo. | Avg. |
| CLIP4Clip | 25.5 | 13.8 | 33.6 | 20.8 | 29.2 | 38.7 | 32.3 | 18.3 | 28.2 | 18.0 | 23.4 | 35.6 | 26.5 |
| • Tent | 26.6 | 10.7 | 34.8 | 21.4 | 29.6 | 38.9 | 33.7 | 18.8 | 28.7 | 18.3 | 23.8 | 36.4 | 26.8 |
| • READ | 24.5 | 16.1 | 33.0 | 20.2 | 28.3 | 38.9 | 32.0 | 18.0 | 27.9 | 18.1 | 22.7 | 37.3 | 26.4 |
| • SAR | 27.0 | 12.3 | 35.0 | 22.0 | 29.6 | 39.0 | 33.6 | 19.4 | 28.7 | 18.9 | 23.4 | 38.1 | 27.3 |
| • EATA | 29.7 | 18.8 | 36.4 | 24.3 | 30.6 | 38.6 | 34.5 | 21.8 | 30.6 | 19.6 | 24.5 | **37.3** | 28.9 |
| • TCR | 30.7 | 23.4 | 35.5 | 28.4 | 30.6 | 38.5 | 34.1 | 22.1 | 30.5 | 20.1 | 24.5 | 37.8 | 29.7 |
| • Ours | **34.8** | **31.1** | **39.1** | **33.3** | **35.2** | **39.4** | **37.0** | **25.0** | **35.5** | **22.2** | **27.9** | **38.1** | **33.2** |
| Xpool | 28.3 | 15.8 | 36.9 | 23.2 | 35.1 | 40.8 | 33.3 | 20.4 | 34.7 | 20.1 | 35.1 | 36.9 | 30.1 |
| • Tent | 30.3 | 13.1 | 38.3 | 24.4 | 35.7 | 41.4 | 34.4 | 22.2 | 35.9 | 20.4 | 35.4 | 36.8 | 30.7 |
| • READ | 27.6 | 17.2 | 35.5 | 23.1 | 34.3 | 40.7 | 32.3 | 19.7 | 33.9 | 20.0 | 34.9 | 36.8 | 29.7 |
| • SAR | 30.2 | 15.5 | 38.1 | 24.4 | 35.8 | 41.1 | 34.5 | 21.9 | 35.6 | 20.9 | 35.5 | 36.9 | 30.9 |
| • EATA | 34.2 | 24.0 | 40.6 | 30.3 | 36.3 | 41.3 | 35.5 | 23.8 | 35.8 | 23.8 | 36.0 | 37.5 | 33.3 |
| • TCR | 32.0 | 25.4 | 40.5 | 28.6 | 37.2 | 41.2 | 35.4 | 23.3 | 36.3 | 22.9 | 36.4 | 37.1 | 33.0 |
| • Ours | **38.7** | **32.5** | **42.9** | **34.6** | **38.2** | **43.4** | **38.2** | **28.3** | **38.8** | **27.4** | **37.3** | **39.6** | **36.7** |

To further challenge the robustness of our method, we evaluate performance under mixed perturbation types, where different percentages of queries in each batch are corrupted with randomly selected perturbations from our MLVP benchmark. Tab. 18 reports the results across different noise ratios, from 20% to 80% of queries being corrupted. This setting represents the most realistic deployment scenario where various types of corruptions occur unpredictably. HAT-VTR maintains superior performance across all noise ratios, achieving an average improvement of 3.5% R@1 points over TCR. Crucially, our method shows strong resilience even at high corruption ratios, maintaining 27.6% R@1 when 80% of queries are corrupted compared to TCR's 21.7%, demonstrating the robustness of our hubness suppression approach under diverse and unpredictable perturbation patterns.

Table 18: Performance under mixed perturbation types with varying percentages of corrupted queries. Results show R@1 (%) on MSRVTT-1kA where different ratios of queries are randomly corrupted with perturbations from our MLVP benchmark.

| Query | *Percentage of Noised Queries* | | | | |
|---|---|---|---|---|---|
| Shift | 20% | 40% | 60% | 80% | Avg. |
| CLIP4Clip | 36.7 | 31.1 | 27.2 | 21.0 | 29.0 |
| • Tent | 36.3 | 31.1 | 27.2 | 21.5 | 29.0 |
| • READ | 36.1 | 30.7 | 26.9 | 21.0 | 28.7 |
| • SAR | 36.4 | 31.1 | 27.3 | 21.4 | 29.1 |
| • EATA | 36.6 | 31.6 | 27.7 | 23.7 | 29.9 |
| • TCR | 36.1 | 30.5 | 26.5 | 21.7 | 28.7 |
| • Ours | **37.4** | **34.4** | **29.3** | **27.6** | **32.2** |
| Xpool | 38.4 | 33.6 | 28.8 | 23.8 | 31.2 |
| • Tent | 39.0 | 34.1 | 29.2 | 23.7 | 31.5 |
| • READ | 38.5 | 33.7 | 28.5 | 23.4 | 31.0 |
| • SAR | 38.9 | 34.2 | 29.3 | 23.8 | 31.6 |
| • EATA | 39.9 | 34.9 | 29.3 | 25.9 | 32.5 |
| • TCR | 38.6 | 34.1 | 29.3 | 24.3 | 31.6 |
| • Ours | **41.6** | **37.6** | **32.3** | **29.0** | **35.1** |

## D.3 MORE RESULTS ON QUERY-SHIFT

To further validate the effectiveness of our proposed HAT-VTR framework, we conduct additional experiments on more datasets under query-shift scenarios. The results consistently demonstrate the superiority of our approach across diverse evaluation settings.

For *v2t* retrieval, we evaluate on LSMDC and MSVD datasets with MLVP video perturbations. On the LSMDC dataset (Tab. 19), HAT-VTR achieves substantial improvements over all baselines, with particularly notable gains on high-level perturbations such as style transfer (7.61% vs. 4.80% for the best baseline TCR with CLIP4Clip). The challenging MSVD dataset (Tab. 20) reveals even more pronounced advantages, where our method delivers consistent improvements across all perturbation categories. Notably, HAT-VTR demonstrates exceptional robustness against low-level corruptions, achieving 33.28% R@1 on Gaussian noise compared to 28.06% for TCR, highlighting our framework's ability to effectively counteract the amplified hubness phenomenon under diverse video corruptions.

We further observe a similar trend for *v2t* retrieval on the DiDeMo dataset (Tab. 24). On this dataset, HAT-VTR again demonstrates exceptional robustness. For instance, on the CLIP4Clip benchmark, our method (23.11%) achieves a significant R@1 improvement of 5.18% over TCR (17.93%). Finally, we further validate the *t2v* task performance on Tab. 25. The results consistently show that HAT-VTR achieves the best average performance across all three perturbation levels (character-, word-, and sentence-level) on both CLIP4Clip and X-Pool models

For *t2v* retrieval under text perturbations, we evaluate on ActivityNet, LSMDC, and MSVD datasets. On ActivityNet (Tab. 21), HAT-VTR consistently outperforms all baselines across character-level, word-level, and sentence-level perturbations, achieving an average improvement of over 4% compared to the strongest baseline. The LSMDC results (Tab. 22) and MSVD results (Tab. 23) further confirm this trend, with our method showing particular strength on word-level perturbations where semantic understanding is crucial. These comprehensive results across multiple datasets and perturbation types validate that our hubness-aware adaptation strategy provides robust performance improvements regardless of the specific corruption mechanism or dataset characteristics.

## D.4 MORE RESULTS ON QUERY-GALLERY-SHIFT

### D.4.1 QGS IN CROSS-DATASET ADAPTATION SCENARIO

We conduct extensive cross-dataset adaptation experiments across five dataset pairs, each presenting unique domain shift challenges that test our method's robustness across diverse video-text domains.

Table 19: Comparisons on *v2t* R@1 on the LSMDC dataset with MLVP video perturbations.

| Query | Low-Level | | | | | | Mid-Level | | | High-Level | | | |
|---|---|---|---|---|---|---|---|---|---|---|---|---|---|
| Shift | Gauss. | Impul. | Fog | Snow | Elastic. | H264. | Motion | Defocus | Occlu. | Style | Event | Tempo. | Avg. |
| CLIP4Clip | 10.61 | 8.61 | 3.60 | 6.11 | 13.51 | 0.40 | 14.01 | 12.61 | 5.11 | 4.40 | 6.91 | 11.21 | 8.09 |
| • Tent | 10.91 | 9.41 | 2.40 | 4.90 | 14.11 | 0.10 | 13.71 | 12.71 | 4.60 | 4.40 | 7.21 | 11.31 | 7.98 |
| • READ | 10.41 | 7.91 | 4.90 | 6.01 | 12.91 | 0.30 | 13.71 | 12.31 | 5.11 | 4.50 | 6.51 | 11.31 | 7.99 |
| • SAR | 11.01 | 9.61 | 2.70 | 5.41 | 14.01 | 0.30 | 13.61 | 12.61 | 5.31 | 4.40 | 7.31 | 11.41 | 8.14 |
| • EATA | 12.01 | 11.11 | 1.20 | 6.61 | 15.22 | 0.10 | 14.61 | 12.61 | 5.21 | 5.11 | 7.11 | 11.71 | 8.55 |
| • TCR | 11.61 | 11.51 | 6.21 | 7.91 | 14.21 | **0.80** | 13.81 | 12.71 | 5.51 | 4.80 | 6.41 | 11.91 | 8.95 |
| • Ours | **13.81** | **14.21** | **9.51** | **11.51** | **16.62** | **0.80** | **16.02** | **15.12** | **8.71** | **7.61** | **9.91** | **13.01** | **11.40** |
| Xpool | 9.31 | 7.21 | 4.00 | 5.31 | 14.51 | 0.90 | 13.71 | 11.81 | 6.51 | 4.20 | 10.21 | 11.01 | 8.22 |
| • Tent | 9.71 | 8.01 | 2.70 | 5.01 | 15.22 | 0.80 | 13.91 | 12.21 | 5.91 | 4.10 | 10.31 | 11.31 | 8.27 |
| • READ | 9.21 | 6.71 | 4.20 | 5.41 | 13.71 | 1.00 | 13.41 | 11.71 | 6.61 | 4.40 | 9.91 | 10.71 | 8.08 |
| • SAR | 9.91 | 8.11 | 2.70 | 5.21 | 15.02 | 0.90 | 14.21 | 12.11 | 6.31 | 4.30 | 10.31 | 11.21 | 8.36 |
| • EATA | 11.61 | 11.31 | 0.60 | 4.50 | 14.81 | 0.30 | 15.12 | 13.21 | 6.11 | 5.11 | 10.61 | 11.01 | 8.69 |
| • TCR | 11.51 | 9.81 | 5.51 | 7.01 | 14.31 | 1.10 | 14.81 | 12.41 | 7.71 | 5.51 | 10.61 | 11.31 | 9.30 |
| • Ours | **14.31** | **13.61** | **9.71** | **10.81** | **16.52** | **1.30** | **16.32** | **15.42** | **7.81** | **8.71** | **13.21** | **12.41** | **11.68** |

Table 20: Comparisons on *v2t* R@1 on the MSVD dataset with MLVP video perturbations.

| Query | Low-Level | | | | | | Mid-Level | | | High-Level | | | |
|---|---|---|---|---|---|---|---|---|---|---|---|---|---|
| Shift | Gauss. | Impul. | Fog | Snow | Elastic. | H264. | Motion | Defocus | Occlu. | Style | Event | Tempo. | Avg. |
| CLIP4Clip | 23.88 | 19.40 | 37.01 | 20.15 | 26.27 | 21.04 | 41.04 | 20.75 | 20.75 | 14.48 | 26.87 | 48.36 | 26.67 |
| • Tent | 25.07 | 21.04 | 37.91 | 20.75 | 27.46 | 22.24 | 41.49 | 21.19 | 21.04 | 14.03 | 27.01 | 47.61 | 27.24 |
| • READ | 23.28 | 17.76 | 36.42 | 20.00 | 25.97 | 21.94 | 40.30 | 19.85 | 20.30 | 14.78 | 27.31 | 48.51 | 26.37 |
| • SAR | 24.48 | 21.04 | 37.01 | 21.34 | 27.61 | 22.24 | 41.49 | 21.34 | 21.04 | 14.48 | 27.01 | 48.21 | 27.27 |
| • EATA | 27.76 | 25.52 | 40.90 | 24.78 | 32.39 | 25.37 | 45.37 | 24.33 | 22.09 | 16.27 | 28.66 | 47.76 | 30.10 |
| • TCR | 28.06 | 25.07 | 42.69 | 24.93 | 32.84 | 23.43 | 44.78 | 23.88 | 22.24 | 17.76 | 28.96 | 47.46 | 30.18 |
| • Ours | **33.28** | **34.78** | **46.57** | **34.93** | **37.46** | **28.21** | **49.10** | **29.25** | **25.97** | **22.09** | **36.12** | **49.70** | **35.62** |
| Xpool | 26.42 | 21.34 | 41.94 | 25.07 | 32.84 | 25.52 | 41.49 | 24.48 | 28.21 | 20.15 | 43.43 | 49.70 | 31.72 |
| • Tent | 28.21 | 22.54 | 42.69 | 24.93 | 33.58 | 25.07 | 42.09 | 25.22 | 27.61 | 20.75 | 43.73 | 49.85 | 32.19 |
| • READ | 25.67 | 21.34 | 41.64 | 25.07 | 32.39 | 25.52 | 41.94 | 23.73 | 28.51 | 20.00 | 44.03 | 49.85 | 31.64 |
| • SAR | 28.21 | 23.28 | 42.39 | 25.37 | 33.13 | 25.97 | 42.09 | 25.37 | 28.06 | 20.45 | 43.58 | **50.15** | 32.34 |
| • EATA | 30.00 | 28.21 | 42.84 | 28.36 | 36.12 | 26.27 | 43.28 | 25.97 | 28.66 | 22.09 | 44.18 | 49.85 | 33.82 |
| • TCR | 32.09 | 26.72 | 43.58 | 28.06 | 34.78 | 24.93 | 43.88 | 26.42 | 28.96 | 20.90 | 45.37 | 49.40 | 33.76 |
| • Ours | **36.57** | **35.37** | **46.42** | **35.22** | **43.13** | **30.00** | **45.82** | **30.45** | **31.34** | **26.72** | **48.66** | 48.96 | **38.22** |

The MSRVTT→LSMDC and LSMDC→MSRVTT transfers (Tab. 26) represent a shift between general YouTube content and cinematic movie clips. HAT-VTR achieves substantial improvements, with 17.42% R@1 compared to 14.41% for TCR on MSRVTT→LSMDC, and an even more pronounced gain of 36.60% versus 32.60% on the reverse direction. This asymmetric performance reflects the semantic complexity difference between everyday YouTube videos and narrative-driven movie scenes.

The MSRVTT→MSVD and MSVD→MSRVTT transfers (Tab. 27) involve two YouTube-based datasets with different scales and annotation styles. Our method demonstrates consistent superiority, achieving 55.67% R@1 on MSRVTT→MSVD versus 52.69% for TCR, and a remarkable 37.50% versus 33.10% on MSVD→MSRVTT. The larger gains on the MSVD→MSRVTT direction suggest our hubness mitigation is particularly effective when adapting from smaller to larger-scale datasets.

The ActivityNet→LSMDC and LSMD→ActivityNet transfers (Tab. 28) represent perhaps the most challenging domain gap, spanning from paragraph-level activity descriptions to cinematic narratives. HAT-VTR shows exceptional performance improvements, achieving 18.02% R@1 versus 14.81% for TCR on ActivityNet→LSMDC, and a dramatic 31.69% versus 14.34% on the reverse direction, highlighting our method's ability to handle severe semantic distribution shifts.

The ActivityNet→MSVD and MSVD→ActivityNet transfers (Tab. 29) involve adapting between paragraph-level and sentence-level video descriptions. Our method consistently outperforms baselines, with particularly strong performance on MSVD→ActivityNet (34.76% vs. 17.06% for TCR), demonstrating effective adaptation from simple activity descriptions to complex paragraph-level understanding.

Table 21: Comparisons on *t2v* Recall@1 (%) on the ActivityNet dataset under text perturbations.

| Query Shift | Character-Level | | | | | Word-Level | | | | | Sentence-Level | | | | | Avg. |
|---|---|---|---|---|---|---|---|---|---|---|---|---|---|---|---|---|
| | OCR | CI | CR | CS | CD | SR | WI | WS | WD | IP | Backtrans. | Formal | Casual | Passive | Active | |
| CLIP4Clip | 26.52 | 17.53 | 18.20 | 21.50 | 19.79 | 29.81 | 32.03 | 31.65 | 32.19 | 33.35 | 26.83 | 31.42 | 32.11 | 25.24 | 26.42 | 26.97 |
| • Tent | 26.87 | 18.39 | 18.55 | 21.84 | 19.50 | 29.94 | 31.87 | 31.34 | 31.79 | 33.23 | 27.13 | 31.04 | 31.97 | 25.63 | 26.48 | 27.04 |
| • READ | 24.79 | 14.93 | 15.50 | 20.66 | 18.69 | 29.33 | 31.36 | 30.75 | 31.81 | 32.15 | 26.01 | 31.10 | 31.14 | 24.57 | 25.48 | 25.88 |
| • SAR | 26.72 | 18.30 | 18.70 | 21.82 | 19.83 | 30.04 | 32.09 | 31.38 | 32.03 | 33.44 | 27.11 | 31.34 | 32.21 | 25.63 | 26.32 | 27.13 |
| • EATA | 25.87 | 18.55 | 19.06 | 21.29 | 19.28 | 28.41 | 30.97 | 30.73 | 30.95 | 31.95 | 26.99 | 30.00 | 30.69 | 24.73 | 25.71 | 26.35 |
| • TCR | 26.44 | 18.30 | 18.47 | 20.68 | 19.52 | 28.80 | 31.12 | 30.45 | 31.60 | 30.81 | 26.48 | 31.22 | 29.69 | 25.08 | 26.01 | 26.31 |
| • Ours | **30.95** | **20.42** | **21.29** | **25.02** | **23.14** | **35.33** | **37.18** | **36.85** | **37.87** | **39.58** | **32.13** | **36.87** | **37.20** | **29.06** | **30.59** | **31.57** |
| Xpool | 25.46 | 16.86 | 17.63 | 20.52 | 18.77 | 28.53 | 30.02 | 30.14 | 30.77 | 32.01 | 26.85 | 30.40 | 30.26 | 25.67 | 26.83 | 26.05 |
| • Tent | 25.24 | 17.84 | 18.45 | 20.83 | 19.12 | 28.29 | 29.79 | 29.59 | 30.36 | 31.42 | 26.87 | 30.36 | 30.22 | 25.81 | 26.32 | 26.03 |
| • READ | 24.16 | 14.50 | 14.99 | 19.73 | 17.88 | 28.37 | 30.24 | 29.90 | 30.51 | 31.50 | 26.03 | 30.34 | 30.38 | 25.56 | 26.50 | 25.37 |
| • SAR | 25.14 | 17.75 | 18.41 | 20.81 | 19.08 | 28.51 | 30.02 | 29.96 | 30.59 | 31.60 | 26.93 | 30.43 | 30.34 | 25.61 | 26.38 | 26.10 |
| • EATA | 25.16 | 17.82 | 18.10 | 20.89 | 18.63 | 28.00 | 29.14 | 28.90 | 29.25 | 30.77 | 27.01 | 30.18 | 29.90 | 24.93 | 26.19 | 25.66 |
| • TCR | 25.06 | 17.88 | 17.63 | 20.26 | 18.91 | 26.99 | 29.96 | 29.20 | 29.92 | 29.65 | 26.09 | 29.57 | 29.94 | 25.40 | 26.26 | 25.51 |
| • Ours | **28.25** | **19.20** | **20.34** | **23.63** | **21.42** | **31.24** | **33.48** | **33.33** | **33.31** | **34.74** | **29.53** | **33.39** | **33.52** | **27.25** | **28.05** | **28.71** |

Table 22: Comparisons on *t2v* Recall@1 (%) on the LSMCD dataset under text perturbations.

| Query Shift | Character-Level | | | | | Word-Level | | | | | Sentence-Level | | | | | Avg. |
|---|---|---|---|---|---|---|---|---|---|---|---|---|---|---|---|---|
| | OCR | CI | CR | CS | CD | SR | WI | WS | WD | IP | Backtrans. | Formal | Casual | Passive | Active | |
| CLIP4Clip | 9.71 | 3.70 | 4.00 | **5.41** | 4.00 | 13.71 | 14.51 | 14.71 | 15.02 | 14.51 | 12.71 | 14.91 | 14.51 | 15.12 | 15.32 | 11.46 |
| • Tent | 9.51 | 3.60 | 4.00 | **5.41** | 4.10 | 14.01 | 14.41 | 14.71 | 15.52 | 15.02 | 12.71 | 15.02 | 14.91 | 15.02 | 15.52 | 11.56 |
| • READ | 9.61 | **3.80** | **4.10** | 5.31 | 3.90 | 13.71 | 14.51 | 14.51 | 15.22 | 14.61 | 13.01 | 15.02 | 14.61 | 15.22 | 15.32 | 11.50 |
| • SAR | 9.61 | 3.60 | 3.90 | 5.31 | **4.20** | 13.91 | 14.61 | 14.71 | 15.52 | 14.81 | 12.71 | 15.02 | 14.81 | 14.91 | 15.42 | 11.54 |
| • EATA | 9.41 | 3.00 | 3.50 | 5.11 | 4.00 | 13.81 | 14.61 | 14.21 | **15.82** | 14.61 | 12.81 | 15.32 | 15.12 | 14.61 | 15.52 | 11.43 |
| • TCR | 9.71 | 3.00 | **4.10** | 5.31 | **4.20** | 14.01 | 14.71 | 14.61 | 15.42 | 14.91 | 12.91 | 15.22 | 14.81 | 15.32 | 15.12 | 11.56 |
| • Ours | **10.11** | 3.20 | 3.70 | 5.21 | 3.80 | **15.22** | **16.12** | **15.52** | 15.42 | **15.72** | **14.31** | **16.22** | **15.62** | **16.02** | **16.42** | **12.17** |
| Xpool | 10.81 | 4.10 | 4.10 | 5.81 | **4.30** | 14.81 | 16.22 | 15.52 | 16.52 | 16.02 | 15.02 | 16.52 | 16.02 | **16.72** | 16.72 | 12.61 |
| • Tent | 10.71 | 3.90 | 4.20 | 5.81 | 4.10 | 14.71 | 16.02 | 15.42 | **16.72** | 16.42 | 14.71 | 16.52 | 16.12 | 15.52 | 16.72 | 12.51 |
| • READ | 10.51 | **4.40** | 4.00 | 5.71 | 4.10 | 15.02 | 16.22 | 15.42 | 16.42 | 16.12 | 14.81 | 16.22 | 16.12 | 16.52 | 16.72 | 12.55 |
| • SAR | 10.71 | 4.10 | 4.20 | **5.91** | 4.20 | 14.81 | 16.02 | 15.42 | 16.72 | 16.42 | 14.71 | 16.52 | 16.12 | 16.52 | 16.62 | 12.60 |
| • EATA | 11.01 | 3.50 | 4.20 | **5.91** | **4.30** | 14.41 | 16.42 | 15.52 | 16.52 | **17.12** | 14.31 | **16.62** | 16.12 | 16.52 | 16.32 | 12.59 |
| • TCR | 10.81 | 4.20 | 4.30 | 5.71 | 3.90 | 14.41 | **16.52** | 15.12 | 16.52 | 16.32 | 14.51 | 16.32 | 15.82 | 16.62 | 16.62 | 12.51 |
| • Ours | **11.31** | 3.80 | **4.70** | 5.31 | 4.20 | **15.42** | 16.32 | **15.72** | 16.62 | 16.92 | **15.62** | 16.32 | **16.72** | 15.42 | **17.42** | **12.79** |

Finally, the LSMDC→MSVD and MSVD→LSMDC transfers (Tab. 30) span from cinematic content to everyday activities. HAT-VTR maintains its advantage across both directions, achieving 51.94% R@1 versus 46.72% for TCR on LSMDC→MSVD, confirming that our hubness-aware adaptation strategy provides robust cross-domain transfer capabilities regardless of the specific dataset characteristics or domain gap magnitude.

### D.4.2 QGS in Zero-shot Adaptation Scenario

The zero-shot adaptation scenario represents the most challenging setting, where models must adapt directly from pre-training to downstream tasks without any fine-tuning. Tab. 31 presents results on LSMDC and MSVD datasets, revealing that HAT-VTR maintains its effectiveness even in this extreme domain gap scenario.

On the challenging LSMDC dataset, HAT-VTR achieves 15.22% R@1 compared to 10.61% for TCR, representing a remarkable 4.61% improvement in the zero-shot setting. This substantial gain highlights our method's ability to rapidly adapt to new domains without prior exposure to task-specific data. The consistent improvements across both datasets and retrieval directions confirm that our hubness suppression mechanism provides robust adaptation capabilities even when facing the largest possible domain shifts encountered in practical deployment scenarios.

### D.5 Comparison Results with Different Severity Degrees

#### D.5.1 Results on MLVP

To examine robustness progression across perturbation intensities, we evaluate HAT-VTR and baseline methods on MSRVTT-1kA across severity degrees 1-4, complementing the main paper's severity degree 5 results.

Table 23: Comparisons on *t2v* Recall@1 (%) on the MSVD dataset under text perturbations.

| Query | Character-Level | | | | | Word-Level | | | | | Sentence-Level | | | | | |
|---|---|---|---|---|---|---|---|---|---|---|---|---|---|---|---|---|
| Shift | OCR | CI | CR | CS | CD | SR | WI | WS | WD | IP | Backtrans. | Formal | Casual | Passive | Active | Avg. |
| CLIP4Clip | 32.54 | 13.73 | 12.54 | **22.24** | 12.09 | 52.69 | 54.93 | 54.33 | 55.52 | 55.37 | 51.79 | 56.42 | 54.48 | 53.58 | 55.37 | 42.51 |
| • Tent | 32.84 | 13.88 | 12.54 | **22.24** | 12.24 | 52.84 | 54.48 | 54.03 | 55.22 | 54.93 | 51.79 | 56.27 | 54.48 | 53.28 | 55.37 | 42.43 |
| • READ | 32.39 | 13.73 | 12.54 | 21.94 | 12.09 | 52.84 | 54.48 | 54.18 | 55.37 | 55.37 | 51.79 | 56.57 | 54.48 | 53.43 | 55.52 | 42.45 |
| • SAR | 32.99 | 13.88 | **12.69** | 22.24 | 12.24 | 52.84 | 54.63 | 54.03 | 55.37 | 54.93 | 51.79 | 56.42 | 54.48 | 52.99 | 55.52 | 42.47 |
| • EATA | 32.24 | 14.03 | 12.54 | 21.64 | 12.69 | 52.09 | 54.78 | 53.43 | 54.93 | 54.48 | 51.19 | 55.07 | 54.48 | 52.99 | 54.93 | 42.10 |
| • TCR | 32.54 | 13.43 | 12.54 | 22.09 | 12.24 | 53.13 | 54.93 | 53.58 | 54.93 | 55.07 | 52.09 | 55.37 | 55.07 | 53.13 | 55.07 | 42.35 |
| • Ours | **34.18** | **14.33** | 12.24 | 20.90 | **12.99** | **55.07** | **56.27** | **56.87** | **56.87** | **56.27** | **53.88** | **58.36** | **57.01** | **56.27** | **58.36** | **43.99** |
| Xpool | 32.39 | 16.27 | 13.73 | 20.90 | 13.58 | 51.94 | 54.93 | 53.43 | 54.48 | 56.42 | 52.69 | 56.12 | 55.52 | 51.94 | 55.67 | 42.67 |
| • Tent | 32.69 | 16.57 | 14.18 | 21.19 | 13.58 | 51.34 | 54.93 | 53.43 | 54.48 | 56.27 | 52.54 | 56.12 | 55.67 | 51.94 | 55.82 | 42.73 |
| • READ | 32.39 | 16.12 | 13.88 | 20.90 | 13.43 | 51.79 | 54.93 | 53.73 | 54.33 | 56.42 | 52.69 | 56.12 | 55.37 | 51.94 | 55.82 | 42.66 |
| • SAR | 32.69 | 16.57 | 14.03 | 21.19 | 13.58 | 51.49 | 55.07 | 53.43 | 54.48 | 56.42 | 52.54 | 56.27 | 55.52 | 51.94 | 55.82 | 42.74 |
| • EATA | 33.58 | **16.72** | **14.78** | 21.49 | 14.18 | 50.60 | 54.48 | 53.73 | 53.58 | 55.22 | 52.24 | 55.37 | 55.37 | 52.39 | 55.82 | 42.64 |
| • TCR | 33.43 | 15.97 | 14.18 | 22.09 | 12.84 | 50.45 | 55.07 | 52.84 | 53.58 | 55.07 | 52.84 | 56.27 | 55.22 | 51.94 | 55.97 | 42.52 |
| • Ours | **35.07** | 16.57 | 13.43 | **22.39** | **14.63** | **54.33** | **56.42** | **56.42** | **57.61** | **57.91** | **55.37** | **59.10** | **57.76** | **56.87** | **58.51** | **44.83** |

Table 24: Comparisons on *v2t* R@1 on the DiDeMo dataset with the highest severity degree.

| Query | Low-Level | | | | | | Mid-Level | | | High-Level | | | |
|---|---|---|---|---|---|---|---|---|---|---|---|---|---|
| Shift | Gauss. | Impul. | Fog | Snow | Elastic. | H264. | Motion | Defocus | Occlu. | Style | Event | Tempo. | Avg. |
| CLIP4Clip | 6.08 | 6.67 | 21.61 | 12.15 | 11.16 | 32.87 | 22.21 | 5.88 | 13.65 | 4.88 | 1.69 | 30.18 | 14.09 |
| • Tent | 4.28 | 4.38 | 20.72 | 10.86 | 12.75 | 32.87 | 23.21 | 4.48 | 11.25 | 4.58 | 1.49 | 29.88 | 13.40 |
| • READ | 6.87 | 7.67 | 22.91 | 12.65 | 10.66 | 32.67 | 20.02 | 5.88 | 14.24 | 4.78 | 1.79 | 29.68 | 14.15 |
| • SAR | 4.58 | 4.78 | 22.31 | 11.85 | 12.55 | 33.17 | 23.01 | 5.08 | 13.25 | 5.08 | 1.79 | 30.18 | 13.97 |
| • EATA | 5.68 | 2.29 | 24.90 | 10.66 | 16.93 | 33.57 | 25.30 | 6.67 | 14.94 | 5.98 | 1.59 | 29.98 | 14.87 |
| • TCR | 13.15 | 14.54 | 27.29 | 18.73 | 18.03 | 33.76 | 23.80 | 10.76 | 16.33 | 6.77 | 2.79 | 29.18 | 17.93 |
| • Ours | **20.32** | **19.12** | **35.26** | **25.00** | **25.20** | **37.55** | **29.48** | **16.73** | **22.91** | **8.17** | **5.58** | **31.97** | **23.11** |
| Xpool | 9.36 | 10.56 | 25.20 | 15.64 | 15.24 | 37.25 | 24.20 | 6.37 | 18.73 | 7.87 | 29.88 | 30.98 | 19.27 |
| • Tent | 8.67 | 9.16 | 23.51 | 15.64 | 16.73 | 36.95 | 25.80 | 5.98 | 15.24 | 7.57 | 30.38 | 30.98 | 18.88 |
| • READ | 10.16 | 10.76 | 27.19 | 15.84 | 14.34 | 36.85 | 22.91 | 6.57 | 20.32 | 7.97 | 29.58 | 31.08 | 19.46 |
| • SAR | 9.86 | 10.16 | 24.50 | 16.14 | 16.73 | 36.95 | 25.50 | 6.08 | 18.23 | 7.97 | 30.38 | 31.08 | 19.47 |
| • EATA | 13.75 | 9.96 | 26.59 | 12.75 | 19.42 | 37.75 | 26.89 | 7.47 | 21.02 | 6.47 | 31.47 | 31.08 | 20.39 |
| • TCR | 16.93 | 14.84 | 32.37 | 23.21 | 20.02 | 38.05 | 27.79 | 10.76 | 23.61 | 7.57 | 30.28 | 30.68 | 23.01 |
| • Ours | **20.32** | **22.21** | **37.75** | **27.69** | **30.78** | **39.74** | **33.17** | **17.53** | **26.39** | **11.55** | **33.96** | **33.37** | **27.87** |

Tables 32 33 34 35 reveal consistent patterns: HAT-VTR maintains superior performance across all severity levels, with improvement margins typically increasing as perturbations intensify. At severity degree 1 (Tab. 32), our method achieves 39.8% average R@1 versus 36.5% for TCR with CLIP4Clip, demonstrating effectiveness even under mild corruptions. The performance gap widens progressively—at severity degree 4 (Tab. 35), HAT-VTR reaches 32.2% compared to 27.0% for TCR, representing a 5.2% improvement.

We acknowledge certain exceptions, such as temporal scrambling at severity degree 1, where EATA scores 39.3% compared to our 38.8%. This highlights a need for further exploration in addressing mild temporal disruptions (see Sec. F.1 for more details). Nonetheless, our method clearly excels across various perturbation types and intensities overall.

### D.5.2    RESULTS ON TEXT PERTURBATIONS

To comprehensively evaluate text-to-video (t2v) retrieval robustness, we examine HAT-VTR performance across different severity levels of text perturbations on MSRVTT-1kA. Following the hierarchical text perturbation framework from Qiu et al. (2024), we test character-level (OCR, CI, CR, CS, CD) and word-level (SR, WI, WS, WD, IP) corruptions across severity degrees 1-7.

Fig. 12 demonstrates HAT-VTR's consistent superiority across all perturbation types. HAT-VTR achieves 45.3% average R@1 compared to 43.3% for TCR, with the most significant improvements observed on word-level perturbations such as synonym replacement (SR) and word insertion (WI), where our method reaches over 40% R@1. Character-level perturbations (OCR, CI, CR, CS, CD) show more modest improvements but consistent gains, with performance around 23-28% R@1. The results indicate that HAT-VTR's hubness mitigation strategy is particularly effective for semantic-preserving word-level transformations while maintaining robustness across all perturbation categories.

Table 25: Comparisons on *t2v* Recall@1 (%) on the DiDeMo dataset under text perturbations.

| Query Shift | Character-Level | | | | | Word-Level | | | | | Sentence-Level | | | | | Avg. |
|---|---|---|---|---|---|---|---|---|---|---|---|---|---|---|---|---|
| | OCR | CI | CR | CS | CD | SR | WI | WS | WD | IP | Backtrans. | Formal | Casual | Passive | Active | |
| CLIP4Clip | 28.39 | 19.72 | 20.82 | 21.12 | 21.51 | 34.56 | 36.35 | 35.26 | 35.76 | 37.75 | 27.79 | 34.46 | 35.76 | 34.46 | 34.46 | 30.54 |
| ● Tent | 28.09 | 19.42 | 21.31 | 21.41 | 21.71 | 34.06 | 36.16 | 35.36 | 35.46 | 37.75 | 27.99 | 34.36 | 35.86 | 34.56 | 34.36 | 30.52 |
| ● READ | 28.88 | 19.82 | 20.92 | 20.92 | 21.91 | 34.76 | 36.06 | 35.66 | 35.46 | 37.95 | 28.09 | 34.46 | 35.56 | 34.46 | 34.26 | 30.61 |
| ● SAR | 28.19 | 19.52 | 21.22 | 21.41 | 21.91 | 34.26 | 36.16 | 35.26 | 35.66 | 37.75 | 27.89 | 34.46 | 35.56 | 34.76 | 34.06 | 30.54 |
| ● EATA | 27.09 | 19.72 | 20.72 | 21.81 | 20.82 | 33.47 | 35.16 | 34.46 | 34.26 | 37.65 | 27.29 | 34.26 | 34.36 | 33.96 | 33.86 | 29.93 |
| ● TCR | 27.79 | 18.43 | 21.02 | 22.61 | 20.62 | 33.76 | 35.26 | 35.06 | 34.56 | 38.25 | 27.19 | 34.26 | 34.66 | 33.96 | 33.67 | 30.07 |
| ● Ours | **28.89** | **21.51** | **21.91** | **23.31** | **21.81** | **36.25** | **38.15** | **36.75** | **37.35** | **38.55** | **30.58** | **36.46** | **37.55** | **34.26** | **35.46** | **31.92** |
| Xpool | 31.37 | 22.01 | 24.00 | 25.70 | 24.30 | 39.74 | 40.14 | 41.83 | 41.24 | 42.03 | 30.78 | 38.25 | 39.74 | 37.45 | 39.24 | 34.52 |
| ● Tent | 31.47 | 22.41 | **24.60** | 26.10 | 24.50 | 39.64 | 40.44 | 41.73 | 41.04 | 42.33 | 30.98 | 38.94 | 39.84 | 37.75 | 38.75 | 34.70 |
| ● READ | 31.37 | 22.01 | 22.91 | 26.00 | 23.80 | 39.74 | 40.24 | 41.73 | 40.94 | 41.63 | 30.58 | 38.45 | 39.74 | 37.45 | 39.14 | 34.38 |
| ● SAR | 31.47 | 22.71 | 24.30 | 26.20 | 24.50 | 39.54 | 40.64 | 42.13 | 41.33 | 42.43 | 30.78 | 38.84 | 39.84 | 37.55 | 38.75 | 34.73 |
| ● EATA | 31.67 | 23.61 | 23.41 | 26.00 | **24.60** | 40.04 | 40.04 | 41.14 | 40.24 | 41.24 | 30.68 | 38.55 | 39.34 | 37.95 | 39.04 | 34.52 |
| ● TCR | 31.87 | 22.61 | 24.00 | 25.60 | 24.00 | 38.65 | 40.24 | 40.04 | 40.34 | 40.94 | 29.38 | 38.65 | 38.45 | 37.75 | 37.65 | 34.01 |
| ● Ours | **32.17** | **24.30** | **24.60** | **27.49** | **24.60** | **40.24** | **43.23** | **42.73** | **43.23** | **44.42** | **32.17** | **40.44** | **41.04** | **37.95** | **39.94** | **35.90** |

Table 26: Comparisons on QGS *Cross-dataset Adaptation* between MSRVTT and LSMDC.

| Cross Dataset | MSRVTT→LSMDC | | | | | | LSMDC→MSRVTT | | | | | |
|---|---|---|---|---|---|---|---|---|---|---|---|---|
| | v2t | | | t2v | | | v2t | | | t2v | | |
| *Metrics* | **R@1** | **R@5** | **R@10** | **R@1** | **R@5** | **R@10** | **R@1** | **R@5** | **R@10** | **R@1** | **R@5** | **R@10** |
| CLIP4Clip | 14.91 | 28.43 | 36.84 | 15.12 | 30.03 | 38.94 | 30.30 | 57.10 | 66.90 | 29.60 | 57.20 | 67.20 |
| ● Tent | 15.12 | 28.83 | 37.24 | 15.42 | 29.93 | 38.84 | 30.90 | 56.80 | 67.30 | 30.30 | 57.50 | 66.80 |
| ● READ | 14.51 | 28.43 | 36.44 | 15.32 | 30.23 | 38.74 | 29.10 | 56.40 | 66.60 | 29.60 | 57.30 | 67.60 |
| ● SAR | 15.32 | 29.23 | 37.44 | 15.32 | 29.93 | 38.94 | 30.80 | 56.80 | 67.30 | 30.50 | 57.80 | 67.20 |
| ● EATA | 15.82 | 29.83 | 38.74 | 15.52 | 29.93 | 39.34 | 31.60 | 58.50 | 68.40 | 31.70 | 58.30 | 68.00 |
| ● TCR | 14.41 | 30.23 | 38.04 | 15.22 | 30.23 | 38.94 | 32.60 | 58.60 | 68.60 | 31.10 | 58.60 | 68.60 |
| ● Ours | **17.42** | **33.13** | **40.04** | **16.32** | **32.83** | **40.94** | **36.60** | **61.70** | **72.00** | **36.70** | **61.10** | **72.30** |
| Xpool | 16.82 | 33.33 | 41.24 | 16.82 | 33.33 | 41.24 | 29.80 | 54.00 | 65.20 | 32.60 | 57.00 | 67.00 |
| ● Tent | 17.02 | 33.33 | 41.24 | 17.02 | 33.33 | 41.24 | 29.70 | 53.90 | 65.30 | 32.90 | 57.20 | 67.20 |
| ● READ | 16.82 | 32.93 | 40.94 | 16.82 | 32.93 | 40.94 | 30.20 | 54.10 | 65.10 | 32.50 | 56.90 | 66.90 |
| ● SAR | 16.92 | 33.33 | 41.14 | 16.92 | 33.33 | 41.14 | 29.90 | 53.90 | 65.40 | 32.80 | 57.20 | 67.50 |
| ● EATA | 16.92 | **33.53** | 40.84 | 16.92 | **33.53** | 40.84 | 30.20 | 54.50 | 65.80 | 33.40 | 57.80 | 69.20 |
| ● TCR | 16.82 | **33.53** | 41.04 | 16.82 | **33.53** | 41.04 | 30.10 | 54.50 | 65.60 | 33.00 | 57.20 | 67.70 |
| ● Ours | **17.32** | **33.53** | **42.14** | **17.32** | **33.53** | **42.14** | **35.30** | **60.20** | **69.60** | **36.10** | **61.50** | **71.20** |

Table 27: Comparisons on QGS *Cross-dataset Adaptation* between MSRVTT and MSVD.

| Cross Dataset | MSRVTT→MSVD | | | | | | MSVD→MSRVTT | | | | | |
|---|---|---|---|---|---|---|---|---|---|---|---|---|
| | v2t | | | t2v | | | v2t | | | t2v | | |
| *Metrics* | **R@1** | **R@5** | **R@10** | **R@1** | **R@5** | **R@10** | **R@1** | **R@5** | **R@10** | **R@1** | **R@5** | **R@10** |
| CLIP4Clip | 54.03 | 83.43 | 90.90 | 54.63 | 80.90 | **90.90** | 32.00 | 57.90 | 68.50 | 35.30 | 60.10 | 70.50 |
| ● Tent | 54.03 | 83.58 | 90.90 | 54.48 | 80.90 | 90.90 | 32.50 | 58.50 | 69.30 | **35.70** | 60.00 | 70.80 |
| ● READ | 53.58 | 83.28 | 90.90 | 54.63 | 80.90 | 90.90 | 31.70 | 57.20 | 68.40 | **35.70** | 60.30 | 70.70 |
| ● SAR | 53.73 | 83.28 | 90.75 | 54.48 | 80.90 | 90.90 | 32.20 | 58.50 | 69.60 | 35.60 | 60.10 | 70.80 |
| ● EATA | 53.88 | 82.99 | 90.90 | 54.63 | 81.19 | 90.45 | 33.70 | 60.10 | 71.40 | 35.30 | 60.40 | 70.20 |
| ● TCR | 52.69 | **83.88** | 91.04 | 54.18 | 80.90 | 90.45 | 33.10 | 57.80 | 69.30 | 34.80 | 59.70 | 69.90 |
| ● Ours | **55.67** | 82.99 | **91.49** | 54.78 | **84.33** | 90.45 | **37.50** | **64.40** | **74.10** | 34.00 | **63.20** | **74.00** |
| Xpool | 54.48 | 85.22 | 91.49 | 55.97 | 85.37 | 91.79 | 35.50 | 61.80 | 73.20 | 38.10 | 63.00 | 72.80 |
| ● Tent | 54.63 | 85.22 | 91.79 | 55.97 | 85.52 | 91.79 | 35.70 | 61.80 | 73.60 | 38.10 | 62.60 | 72.80 |
| ● READ | 54.18 | 85.37 | 91.64 | 55.97 | 85.22 | 91.79 | 35.20 | 61.20 | 73.10 | 37.80 | 63.00 | 72.70 |
| ● SAR | 54.33 | 85.22 | 91.64 | 55.97 | 85.52 | 91.79 | 35.80 | 62.60 | 73.70 | 38.00 | 62.80 | 72.80 |
| ● EATA | 54.48 | **85.97** | 91.79 | 55.97 | 85.22 | 91.79 | 36.60 | 62.90 | 75.20 | 38.00 | 62.50 | 73.10 |
| ● TCR | 54.33 | 84.93 | **92.39** | 55.37 | 85.67 | 91.34 | 36.80 | 62.30 | 74.30 | 38.00 | 62.60 | 72.90 |
| ● Ours | **57.91** | **85.97** | **92.39** | **57.46** | **85.97** | **92.84** | **40.40** | **65.60** | **77.30** | **38.70** | **65.20** | **75.10** |

Table 28: Comparisons on QGS *Cross-dataset Adaptation* between ActivityNet and LSMDC.

| Cross | ActivityNet→LSMDC | | | | | | LSMDC→ActivityNet | | | | | |
| Dataset | v2t | | | t2v | | | v2t | | | t2v | | |
| *Metrics* | **R@1** | **R@5** | **R@10** | **R@1** | **R@5** | **R@10** | **R@1** | **R@5** | **R@10** | **R@1** | **R@5** | **R@10** |
|---|---|---|---|---|---|---|---|---|---|---|---|---|
| CLIP4Clip | 13.41 | 27.13 | 34.83 | 16.12 | 28.63 | 35.74 | 21.35 | 46.21 | 59.87 | 20.28 | 44.91 | 58.71 |
| ● Tent | 14.41 | 27.13 | 35.04 | 15.92 | 28.43 | 35.74 | 21.50 | 47.02 | 59.96 | 21.58 | 47.24 | 60.48 |
| ● READ | 13.41 | 26.23 | 34.63 | **16.32** | 28.23 | 35.64 | 11.78 | 29.37 | 41.12 | 17.84 | 39.35 | 53.12 |
| ● SAR | 14.71 | 27.03 | 34.83 | 15.92 | 28.43 | 35.84 | 21.56 | 46.94 | 60.38 | 21.94 | 47.41 | 60.85 |
| ● EATA | 14.41 | 26.73 | 35.24 | 15.92 | 29.33 | 36.84 | 18.95 | 42.02 | 54.91 | 22.45 | 46.82 | 60.85 |
| ● TCR | 14.81 | 27.93 | 37.44 | 16.02 | 29.03 | 36.14 | 14.34 | 33.66 | 44.86 | 22.05 | 47.14 | 61.93 |
| ● Ours | **18.02** | **32.93** | **40.44** | 15.72 | **31.33** | **39.64** | **31.69** | **57.51** | **70.73** | **31.40** | **58.71** | **70.55** |
| Xpool | 14.31 | 28.43 | 36.74 | 15.72 | 31.03 | 37.64 | 18.53 | 40.33 | 54.02 | 19.83 | 44.30 | 56.74 |
| ● Tent | 14.81 | 28.73 | 36.94 | 16.22 | 31.53 | 37.94 | 17.82 | 39.45 | 52.59 | 21.21 | 46.45 | 59.43 |
| ● READ | 14.21 | 27.93 | 36.44 | 15.62 | 30.93 | 37.54 | 15.29 | 36.04 | 49.26 | 17.84 | 39.60 | 51.60 |
| ● SAR | 14.61 | 28.73 | 36.84 | 15.92 | 31.43 | 38.04 | 18.32 | 40.21 | 53.30 | 22.01 | 47.28 | 60.32 |
| ● EATA | 15.22 | 29.03 | 36.54 | 15.72 | **32.23** | 38.04 | 14.72 | 34.86 | 46.04 | 23.55 | 49.40 | 63.11 |
| ● TCR | 15.32 | 29.43 | 36.74 | 15.82 | 31.01 | 37.84 | 18.69 | 40.23 | 53.02 | 21.82 | 47.41 | 60.42 |
| ● Ours | **17.72** | **33.33** | **41.84** | **16.72** | 31.93 | **40.84** | **28.23** | **53.47** | **67.01** | **26.80** | **53.39** | **65.87** |

Table 29: Comparisons on QGS *Cross-dataset Adaptation* between ActivityNet and MSVD.

| Cross | ActivityNet→MSVD | | | | | | MSVD→ActivityNet | | | | | |
| Dataset | v2t | | | t2v | | | v2t | | | t2v | | |
| *Metrics* | **R@1** | **R@5** | **R@10** | **R@1** | **R@5** | **R@10** | **R@1** | **R@5** | **R@10** | **R@1** | **R@5** | **R@10** |
|---|---|---|---|---|---|---|---|---|---|---|---|---|
| CLIP4Clip | 52.99 | 83.13 | 89.70 | 54.18 | 80.15 | 88.21 | 29.00 | 57.64 | 70.82 | 26.36 | 53.51 | 67.52 |
| ● Tent | 53.13 | 83.28 | 89.55 | 54.33 | 80.75 | 87.91 | 29.06 | 58.61 | 71.00 | 26.05 | 53.04 | 67.26 |
| ● READ | 52.99 | 82.84 | 89.70 | 54.33 | 80.15 | 88.21 | 24.45 | 50.62 | 63.07 | 23.82 | 49.46 | 63.17 |
| ● SAR | 53.13 | 82.99 | 89.70 | 54.33 | 80.45 | 87.91 | 29.57 | 58.71 | 71.43 | 26.26 | 53.04 | 67.58 |
| ● EATA | 52.69 | 83.73 | 90.00 | 54.18 | 80.90 | 88.06 | 29.16 | 57.05 | 70.43 | 25.69 | 52.23 | 66.81 |
| ● TCR | 53.28 | 82.54 | 88.81 | 53.73 | 80.75 | 88.36 | 17.06 | 37.60 | 49.05 | 26.46 | 52.43 | 67.32 |
| ● Ours | **55.97** | **83.43** | **90.30** | **56.27** | **82.39** | **89.40** | **34.76** | **62.13** | **74.76** | **34.72** | **63.21** | **75.15** |
| Xpool | 53.13 | 82.84 | 88.81 | 51.49 | 80.90 | **88.96** | 25.56 | 52.59 | 66.50 | 25.12 | 52.39 | 66.22 |
| ● Tent | 52.69 | 83.13 | 88.81 | 51.64 | 80.45 | 88.81 | 25.36 | 52.80 | 66.81 | 24.93 | 52.74 | 66.20 |
| ● READ | 53.13 | 82.69 | 88.81 | 51.49 | 80.75 | 88.96 | 21.15 | 46.43 | 60.12 | 22.62 | 48.34 | 61.93 |
| ● SAR | 52.54 | 83.13 | 88.81 | 51.64 | 80.45 | 88.96 | 25.40 | 52.96 | 66.85 | 25.14 | 52.74 | 66.36 |
| ● EATA | 52.69 | **83.43** | 88.81 | 51.94 | 80.60 | 88.81 | 24.93 | 50.99 | 64.53 | 25.97 | 52.72 | 66.56 |
| ● TCR | 52.84 | 82.69 | 89.25 | 51.19 | **81.04** | 88.81 | 25.87 | 53.08 | 67.07 | 24.93 | 52.86 | 66.34 |
| ● Ours | **55.37** | 81.49 | **90.15** | **55.67** | 80.75 | 88.81 | **31.10** | **58.41** | **70.96** | **30.53** | **59.16** | **72.26** |

Table 30: Comparisons on QGS *Cross-dataset Adaptation* between LSMDC and MSVD.

| Cross | LSMDC→MSVD | | | | | | MSVD→LSMDC | | | | | |
| Dataset | v2t | | | t2v | | | v2t | | | t2v | | |
| *Metrics* | **R@1** | **R@5** | **R@10** | **R@1** | **R@5** | **R@10** | **R@1** | **R@5** | **R@10** | **R@1** | **R@5** | **R@10** |
|---|---|---|---|---|---|---|---|---|---|---|---|---|
| CLIP4Clip | 48.36 | 79.70 | 87.61 | 48.21 | 76.12 | 86.87 | 14.91 | 28.43 | 36.84 | 15.12 | 30.03 | 38.94 |
| ● Tent | 48.96 | 79.85 | 87.61 | 47.61 | 76.12 | 86.57 | 15.12 | 28.83 | 37.24 | 15.42 | 29.93 | 38.84 |
| ● READ | 48.66 | 79.85 | 87.61 | 48.06 | 76.12 | 86.72 | 14.51 | 28.43 | 36.44 | 15.32 | 30.03 | 38.74 |
| ● SAR | 48.06 | 79.85 | 87.61 | 47.61 | 76.12 | 86.57 | 15.32 | 29.23 | 37.44 | 15.32 | 29.93 | 38.94 |
| ● EATA | 48.93 | 81.04 | 87.61 | 48.66 | 75.97 | 86.42 | 15.82 | 29.83 | 38.74 | 15.52 | 29.93 | 39.34 |
| ● TCR | 46.72 | 80.45 | 88.06 | 47.91 | 76.27 | 86.42 | 14.41 | 30.23 | 38.04 | 15.22 | 30.23 | 38.94 |
| ● Ours | **51.94** | **81.64** | **88.51** | **51.64** | **79.85** | **88.96** | **17.42** | 33.13 | 40.04 | **16.32** | 32.83 | **40.94** |
| Xpool | 46.12 | 78.51 | 86.57 | 47.76 | 77.61 | 87.91 | 16.82 | 33.33 | 41.24 | 16.82 | 33.33 | 41.24 |
| ● Tent | 46.87 | 79.10 | 86.57 | 48.21 | 77.76 | **88.06** | 17.02 | 33.33 | 41.24 | 17.02 | 33.33 | 41.24 |
| ● READ | 46.72 | 78.51 | 86.57 | 47.76 | 77.61 | 87.91 | 16.82 | 32.93 | 40.94 | 16.82 | 32.93 | 40.94 |
| ● SAR | 46.57 | 78.66 | 86.57 | 48.06 | **77.91** | **88.06** | 16.92 | 33.33 | 41.14 | 16.92 | 33.33 | 41.14 |
| ● EATA | 47.16 | 79.55 | 86.42 | 50.00 | **77.91** | 87.91 | 16.92 | **33.53** | 40.84 | 16.92 | **33.53** | 40.84 |
| ● TCR | 46.57 | 78.96 | 86.57 | 49.10 | 77.01 | 87.91 | 16.82 | **33.53** | 41.04 | 16.82 | **33.53** | 41.04 |
| ● Ours | **54.03** | **81.94** | **89.70** | 51.49 | **77.91** | 86.72 | **17.32** | **33.53** | **42.14** | **17.32** | **33.53** | **42.14** |

Table 31: Comparisons on QGS *Zero-shot Adaptation* using LSMDC and MSVD.

| QGS | LSMDC | | | | | | MSVD | | | | | |
| Zero-shot | *v2t* | | | *t2v* | | | *v2t* | | | *t2v* | | |
| *Metrics* | **R@1** | **R@5** | **R@10** | **R@1** | **R@5** | **R@10** | **R@1** | **R@5** | **R@10** | **R@1** | **R@5** | **R@10** |
| CLIP | 7.61 | 18.62 | 26.73 | 13.71 | 27.43 | 34.33 | 44.03 | 73.73 | 82.54 | 47.16 | 71.64 | 81.79 |
| • Tent | 7.21 | 19.52 | 26.43 | 13.61 | 27.23 | 33.93 | 44.33 | 74.18 | 83.88 | 47.16 | 71.49 | 82.09 |
| • READ | 8.31 | 19.22 | 26.73 | 13.71 | 27.33 | 33.93 | 43.58 | 73.28 | 82.24 | 47.01 | 71.79 | 81.34 |
| • SAR | 7.31 | 19.02 | 26.53 | 13.51 | 27.33 | 34.03 | 44.48 | 73.88 | 83.43 | 47.16 | 71.64 | 82.09 |
| • EATA | 4.40 | 14.81 | 22.72 | 13.51 | 26.03 | 35.14 | 44.78 | 74.93 | 84.48 | 46.57 | 72.84 | 83.13 |
| • TCR | 10.61 | 24.92 | 31.53 | 13.71 | 27.53 | 35.14 | 47.31 | 75.67 | 85.97 | 47.46 | 72.99 | 83.43 |
| • Ours | **15.22** | **30.13** | **37.64** | **13.91** | **29.33** | **36.74** | **53.28** | **79.10** | **86.27** | **52.09** | **77.16** | **85.82** |

Table 32: Comparisons *v2t* results on the MSRVTT-1kA with severity degree 1.

| Query | Low-Level | | | | | | Mid-Level | | | High-Level | | | |
| Shift | Gauss. | Impul. | Fog | Snow | Elastic. | H264. | Motion | Defocus | Occlu. | Style | Event | Tempo. | Avg. |
| CLIP4Clip | 38.3 | 26.8 | 38.3 | 29.6 | 39.0 | 42.0 | 39.2 | 32.3 | 34.9 | 28.6 | 37.6 | 39.0 | 35.5 |
| • Tent | 38.8 | 27.1 | 39.4 | 30.3 | 39.6 | 42.0 | 39.7 | 33.9 | 35.1 | 30.4 | 37.9 | 38.8 | 36.1 |
| • READ | 37.3 | 26.6 | 37.8 | 29.4 | 38.7 | 41.7 | 39.0 | 31.3 | 34.1 | 26.5 | 36.9 | 38.8 | 34.8 |
| • SAR | 38.5 | 27.9 | 39.5 | 30.6 | 39.8 | 42.2 | 39.7 | 33.9 | 35.0 | 30.5 | 37.8 | 38.8 | 36.2 |
| • EATA | 39.0 | 31.1 | 40.2 | 33.2 | 39.5 | 42.7 | 39.4 | 34.9 | 36.2 | 31.5 | 38.8 | **39.3** | 37.2 |
| • TCR | 38.8 | 33.6 | 38.7 | 34.2 | 38.4 | 40.9 | 37.6 | 33.1 | 36.6 | 31.3 | 36.6 | 38.6 | 36.5 |
| • Ours | **41.1** | **38.7** | **41.8** | **38.9** | **40.7** | **42.5** | **41.9** | **37.4** | **39.3** | **35.6** | **40.6** | 38.8 | **39.8** |
| Xpool | 40.5 | 30.6 | 41.6 | 32.1 | 42.9 | 45.0 | 41.7 | 36.5 | 39.1 | 31.3 | 40.4 | 39.4 | 38.4 |
| • Tent | 41.1 | 32.0 | 41.7 | 33.8 | 44.2 | 45.8 | 41.9 | 38.3 | 40.0 | 33.4 | 40.4 | 40.3 | 39.4 |
| • READ | 40.7 | 30.0 | 41.0 | 31.8 | 43.0 | 44.7 | 41.7 | 35.5 | 38.9 | 29.8 | 39.8 | 39.1 | 38.0 |
| SAR | 41.1 | 32.0 | 41.7 | 33.2 | 44.1 | 45.5 | 42.2 | 37.7 | 40.3 | 33.2 | 40.4 | 40.2 | 39.3 |
| • EATA | 43.1 | 35.3 | 43.5 | 36.8 | 42.5 | 44.8 | 42.5 | 38.5 | 39.7 | 34.9 | 39.6 | 40.5 | 40.1 |
| • TCR | 41.7 | 33.8 | 42.5 | 35.6 | 42.8 | 44.6 | 42.5 | 37.8 | 40.2 | 34.4 | 40.1 | 39.5 | 39.6 |
| • Ours | **44.9** | **40.4** | **45.7** | **42.3** | **44.6** | **47.9** | **45.1** | **41.6** | **44.0** | **38.2** | **42.4** | **42.6** | **43.3** |

Table 33: Comparisons *v2t* results on the MSRVTT-1kA with severity degree 2.

| Query | Low-Level | | | | | | Mid-Level | | | High-Level | | | |
| Shift | Gauss. | Impul. | Fog | Snow | Elastic. | H264. | Motion | Defocus | Occlu. | Style | Event | Tempo. | Avg. |
| CLIP4Clip | 33.3 | 19.8 | 37.0 | 21.2 | 25.9 | 41.5 | 35.7 | 28.3 | 31.9 | 24.3 | 34.2 | 36.7 | 30.8 |
| • Tent | 35.2 | 19.2 | 38.1 | 21.8 | 25.9 | 41.8 | 36.4 | 29.8 | 32.4 | 26.2 | 34.1 | 36.9 | 31.5 |
| • READ | 31.8 | 21.1 | 36.4 | 20.5 | 25.5 | 41.3 | 35.8 | 27.8 | 31.5 | 22.8 | 33.9 | 36.8 | 30.4 |
| • SAR | 35.2 | 20.5 | 37.6 | 22.5 | 26.2 | 41.6 | 36.3 | 29.7 | 32.7 | 26.3 | 34.1 | 37.1 | 31.7 |
| • EATA | 36.8 | 26.3 | 38.8 | 24.1 | 27.3 | 43.3 | 37.2 | 31.1 | 34.7 | 29.3 | 36.1 | 36.6 | 33.5 |
| • TCR | 36.1 | 28.9 | 37.7 | 27.6 | 27.9 | 40.9 | 37.3 | 30.2 | 32.8 | 28.8 | 32.8 | **37.5** | 33.2 |
| • Ours | **39.5** | **35.9** | **41.3** | **34.6** | **31.8** | **43.0** | **40.4** | **34.5** | **37.6** | **33.1** | **39.2** | 37.2 | **37.3** |
| Xpool | 38.9 | 22.3 | 41.4 | 24.6 | 31.4 | 44.9 | 38.4 | 31.4 | 38.3 | 28.4 | 37.5 | 38.9 | 34.7 |
| • Tent | 40.0 | 22.1 | 41.9 | 26.4 | 32.6 | 46.0 | 38.9 | 33.4 | 38.4 | 29.5 | 37.4 | 39.1 | 35.5 |
| • READ | 37.4 | 23.2 | 40.7 | 23.8 | 30.6 | 44.7 | 37.6 | 29.7 | 37.8 | 27.2 | 37.2 | 38.6 | 34.0 |
| • SAR | 39.6 | 24.2 | 42.1 | 26.4 | 31.8 | 45.6 | 38.9 | 32.7 | 38.5 | 29.7 | 37.8 | 39.1 | 35.5 |
| • EATA | 40.4 | 31.9 | 44.1 | 30.5 | 32.9 | 45.8 | 39.5 | 33.8 | 38.1 | 32.0 | 37.7 | 38.3 | 37.1 |
| • TCR | 40.6 | 32.1 | 43.1 | 30.2 | 32.3 | 44.8 | 40.0 | 32.6 | 38.8 | 30.6 | 37.9 | 38.4 | 36.8 |
| • Ours | **44.2** | **36.7** | **44.3** | **36.4** | **35.3** | **47.5** | **42.1** | **38.7** | **42.1** | **36.0** | **40.5** | **40.0** | **40.3** |

Table 34: Comparisons *v2t* results on the MSRVTT-1kA with severity degree 3.

| Query Shift | Low-Level | | | | | | Mid-Level | | | High-Level | | | |
| --- | --- | --- | --- | --- | --- | --- | --- | --- | --- | --- | --- | --- | --- |
| | Gauss. | Impul. | Fog | Snow | Elastic. | H264. | Motion | Defocus | Occlu. | Style | Event | Tempo. | Avg. |
| CLIP4Clip | 27.1 | 14.8 | 35.2 | 20.4 | 36.5 | 40.7 | 34.0 | 18.9 | 29.2 | 18.0 | 30.1 | 36.4 | 28.4 |
| • Tent | 29.5 | 10.8 | 36.4 | 21.1 | 36.5 | 41.0 | 35.4 | 20.1 | 30.3 | 17.8 | 30.6 | 36.7 | 28.9 |
| • READ | 25.8 | 16.8 | 34.3 | 20.3 | 35.5 | 40.5 | 33.2 | 17.3 | 29.5 | 17.7 | 30.4 | 36.0 | 28.1 |
| • SAR | 29.8 | 14.3 | 36.6 | 22.0 | 36.5 | 41.0 | 35.5 | 20.6 | 30.3 | 19.0 | 30.6 | 36.7 | 29.4 |
| • EATA | 34.5 | 22.7 | 37.4 | 23.9 | 36.9 | 41.9 | 34.8 | 23.6 | 31.7 | 21.5 | 31.4 | 36.1 | 31.4 |
| • TCR | 32.7 | 26.6 | 37.1 | 28.2 | 37.7 | 39.8 | 36.0 | 22.0 | 31.6 | 21.6 | 29.9 | 34.9 | 31.5 |
| • Ours | **36.0** | **34.0** | **40.3** | **33.5** | **40.3** | **42.6** | **37.4** | **28.8** | **36.8** | **27.0** | **35.0** | **36.9** | **35.7** |
| Xpool | 32.0 | 17.7 | 38.6 | 24.9 | 40.5 | 43.6 | 34.1 | 21.7 | 35.6 | 19.8 | 35.3 | 37.0 | 31.7 |
| • Tent | 34.3 | 15.3 | 39.5 | 26.0 | 41.0 | 44.1 | 36.0 | 23.7 | 35.6 | 21.4 | 34.9 | 36.9 | 32.4 |
| • READ | 29.8 | 18.8 | 37.6 | 23.8 | 40.6 | 43.8 | 32.8 | 18.9 | 35.0 | 19.4 | 35.5 | 37.0 | 31.1 |
| • SAR | 34.3 | 18.1 | 39.6 | 25.8 | 40.9 | 43.8 | 35.7 | 24.0 | 35.5 | 22.4 | 35.0 | 37.2 | 32.7 |
| • EATA | 37.7 | 27.2 | 42.0 | 30.2 | 41.7 | 43.4 | 37.2 | 26.5 | 35.8 | 26.1 | 36.0 | 36.7 | 35.0 |
| • TCR | 35.2 | 29.3 | 41.0 | 30.5 | 42.6 | 43.9 | 36.1 | 23.7 | 36.2 | 25.7 | 35.5 | 36.5 | 34.7 |
| • Ours | **41.0** | **35.8** | **44.2** | **36.2** | **44.7** | **46.0** | **40.9** | **30.6** | **39.5** | **30.3** | **39.1** | **38.1** | **38.9** |

Table 35: Comparisons *v2t* results on the MSRVTT-1kA with severity degree 4.

| Query Shift | Low-Level | | | | | | Mid-Level | | | High-Level | | | |
| --- | --- | --- | --- | --- | --- | --- | --- | --- | --- | --- | --- | --- | --- |
| | Gauss. | Impul. | Fog | Snow | Elastic. | H264. | Motion | Defocus | Occlu. | Style | Event | Tempo. | Avg. |
| CLIP4Clip | 17.8 | 8.4 | 32.0 | 16.4 | 28.9 | 37.5 | 29.3 | 9.9 | 25.8 | 10.9 | 27.0 | 34.2 | 23.2 |
| • Tent | 19.0 | 4.6 | 33.1 | 16.6 | 30.0 | 37.8 | 30.3 | 9.6 | 26.2 | 9.6 | 26.5 | 34.0 | 23.1 |
| • READ | 18.7 | 10.7 | 31.2 | 16.1 | 27.3 | 37.3 | 28.0 | 9.7 | 24.8 | 12.0 | 27.2 | 33.9 | 23.1 |
| • SAR | 21.1 | 5.6 | 32.6 | 17.5 | 29.7 | 37.8 | 30.6 | 10.5 | 26.9 | 10.6 | 26.7 | 33.7 | 23.6 |
| • EATA | 25.1 | 1.9 | 34.4 | 20.7 | 32.1 | 38.6 | 31.5 | 13.0 | 28.6 | 13.3 | 26.8 | 33.4 | 25.0 |
| • TCR | 26.6 | 17.3 | 35.5 | 22.7 | 33.6 | 37.6 | 31.4 | 15.4 | 28.0 | 15.3 | 26.7 | 34.1 | 27.0 |
| • Ours | **31.8** | **29.1** | **39.6** | **29.2** | **36.9** | **39.2** | **35.2** | **21.9** | **34.4** | **21.1** | **31.6** | **35.9** | **32.2** |
| Xpool | 21.6 | 11.1 | 35.6 | 19.8 | 34.4 | 41.0 | 29.0 | 12.4 | 32.9 | 12.3 | 33.7 | 35.3 | 26.6 |
| • Tent | 24.6 | 7.0 | 37.4 | 21.2 | 35.7 | 41.1 | 31.2 | 14.6 | 33.7 | 12.0 | 33.8 | 35.1 | 27.3 |
| • READ | 20.8 | 13.2 | 34.6 | 19.2 | 33.5 | 40.6 | 27.7 | 11.4 | 32.5 | 13.3 | 33.5 | 35.2 | 26.3 |
| • SAR | 25.0 | 8.2 | 37.6 | 21.4 | 35.4 | 41.0 | 31.4 | 14.7 | 33.5 | 12.6 | 34.2 | 35.2 | 27.5 |
| • EATA | 29.1 | 2.1 | 39.2 | 23.7 | 37.4 | 41.4 | 33.3 | 16.3 | 33.6 | 16.9 | 34.1 | 35.7 | 28.6 |
| • TCR | 27.3 | 21.4 | 39.2 | 24.7 | 38.0 | 40.6 | 33.1 | 15.1 | 34.6 | 16.3 | 34.0 | 35.7 | 30.0 |
| • Ours | **35.1** | **30.1** | **43.7** | **32.1** | **41.1** | **42.5** | **37.4** | **23.0** | **37.4** | **21.2** | **37.1** | **36.8** | **34.8** |

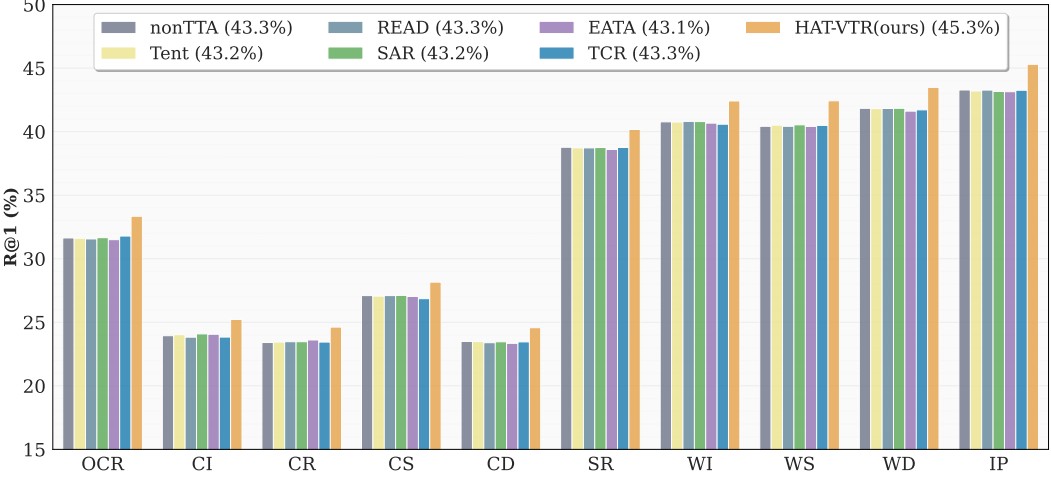

Figure 12: Comparison results of *t2v* on MSRVTT-1kA with text perturbations at mean severity.

Table 36: Multi-Level Video Perturbation (MLVP) Benchmark Summary

| Category | Perturbation | Description | Severities |
|---|---|---|---|
| Low-level | Gaussian Noise | Thermal sensor noise from low-light conditions or high ISO settings. | 5 |
| | Impulse Noise | Salt-and-pepper noise from defective sensor pixels and transmission errors. | 5 |
| | Fog | Atmospheric fog with fractal patterns generated by diamond-square algorithm. | 5 |
| | Snow | Falling snow with motion blur effects and temporal scrolling patterns. | 5 |
| | Elastic Distortion | Non-linear spatial deformations from lens aberrations and camera shake. | 5 |
| | H.264 Compression | Video compression artifacts from bitrate constraints using FFmpeg encoding. | 5 |
| Mid-level | Motion Blur | Directional blur from object/camera motion with adaptive kernel sizing. | 5 |
| | Video Defocus | Circular defocus blur simulating depth-of-field and autofocus failures. | 5 |
| | Main Object Occlusion | Semantic-aware occlusion targeting main objects identified by Algo 3. | 5 |
| High-level | Style Transfer | Artistic style transfer using AdaIN Huang & Belongie (2017) with temporal consistency. | 5 |
| | Event Insertion | Contextual disruption by splicing semantically similar video segments. | 5 |
| | Temporal Scrambling | Narrative disruption through temporal trimming and chunk reordering. | 5 |
| Total | 12 | — | 60 |

# E    MLVP Benchmark Implementation Details

This section provides a comprehensive technical breakdown of our Multi-Level Video Perturbation (MLVP) benchmark, which extends the systematic image-text robustness evaluation paradigm from Qiu et al. (2024) to the video domain. As outlined in the main paper, our benchmark is designed to systematically probe the spatio-temporal vulnerabilities of VTR models. We provide two tables for reference:

Tab. 36 offers a high-level summary, organizing the 12 distinct perturbation types into our three-level hierarchy (Low-level, Mid-level, and High-level) and describing the real-world degradation each simulates. Tab. 37 presents a more granular view, detailing the specific parameter values that control the intensity for each of the five severity levels.

The core design of MLVP is to move beyond simple frame-wise image corruptions and introduce perturbations that are authentic to the video modality. This is governed by two key principles.

- **Temporal Consistency:** To simulate realistic and continuous phenomena, perturbations are applied cohesively across a video's duration. For instance, a single noise pattern, a fixed geometric transformation, or a consistent artistic style is applied to all frames of a sequence. This ensures that the challenge posed to the model is inherently spatio-temporal, rather than a series of independent image-level degradations.

- **Principled Severity Scaling:** Each perturbation is rendered at five distinct degrees of severity. The parameters are carefully chosen to create a smooth degradation gradient, allowing for a fine-grained analysis of model robustness. To ensure comparability and build upon established standards, for perturbations that have direct counterparts in the original image-based benchmark Qiu et al. (2024) (e.g., Gaussian Noise, Impulse Noise), we

adopt the same severity parameter settings. The following sections detail the motivation and implementation for each of the 12 perturbation types.

Table 37: Video Perturbation Parameter Specifications by Severity Degree

| Method | Parameters |
|---|---|
| Gaussian Noise | Apply temporally consistent Gaussian noise with standard deviations of 0.08, 0.12, 0.18, 0.26, 0.38 based on severity levels 1-5 |
| Impulse Noise | Apply salt-and-pepper noise with the same spatial mask across frames, affecting 0.03, 0.06, 0.09, 0.17, 0.27 proportion of pixels for severity levels 1-5 |
| Fog | Generate plasma fractal using diamond-square algorithm with fog intensity and wibble decay parameters: (1.5, 2), (2, 2), (2.5, 1.7), (2.5, 1.5), (3, 1.4) for severity levels 1-5 |
| Snow | Create falling snow with parameters (loc, scale, zoom, threshold, blur_r, blur_s, blend): (0.1, 0.3, 3, 0.5, 10, 4, 0.8), (0.2, 0.3, 2, 0.5, 12, 4, 0.7), (0.55, 0.3, 4, 0.9, 12, 8, 0.7), (0.55, 0.3, 4.5, 0.85, 12, 8, 0.65), (0.55, 0.3, 2.5, 0.85, 12, 12, 0.55) |
| Elastic Distortion | Apply elastic transformation with (alpha, sigma, affine_magnitude): (244×2, 244×0.7, 244×0.1), (244×2, 244×0.08, 244×0.2), (244×0.05, 244×0.01, 244×0.02), (244×0.07, 244×0.01, 244×0.02), (244×0.12, 244×0.01, 244×0.02) |
| H.264 Compression | Encode video using FFmpeg with target bitrates of 500k, 250k, 100k, 50k, 25k bps for severity levels 1-5 |
| Motion Blur | Apply adaptive motion blur with base kernel sizes for low/high motion regions: (5,9), (7,13), (9,17), (11,21), (13,25) for severity levels 1-5 |
| Video Defocus | Apply adaptive defocus blur with base radii scaling by factor of 2 for high-motion regions: 3, 4, 6, 8, 10 for severity levels 1-5 |
| Main Object Occlusion | Occlude main objects with area ratios relative to object size: 30%, 40%, 50%, 60%, 80% for severity levels 1-5 |
| Style Transfer | Apply AdaIN style transfer with alpha interpolation parameters: 0.2, 0.4, 0.6, 0.8, 1.0 for severity levels 1-5 |
| Event Insertion | Insert semantically similar video segments with insertion ratios: 30%, 40%, 50%, 60%, 70% for severity levels 1-5 |
| Temporal Scrambling | Trim video content with retention ratios of 60%, 70%, 80%, 90%, 95% followed by chunk scrambling with increasing complexity |

### E.1 LOW-LEVEL VIDEO PERTURBATIONS

Low-level perturbations modify pixel-level values to simulate common degradations from sensor noise, environmental conditions, and digital processing. These corruptions are designed to challenge a model's foundational visual processing while preserving the underlying temporal structure and semantic content of the video.

**Gaussian Noise.** To simulate the thermal noise inherent in digital camera sensors, a common artifact in low-light conditions or at high ISO settings, we introduce additive Gaussian noise. To ensure the degradation is temporally coherent, mirroring a real sensor's consistent noise profile, a single noise pattern is sampled from a zero-mean Gaussian distribution and applied identically to every frame in the sequence. The five degrees of severity are controlled by varying the standard deviation ($\sigma$) of this distribution.

**Impulse Noise.** This perturbation, commonly known as salt-and-pepper noise, models artifacts arising from defective sensor pixels or digital transmission errors, challenging a model's robustness to sparse, high-intensity corruptions. Our implementation maintains temporal consistency by applying a fixed spatial mask across all frames, where a severity-controlled proportion of pixels are randomly set to maximum (salt) or minimum (pepper) intensity.

**Fog.** To simulate the reduced visibility and contrast from atmospheric fog, which exhibits complex, fractal-like density patterns unlike uniform haze, our implementation leverages the diamond-

square algorithm. A single, temporally consistent plasma fractal is generated to serve as a fog density map, which is then additively blended with each frame. The intensity of this fog layer is scaled to adjust the severity.

**Snow.** Falling snow introduces dynamic, semi-random occlusions that can obscure objects and motion cues. We simulate this effect by generating an extended "snow curtain" that scrolls vertically across the video at a constant velocity, creating a consistent falling motion. To enhance realism, the particles within the curtain have motion blur applied, simulating the appearance of fast-moving snowflakes.

**Elastic Distortion.** This perturbation models the non-linear spatial warping that can result from lens aberrations, minor camera shake, or atmospheric phenomena like heat haze. To simulate a persistent distortion, a fixed displacement field is generated using Gaussian-smoothed random noise and applied to every frame in the video. The magnitude and smoothness of the deformation field are controlled by the severity degree, testing the model's invariance to geometric deformations.

---

**Algorithm 1** H.264 Video Compression Perturbation

---

**Require:** Video path $V$, severity level $s \in \{1, 2, 3, 4, 5\}$, output frame count $n$
**Ensure:** Compressed video tensor $T \in \mathbb{R}^{n \times C \times H \times W}$
1: bitrates $\leftarrow [500k, 250k, 100k, 50k, 25k]$
2: $b \leftarrow$ bitrates$[s - 1]$ ▷ Select bitrate by severity
3: $F, H, W,$ fps $\leftarrow$ LoadVideo$(V)$ ▷ Extract frames and metadata
4: temp_path $\leftarrow$ CreateTempFile("temp.mp4")
5: FfmpegEncode$(F,$ temp_path$, b, H, W,$ fps$)$ ▷ Compress with target bitrate
6: $F_{\text{compressed}} \leftarrow$ FfmpegDecode(temp_path) ▷ Decode compressed video
7: DeleteFile(temp_path) ▷ Cleanup temporary file
8: indices $\leftarrow$ UniformSample$(|F_{\text{compressed}}|, n)$ ▷ Sample frames uniformly
9: $F_{\text{sampled}} \leftarrow [F_{\text{compressed}}[i]$ for $i$ in indices$]$
10: **if** $|F_{\text{sampled}}| < n$ **then** ▷ Pad if necessary
11:     $F_{\text{sampled}} \leftarrow F_{\text{sampled}} +$ [repeat last frame]
12: **end if**
13: $T \leftarrow$ FramesToTensor$(F_{\text{sampled}})$ ▷ Convert to normalized tensor
14: **return** $T$

---

**H.264 Compression.** Video compression artifacts are nearly ubiquitous in real-world applications due to storage and bandwidth constraints. To ensure authenticity, we perform actual H.264 encoding and decoding using FFmpeg, as detailed in Algorithm 1. Each video is compressed to a target bitrate corresponding to one of five severity degrees, introducing realistic artifacts such as blocking, ringing, and loss of fine detail into the final decoded frames.

### E.2 Mid-Level Video Perturbations

Mid-level perturbations target more complex, object-centric attributes and motion dynamics. They simulate real-world degradations that are tied to the semantic content and movement within the scene, posing a greater challenge to a model's spatio-temporal reasoning.

**Motion Blur.** Motion blur is a common video artifact where fast-moving objects or camera motion cause non-uniform, directional blurring. To realistically replicate this, our approach is motion-aware and adaptive. We first compute motion vectors between frames (see Algorithm 2) and then apply stronger directional blur kernels to regions with high motion, while leaving static areas less affected. The blur direction is aligned with the local motion vectors, creating a spatially varying and authentic effect.

**Video Defocus.** This perturbation simulates the isotropic blurring from a shallow depth of field or autofocus failures, common in videography. Our implementation is adaptive, using the motion vectors same from motion blur to apply a circular disk blur (bokeh) primarily to moving regions of

---

**Algorithm 2** Motion Vector Extraction for Video Frames

---

**Require:** Video frames $F = \{f_1, f_2, \ldots, f_n\}$, frame indices $I = \{i_1, i_2, \ldots, i_k\}$
**Ensure:** Motion vectors $MV = \{mv_1, mv_2, \ldots, mv_k\}$
 1: Initialize $MV \leftarrow \emptyset$
 2: Initialize GRAYCACHE $\leftarrow \emptyset$          ▷ Cache for grayscale frames
 3: **if** $|I| = 0$ **then**
 4:      **return** $MV$
 5: **end if**
 6: **for** $j = 1$ **to** $|I|$ **do**
 7:      idx $\leftarrow I[j]$
 8:      **if** idx $= 0$ **then**          ▷ First frame has zero motion
 9:          $h, w \leftarrow$ height and width of $F[\text{idx}]$
10:          $mv_{\text{zero}} \leftarrow \text{zeros}(h, w, 2)$          ▷ Zero motion vector
11:          $MV.\text{append}(mv_{\text{zero}})$
12:          **continue**
13:      **end if**
14:      prev_idx $\leftarrow$ idx $- 1$
15:      curr_idx $\leftarrow$ idx
16:      **if** prev_idx $\notin$ GRAYCACHE **then**
17:          GRAYCACHE[prev_idx] $\leftarrow$ RGBTOGRAY($F[\text{prev\_idx}]$)
18:      **end if**
19:      **if** curr_idx $\notin$ GRAYCACHE **then**
20:          GRAYCACHE[curr_idx] $\leftarrow$ RGBTOGRAY($F[\text{curr\_idx}]$)
21:      **end if**
22:      $I_{\text{prev}} \leftarrow$ GRAYCACHE[prev_idx]
23:      $I_{\text{curr}} \leftarrow$ GRAYCACHE[curr_idx]
24:      $mv \leftarrow$ FARNEBACKOPTICALFLOW($I_{\text{prev}}, I_{\text{curr}}$)
25:      $MV.\text{append}(mv)$
26: **end for**
27: **return** $MV$

---

the frame. The radius of the blur kernel is larger for regions with more motion, simulating a frequent scenario where a camera's focus fails to track a moving subject.

**Main Object Occlusion.**  The occlusion of semantically critical objects poses a significant challenge to VTR systems. To create a more realistic and difficult test than simple random occlusion, this perturbation targets the main subject of the video. Our novel pipeline, detailed in Algorithm 3, first employs a vision-language model (Qwen2.5-VL-7B (Bai et al., 2025a)) to generate a caption and identify key semantic nouns. These nouns then serve as open-vocabulary queries for an object detector (OWLv2 (Minderer et al., 2023)) to locate and track the main object, which is subsequently occluded with a black rectangle whose area scales with severity.

---

**Algorithm 3** Main Object Identification for Video Occlusion

---

**Require:** Video path $V$, number of frames $n$
**Ensure:** Main object data {video_caption, ranked_objects_per_frame}
1: $F \leftarrow \text{SAMPLEFRAMES}(V, n)$       ▷ Uniformly sample $n$ frames
2: caption $\leftarrow \text{QWEN2.5VL}(V)$       ▷ Generate video caption using Qwen2.5-VL
3: keywords $\leftarrow \text{EXTRACTNOUNS}(\text{caption})$       ▷ Extract noun phrases with spaCy
4: all_objects $\leftarrow [\ ]$
5: **for** each frame $f_i \in F$ **do**
6:      detections $\leftarrow \text{OWLv2}(f_i, \text{keywords})$       ▷ Open-vocabulary detection
7:      **for** each detection $d \in$ detections **do**
8:          crop $\leftarrow \text{CROPIMAGE}(f_i, d.\text{box})$
9:          $d.\text{embedding} \leftarrow \text{VISUALEMBEDDING}(\text{crop})$
10:          $d.\text{frame\_index} \leftarrow i$
11:          all_objects.append($d$)
12:      **end for**
13: **end for**
14: tracks $\leftarrow \text{ASSOCIATEOBJECTS}(\text{all\_objects})$       ▷ Group by embedding similarity
15: **for** each track $t \in$ tracks **do**
16:      $t.\text{persistence} \leftarrow |t.\text{appearances}|/n$
17: **end for**
18: ranked_frames $\leftarrow \{\}$
19: **for** each object $o \in$ all_objects **do**
20:      score $\leftarrow 0.5 \cdot o.\text{persistence} + 0.3 \cdot o.\text{area\_ratio} + 0.2 \cdot o.\text{confidence}$
21:      ranked_frames[$o.\text{frame\_index}$].append($\{o.\text{label}, o.\text{box}, \text{score}\}$)
22: **end for**
23: **for** each frame index $i$ **do**
24:      $\text{SORTBYSCORE}(\text{ranked\_frames}[i])$       ▷ Descending order
25: **end for**
26: **return** {caption, ranked_frames}

---

### E.3 HIGH-LEVEL VIDEO PERTURBATIONS

High-level perturbations alter the core semantic and temporal structure of a video. They are designed to challenge a model's high-level understanding, including its grasp of style, context, and narrative causality.

**Style Transfer.**  A robust VTR system should recognize semantic content irrespective of artistic style. This perturbation tests such invariance by applying style transfer using Adaptive Instance Normalization (AdaIN) (Huang & Belongie, 2017). To ensure temporal consistency, the style from a single, randomly selected artistic image is transferred to all frames of a given video. The severity level controls the $\alpha$ parameter, which dictates the interpolation strength between the original content and the new style.

**Event Insertion.**  To challenge a model's contextual understanding, this perturbation simulates scenarios like ad insertions or video mashups where an unrelated clip disrupts the narrative. For a given video, we use its CLIP embedding to retrieve a semantically similar (but non-identical) video

from an external corpus, which is constructed from a diverse base of 3 thousand video clips drawn from the MSRVTT training set. A segment from this retrieved video is then spliced into the middle of the original sequence, with the duration of the inserted segment determined by the severity level.

**Temporal Scrambling.** The chronological order of events is often critical to a video's narrative. This perturbation simulates network issues like out-of-order packet delivery by disrupting the video's temporal sequence. We first trim the video, then divide the remaining clip into several equal-length chunks which are subsequently reordered. The scrambling complexity scales with severity, from adjacent swaps to a full random shuffle, directly attacking the model's reliance on temporal coherence and causal reasoning.

### E.4 FURTHER ANALYSIS OF MLVP BENCHMARK

To validate the effectiveness and design of our MLVP benchmark, we conducted further analysis on the performance of representative VTR models under its various challenges. As illustrated in Fig. 13, which plots the Recall@1 performance of CLIP4Clip and X-Pool against the five severity degrees, the dominant trend is a clear and consistent negative correlation between perturbation intensity and retrieval accuracy.

However, we also note some intriguing exceptions. Notably, for H.264 Compression at lower severities (1 and 2), performance slightly improves over the unperturbed baseline. This suggests that certain mild perturbations do not necessarily degrade performance and may hint at future directions for model enhancement. It is also worth noting that our proposed HAT-VTR framework maintains performance gains even in such scenarios, which further demonstrates its robustness (Tab. 32). Another interesting anomaly is observed for Snow and Elastic Distortion, where performance shows a slight uptick when moving from severity 2 to 3. Since the severity parameters for these perturbations were adopted from Qiu et al. (2024), this finding highlights a valuable direction for future work: developing a methodology to define a unified set of severity parameters that generalize robustly across different datasets and data modalities. Despite these minor anomalies, the overall systematic degradation confirms that our principled severity scaling creates a meaningful and measurable gradient for evaluating model robustness, providing a solid foundation for analyzing the failure points of VTR systems.

To provide a qualitative understanding of the challenges posed, we visualize the effects of each perturbation type. Figure 14 showcases the suite of low-level corruptions, demonstrating how perturbations like Gaussian noise introduce pixel-level artifacts, Fog reduces contrast, and H.264 Compression creates blocking effects, all while maintaining temporal consistency. Figure 15 illustrates the more complex mid- and high-level perturbations. These examples highlight the directional, motion-aware nature of Motion Blur, the targeted impairment of Main Object Occlusion, the contextual disruption caused by Event Insertion, and the narrative incoherence introduced by Temporal Scrambling, showcasing the diversity of spatio-temporal challenges in our benchmark.

Furthermore, we visualize the direct impact of our severity scaling principle in Figure 16. Using three representative perturbations—Gaussian noise, Motion Blur, and Style Transfer—the figure contrasts the visual outcome at Severity 1 with that at Severity 5. For instance, Gaussian noise evolves from a fine grain to a heavy, obscuring static. Similarly, Motion Blur intensifies from a subtle directional softening to a severe blur that renders the object almost unrecognizable, while the artistic rendering in Style Transfer becomes progressively more dominant. These examples visually confirm that our parameter adjustments for each severity degree translate into a clear and graduated increase in the intensity of the perturbation, ensuring a comprehensive test of model robustness.

## F LIMITATIONS AND FUTURE WORKS

While HAT-VTR demonstrates substantial improvements across diverse test scenarios, our comprehensive analysis reveals specific scenarios where the method's effectiveness is constrained, providing valuable insights for future research directions.

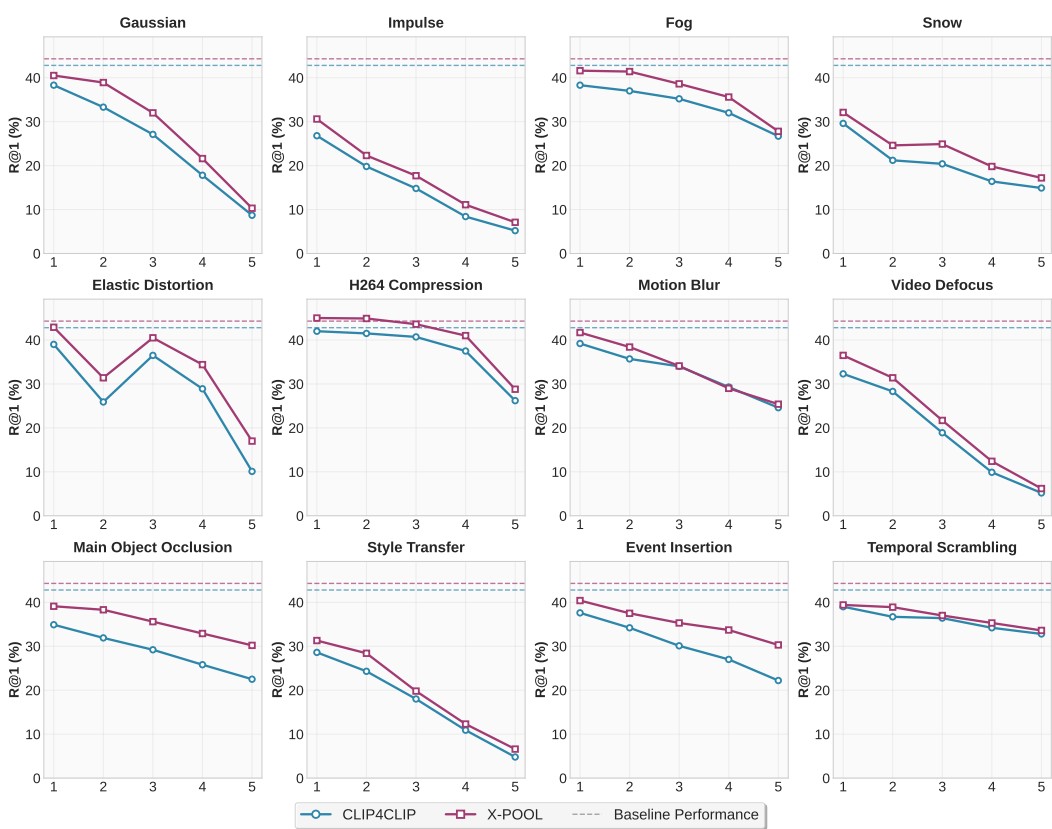

Figure 13: Performance of *v2t* models under different severity degrees in MLVP.

### F.1 PERFORMANCE ANALYSIS ON CHALLENGING SCENARIOS

Our method exhibits limited improvements in certain challenging scenarios, as evidenced by Table 1 2 and 3. Tab. 38 presents the performance breakdown on two representative cases where HAT-VTR shows modest gains: Temporal Scrambling and Backtranslation. For Temporal Scrambling, removing the training component ("w.o. Training") actually yields slightly better performance (33.4 vs 33.1 R@1), suggesting that the TCR-based adaptation may introduce negative optimization in this scenario. Similarly, for Backtranslation, the training component provides minimal benefit (39.8 vs 39.9 R@1), while the HSM module contributes the majority of the improvement.

To understand this phenomenon, Table 39 analyzes the hubness characteristics across different perturbation types. The skewness values reveal the underlying cause: while Gaussian noise induces severe hubness (skewness=9.09 without TTA vs 0.97 with HAT-VTR), Temporal Scrambling exhibits more moderate hubness amplification (skewness=2.19 vs 1.15). Similarly, in the *t2v* setting, OCR perturbation shows limited hubness issues (skewness=1.95 vs 0.39), while Backtranslation demonstrates even milder hubness patterns (skewness=1.36 vs 1.12). This analysis explains why HAT-VTR's hubness-focused approach yields substantial gains for severe hubness scenarios but offers limited improvements when the hubness phenomenon is less pronounced.

Table 38: Ablation study on challenging scenarios where HAT-VTR shows limited improvements. "w.o. Training" removes the TCR-based adaptation component, while "w.o. HSM" removes the Hubness Suppression Memory module.

|  | R@1 | R@5 |
|---|---|---|
| Tempo. (HAT-VTR) | 33.1 | 55.3 |
| • w.o. Training | 33.4 | 55.5 |
| • w.o. HSM | 32.2 | 53.4 |
| Backtrans. (HAT-VTR) | 39.8 | 67.1 |
| • w.o. Training | 39.9 | 67.2 |
| • w.o. HSM | 37.7 | 65.6 |

Table 39: Hubness analysis across challenging perturbation types for HAT-VTR. Skewness values indicate the degree of hubness amplification, with higher values representing more severe hubness issues.

| | w.o.TTA $v2t$ | Gauss. | Temp. | HATVTR $v2t$ | Gauss. | Temp. | w.o. TTA $t2v$ | OCR | Backtrans. | HATVTR $t2v$ | OCR | Backtrans. |
|---|---|---|---|---|---|---|---|---|---|---|---|---|
| Skewness | 1.52 | 9.09 | 2.19 | - | 0.97 | 1.15 | 1.25 | 1.95 | 1.36 | - | 0.39 | 1.12 |
| R@1 | 42.5 | 8.7 | 32.8 | - | 23.1 | 33.1 | 41.6 | 21.5 | 38.5 | - | 24.5 | 39.8 |

## F.2 METHOD LIMITATIONS

Based on this analysis, our approach faces three primary limitations. First, although we observe that perturbations amplify the hubness phenomenon, our work lacks deeper theoretical analysis of the underlying mechanisms governing how different corruption types induce varying degrees of hubness amplification. The heterogeneous hubness manifestation across perturbation types necessitates more nuanced understanding of the relationship between specific corruptions and retrieval failure modes.

Second, our training framework essentially extends TCR learning to the video domain, but extensive experiments reveal that TCR-based adaptation performs well on low-level perturbations with severe hubness issues but struggles with scenarios where hubness amplification is moderate. This fundamental limitation constrains HAT-VTR's performance ceiling, particularly evident in cases where direct HSM reranking without training achieves comparable or better results.

Third, while we explicitly apply HSM for reranking and nearest neighbor selection, integrating hubness suppression directly into the learning loss remains unexplored. This represents a significant opportunity for developing more principled approaches to hubness-aware adaptation that could potentially address the training limitations identified above.

## F.3 MLVP BENCHMARK LIMITATIONS

Our MLVP benchmark, while comprehensive, has inherent limitations that affect evaluation reliability. Following established image-text robustness paradigms, some video perturbations exhibit non-monotonic severity progression (e.g. Elastic Distortion), necessitating more careful calibration of severity parameters across different datasets. Additionally, certain perturbations like Main Object Occlusion and Style Transfer rely heavily on auxiliary model capabilities, potentially limiting the benchmark's ability to simulate authentic real-world video corruptions.

## F.4 FUTURE RESEARCH DIRECTIONS

These limitations suggest several promising research avenues. Future work should focus on developing theoretical frameworks that explain perturbation-specific hubness amplification patterns, potentially leading to adaptive strategies that adjust based on corruption characteristics. The development of training paradigms that effectively handle both severe and moderate hubness scenarios remains crucial, possibly requiring dynamic adaptation mechanisms that activate different components based on hubness severity.

Furthermore, investigating direct integration of hubness suppression into learning objectives could yield more principled adaptation methods. From a benchmarking perspective, developing robust severity calibration methods and exploring perturbation techniques that better simulate real-world corruptions would enhance evaluation reliability.

These limitations notwithstanding, our work establishes a solid foundation for hubness-aware video-text retrieval and provides clear directions for advancing toward more robust cross-modal systems.

## G THE USE OF LARGE LANGUAGE MODELS

Large language models were used as a writing assistant to help polish this manuscript. The usage was limited to improving language clarity, rephrasing sentences, and correcting grammar. LLMs were not used for generating core ideas, experimental results, or analyses. The authors take full responsibility for all content presented in this paper.

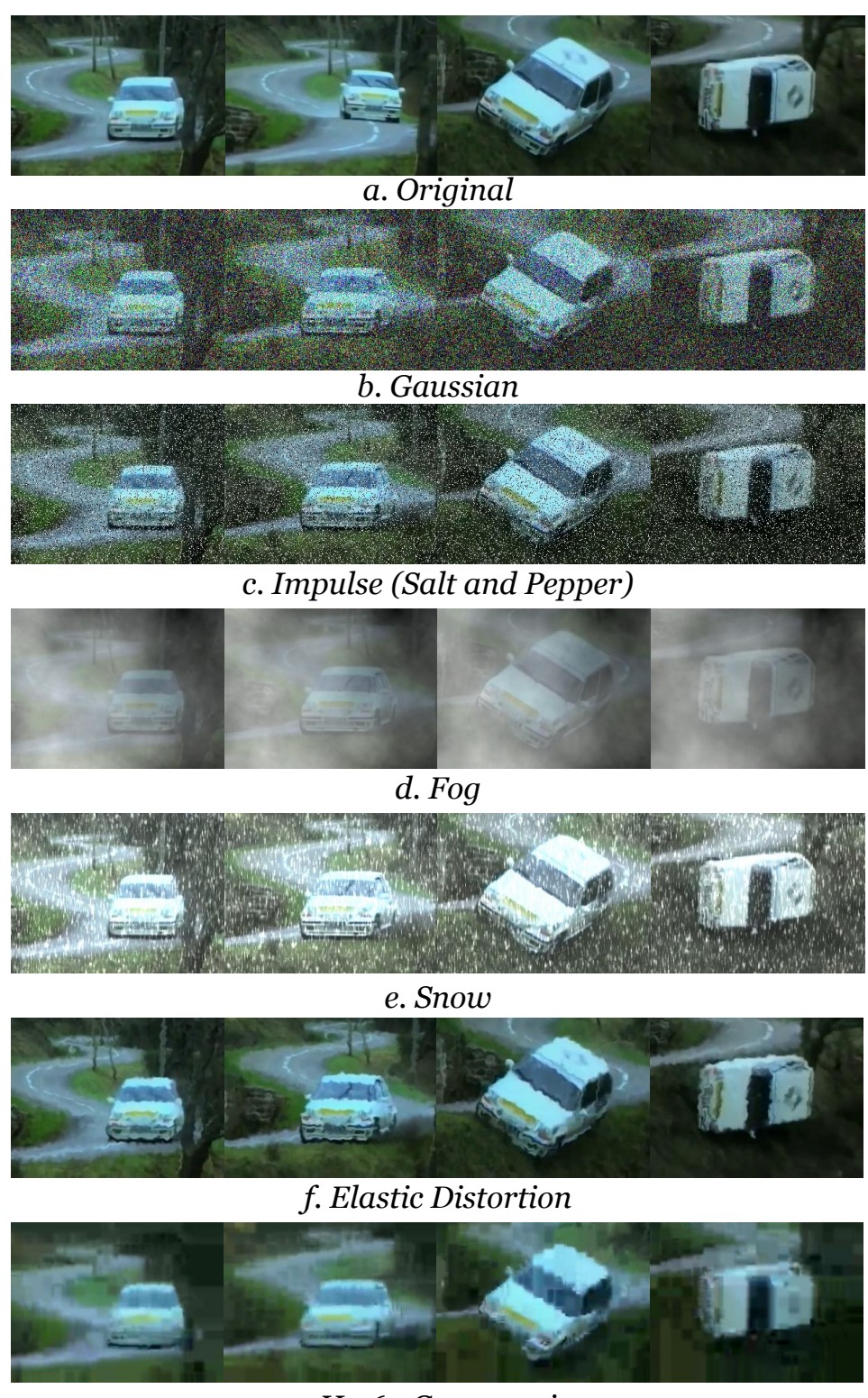

a. Original

b. Gaussian

c. Impulse (Salt and Pepper)

d. Fog

e. Snow

f. Elastic Distortion

g. H.264 Compression

Figure 14: Visualization Examples of Different Low-level Video Perturbations

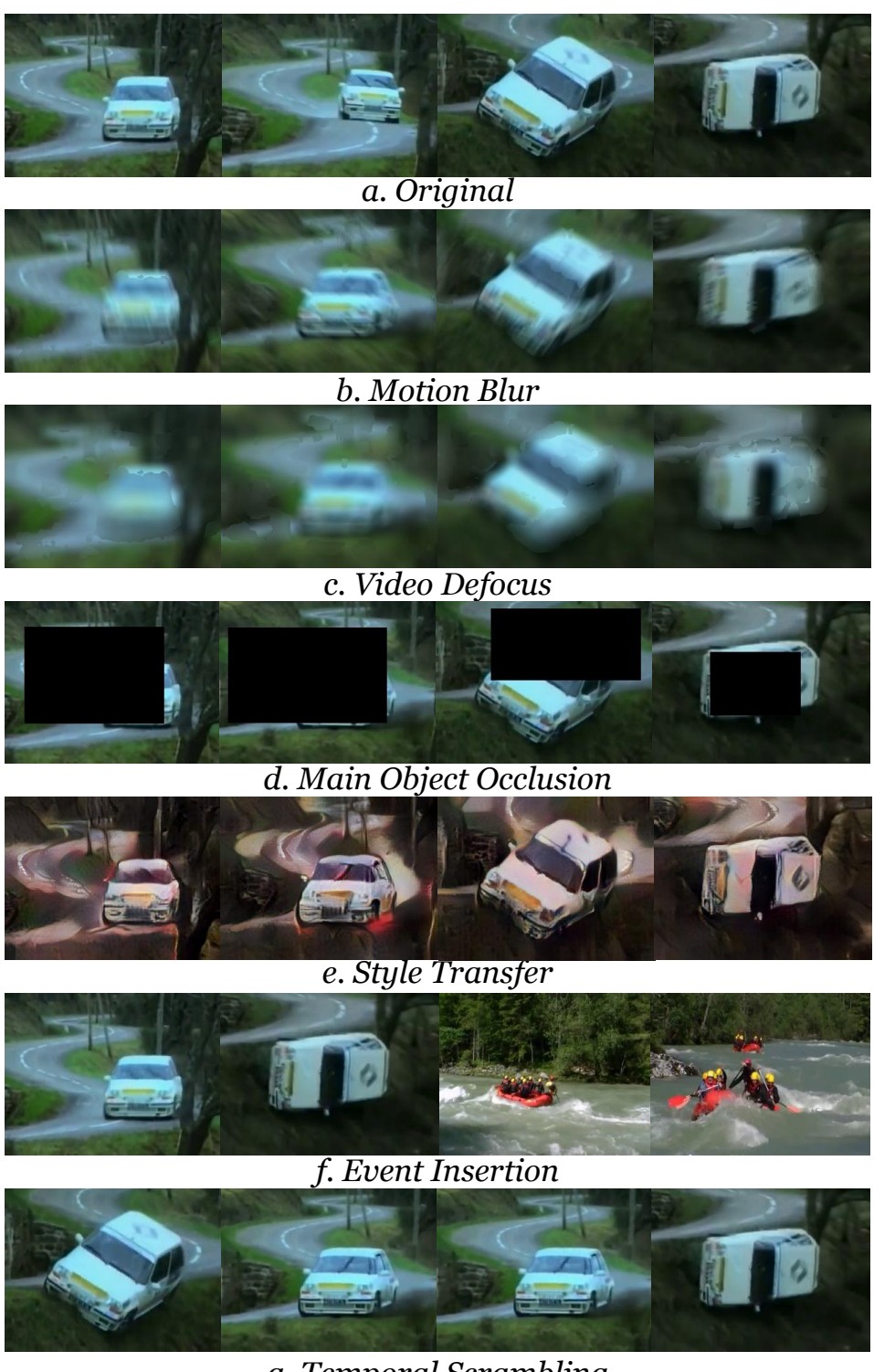

Figure 15: Visualization Examples of Different Mid- and High- level Video Perturbations

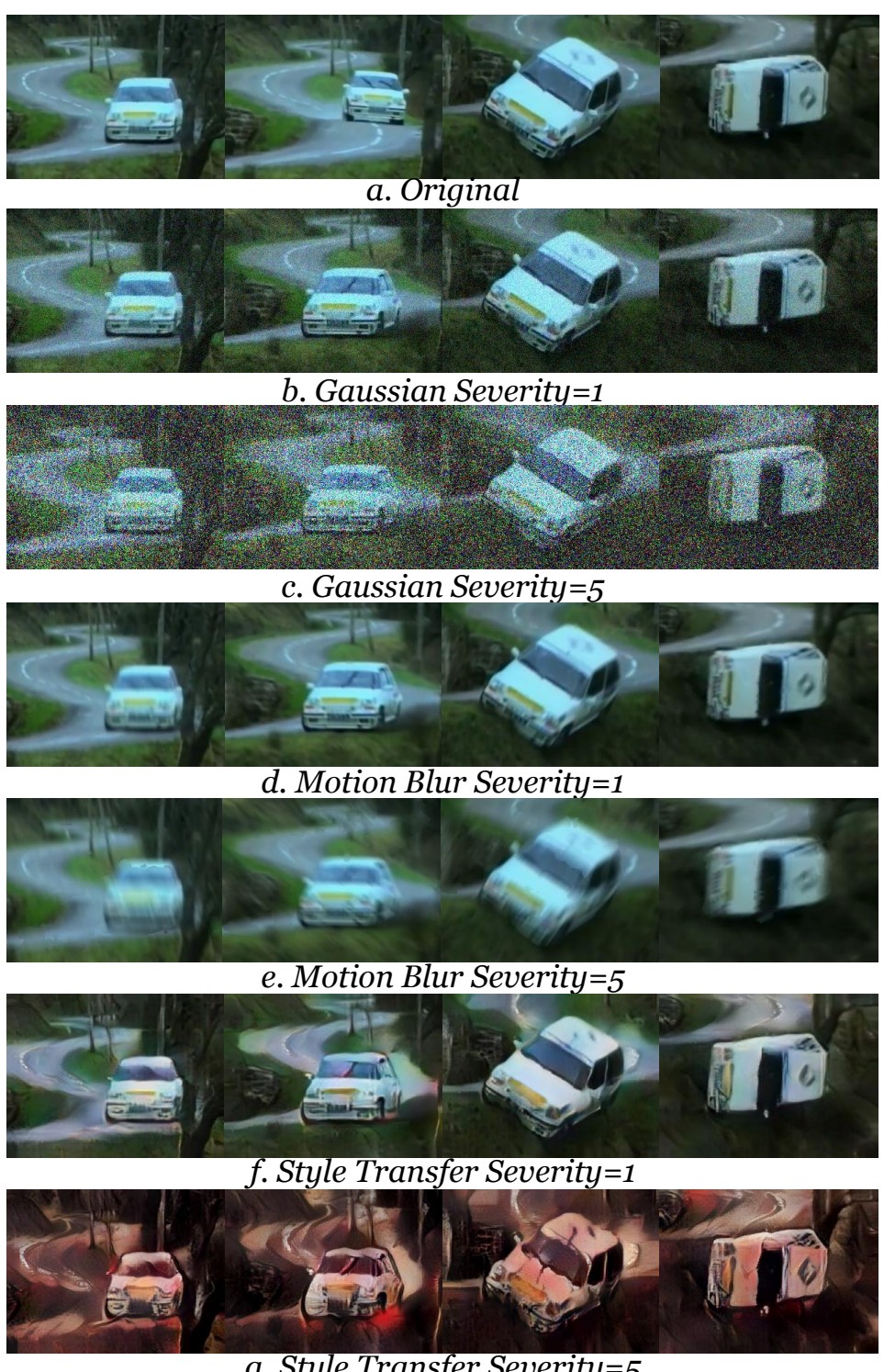

Figure 16: Visualization Examples of Severity Degree Changes in Multi-level Video Perturbations

