# OpenReview forum: "Robust Test-time Video-Text Retrieval: Benchmarking and Adapting for Query Shifts"
_ICLR.cc/2026/Conference — ICLR 2026 Poster_

### Official Review · Reviewer_tpJo · 2025-10-30

**Soundness:** 3
**Presentation:** 2
**Contribution:** 2
**Rating:** 4
**Confidence:** 4

**Summary:**

The authors point out that existing video-text retrieval models exhibit performance degradation when the distribution of query data behaves differently from the training domain. While some solutions have been introduced, they are mainly focused on the image applications, and not applicable for videos due to the complex spatio-temporal dynamics. To systematically evaluate the vulnerability, the authors have introduced their own benchmark revealing the vulnerability arises from a phenomenon called hubness phenomenon. This phenomenon appears when few gallery items become dominant behaving as a hub. To tackle this, the authors introduce a test-time adaptation framework which consists of two key components. First, a hubness suppression memory refines the similarity scores. Second, the multi-granular loss is designed to keep the temporal feature consistent. HAT-VTR is evaluated on four main datasets, which exhibit strong performances.

**Strengths:**

- Problem is well driven with clear motivations. The authors have used their own benchmark to uncover where the vulnerability comes from. The analysis becomes a clear evidence of hubness phenomenon.
- The proposed model has been directly derived from the analysis with the benchmark. The framework is straightforward and is based upon two main purposes.
- Extensive experimental results supports the efficacy of HAT-VTR.

**Weaknesses:**

- Some claims are controversial. For instance, the candidate component barely contributes to final performance (see Table 6). Adding candidate (row 2 and row 4) affects minimal compared to Rerank. This is unexpected as ‘candidate selection’ associates extensive computations.
- Too many temperature parameters: $\tau$ $\alpha$, $\beta$, t in Eq 7 and 8 requires extensive tuning/searching. Still, there are more hyperparameters including those specified as hyperparameters and those not specified (e.g., fraction r) . It is unclear how the proposed model is sensitive to these combinations unless reported. Some of the methods are based on heuristics.
- HAT-VTR requires some computational cost, which only has been reported in Appendix.

**Questions:**

Q1. What is the rationale behind using the first B rows in line 256?

Q2. Any details on posterior reranking? Isn’t the reranking based on some heuristic? How do you *demote* hubs?

---

> ### Author Response · Authors · 2025-11-20
>
> > **[W1]** Some claims are controversial. For instance, the candidate component barely contributes to final performance (see Table 6). Adding candidate (row 2 and row 4) affects minimal compared to Rerank. This is unexpected as ‘candidate selection’ associates extensive computations.
>
> Thank you for your comments. We believe the concern mainly arises from a misunderstanding of the role of the “candidate” component in our method. In the revised version, we therefore rename it to Hubness-Aware Target Selection to better reflect its function and clarify its place within HSM.
>
> Target Selection (previous Candidate Selection) is only used inside the TCR-style adaptation loop to build the Reliable Memory (RM), i.e., to decide which pseudo-positives are used as training targets. **Its goal is to stabilize test-time learning, rather than to get the final retrieval list.** Concretely, we replace TCR’s reliance on raw similarity scores $S$ with our refined scores $\hat{S}_t$ when selecting RM entries. This reduces error accumulation in pseudo-labels, which is reflected by the substantially higher RM accuracy in **Fig. 10.** In contrast, the Posterior Similarity Reranking step operates on the adapted similarity scores at the end of the pipeline and is therefore the main driver of Recall gains, as correctly observed in **Table 6**.
>
> Target Selection does not introduce additional forward passes or large-scale search beyond TCR. It reuses the same similarity matrix already computed for retrieval and applies a light-weight hubness-aware reweighting to select RM entries. The resulting overhead is negligible compared to backbone inference and comparable to TCR’s original pseudo-label selection.
>
> Under this clarified interpretation, **the ablation in Table 6 is not contradictory**: Target Selection primarily improves the reliability of adaptation (RM accuracy and robustness under long test streams), while Posterior Reranking is responsible for most of the direct Recall improvement. We will make this clearer in the paper by renaming the module to Hubness-Aware Target Selection and explicitly distinguishing these two roles of HSM.

---

> ### Author Response · Authors · 2025-11-20
>
> > **[W2]** Too many temperature parameters… requires extensive tuning/searching. Still, there are more hyperparameters …(e.g., fraction r) . It is unclear how the proposed model is sensitive to these combinations unless reported. Some of the methods are based on heuristics.
>
>
> Thank you for your comment. To make the discussion more transparent, we separate the parameters that are newly introduced by HAT-VTR from standard TTA / optimization choices that already appear in prior work, and summarize them here.
>
> - **Table A: Novel Hyperparameters introduced by HAT-VTR**
>
> | Hyperparameters | Description | Ablation Location |
> | --- | --- | --- |
> | $K$ | Size of Hubness Suppression Memory (HSM) queue | Fig. 7 (b) |
> | $\alpha, \beta$ | Temperature scaling factors for gallery/query in HSM | Fig. 5 (a), Tab. 12 |
> | $m$ | Balancing weight for hubness suppression (Eq. 6) | Fig. 5 (b) |
>
> All other important hyperparameters  (including $\tau$ and $t$ ) are standard TTA / contrastive-learning choices that already exist in TCR or related methods. For completeness, we summarize them as well:
>
> - **Table B: Original TCR / Optimization Hyperparameters**
>
> | Hyperparameters | Description | Ablation Location |
> | --- | --- | --- |
> | $\tau$ | Softmax temperature for contrastive loss (Eq. 3) | Fig. 7 (a) |
> | $t$ | Temperature for uniformity loss (Eq. 7, 8) | Fig. 6 (a) |
> | Batch Size | Batch size for online adaptation | Fig. 6 (b) |
> | Learning Rate | Step size for optimizer | Fig. 6 (c) |
> | Reliable Memory Size | Capacity of the reliable sample bank | Fig. 7 (c) |
>
> For the new HAT-VTR parameters ( $K, \alpha, \beta, m$ ), we conduct systematic sensitivity studies **on both MSRVTT and ActivityNet**. The results in **Fig. 5, Fig. 7, and Tab. 12** show that performance varies smoothly over a wide range of values and does not exhibit sharp degradation, indicating that HAT-VTR is robust to these choices. For each setting, we fix one configuration on a validation split and then reuse the same setting across all perturbation types and strengths, i.e., we do not perform per-perturbation tuning. Since our test-time adaptation only updates LayerNorm parameters, the optimization dynamics are relatively stable, which further reduces sensitivity to hyperparameter choices.
>
> Regarding the fraction r, it is only an alternative way to parameterize the HSM size via
> $K = \max(1, \lfloor r \cdot N_G \rfloor)$.
> In ablation study,  we parameterize and tune the memory by $K$ directly, and Fig. 7(b) studies its effect. To avoid confusion and the impression of an additional hyperparameter, we removed $r$ from the text and only keep $K$.
>
> Finally, on the comment that “some of the methods are based on heuristics”: the design of HAT-VTR follows well-established principles from the hubness and test-time adaptation literature. The bilateral reweighting and temperature scaling in HSM are motivated by DSL. The ablations mentioned above show that different reasonable settings of these factors lead to similar behavior, suggesting that the method is not overly dependent on fragile heuristic choices.
>
> > **[W3]** HAT-VTR requires some computational cost, which only has been reported in Appendix.
>
> Thank you for your suggestions. In the original submission,  efficiency results were placed in the appendix due to space constraints. In the revised paper, we move the key analysis into the main paper (**Tables 8 and 9**) and add additional details in **Appendix C.4**. The updated results show that HAT-VTR has runtime and memory consumption comparable to existing TTA baselines, and that the main overhead comes from the standard backward pass shared by all TTA methods, while the HSM module itself is lightweight. This makes the modest and well-quantified computational cost of HAT-VTR explicit in the main text.

---

> ### Author Response · Authors · 2025-11-20
>
> > **[Q1]** What is the rationale behind using the first B rows in line 256?
>
> In our notation,  $B$ is the batch size of the current queries. At test time, **HSM keeps a FIFO queue** $\mathcal{M}$ that stores the similarity matrices of the last  $K$ query batches (Eq. (5)). When forming the aggregated similarity matrix, we always put the current batch $S_t$ on top, followed by the $K-1$ previous batches from the queue. This aggregated matrix is then used to compute hubness statistics and the two weight matrices in Eq. (6).
>
> After this bilateral reweighting, **we only need the refined scores for the current queries, not for all past batches.** Because of the FIFO concatenation order, the first B rows of the refined matrix correspond exactly to the current batch. Therefore, we simply take these first B rows as the output $\hat{S}_t$ for step t, while past rows only serve to provide historical context when estimating hubness. (See Fig. 4)  This design lets HSM exploit recent query history via a bounded memory queue, without storing or recomputing scores for the entire test set.
>
> > **[Q2]** Any details on posterior reranking? Isn’t the reranking based on some heuristic? How do you *demote* hubs?
>
> Posterior reranking in HAT-VTR **is exactly implemented by the HSM update in Eq. (6)**. After aggregating the recent similarity history via Eq. (5), we compute two weight matrices: $W_{\text{gallery}}$ (column-wise softmax) measures how “popular’’ each gallery item is across recent queries, and $W_{\text{query}}$ (row-wise softmax) measures how concentrated each query is on a few gallery items. Eq. (6) combines these two views to obtain the refined similarity matrix $\hat{S}$, and the final ranking is obtained by sorting each row of  $\hat{S}_t$ (the part corresponding to the current batch).
>
> Our posterior reranking follows the DSL-style design [1], which has **been widely adopted in video–text retrieval** (e.g., DSL[1], UDA [2], DADA [3], CLIP-ViP [4], SeLeCT [5]). DSL-style methods are built on a mutual-consistency principle: a true video–text match should be strongly supported from both the query side and the gallery side of the interaction graph. HSM instantiates this principle in a density-aware way through Eq. (6), **rather than relying on an ad-hoc or arbitary heuristic scoring rule**. Moreover, our experiments show that this mechanism is consistently effective: HAT-VTR **achieves strong improvements and hubness reduction across five datasets and multiple perturbation types.**
>
> In terms of hub demotion,  $W_{\text{gallery}}$ explicitly captures hubness: gallery items that are frequently similar to many queries receive higher “popularity’’ weights. When combined with $W_{\text{query}}$ in Eq. (6), such globally popular but low-consensus items are down-weighted, while pairs supported by both query-specific concentration and gallery-level evidence are preserved. **The effect is visible in Fig. 9, where off-diagonal noise is strongly suppressed, and quantitatively confirmed in Table 13**, which shows consistent reductions across multiple hubness metrics under different perturbations.
>
> ### Reference
>
> [1] Cheng, Xing, et al. "Improving video-text retrieval by multi-stream corpus alignment and dual softmax loss." arXiv preprint arXiv:2109.04290 (2021).
>
> [2] Hao, Xiaoshuai, and Wanqian Zhang. "Uncertainty-aware alignment network for cross-domain video-text retrieval." *Advances in Neural Information Processing Systems* 36 (2023): 38284-38296.
>
> [3] Hao, Xiaoshuai, et al. "Dual alignment unsupervised domain adaptation for video-text retrieval." *Proceedings of the IEEE/CVF conference on computer vision and pattern recognition*. 2023.
>
> [4] Xue, Hongwei, et al. "Clip-vip: Adapting pre-trained image-text model to video-language representation alignment." arXiv preprint arXiv:2209.06430 (2022).
>
> [5] Lu, Yu, et al. "Exploiting unlabeled videos for video-text retrieval via pseudo-supervised learning." IEEE Transactions on Image Processing (2024).

---

> ### Author Response · Authors · 2025-11-26
> **Looking forward to your reply**
>
> Dear Reviewer tpJo,
>
> Thank you again for your constructive review. We have carefully addressed all your questions and concerns in our previous response and the revised manuscript.
>
> We would greatly appreciate it if you could spare a moment to check our reply. We hope that our clarifications have satisfactorily resolved your concerns, and we look forward to your feedback.
>
> Kind Regards,
>
> Authors of Paper 7458

---

### Official Review · Reviewer_drGP · 2025-10-31

**Soundness:** 3
**Presentation:** 3
**Contribution:** 3
**Rating:** 4
**Confidence:** 3

**Summary:**

The paper studies robust video–text retrieval (VTR) under query shift and contributes (i) MLVP, a benchmark with 12 video perturbation types × 5 severities spanning low/mid/high-level spatio-temporal corruptions, and (ii) HAT-VTR, a test-time adaptation (TTA) baseline that counters hubness via a Hubness Suppression Memory (HSM) and multi-granular losses for temporal consistency (with a Reliable Memory for stability). Experiments report consistent gains across query-shift settings, including scenarios where both queries and gallery drift.

**Strengths:**

The paper offers a clear diagnosis linking robustness failures under perturbations to amplified hubness (illustrated via k-occurrence distributions), introduces a well-scoped MLVP benchmark spanning 12 perturbation types × 5 severities that moves beyond image-only corruptions to video dynamics, proposes a simple plug-in TTA—HAT-VTR with Hubness Suppression Memory and multi-granular (global/frame) losses—that integrates cleanly with dual-encoder VTR, and demonstrates broad, consistent improvements over prior TTA methods under both query-shift and query+gallery-shift scenarios.

**Weaknesses:**

1.	End-to-end cost of the interactive TTA loop (HSM updates + multi-granular adaptation) is not characterized; please report per-round latency, peak memory, and wall-clock on a commodity GPU and CPU, and discuss amortized cost over sequence length.
2.	Comparisons to classical retrieval debiasing / hubness-reduction baselines are missing; adding such baselines would better isolate HSM’s contribution.
3.	The effect of online adaptation on in-distribution retrieval is unclear; include a no-shift control row to quantify any degradation and implement a switch to disable TTA when drift is not detected.
4.	Hyperparameter sensitivity is under-analyzed—queue size K, mixture weights/temperatures, and stability–plasticity controls likely govern convergence and variance; provide ablations, convergence diagnostics, and seed variance.

**Questions:**

1. What is the per-batch adaptation time and memory for HAT-VTR vs. TCR on a single 3090/A10?
2. Does adaptation degrade R@K on clean test data? Any drift detector to gate adaptation?

---

> ### Author Response · Authors · 2025-11-20
>
> > **[W1]** End-to-end cost of the interactive TTA loop (HSM updates + multi-granular adaptation) is not characterized; please report per-round latency, peak memory, and wall-clock on a commodity GPU and CPU, and discuss amortized cost over sequence length.
>
> We thank the reviewer for emphasizing the importance of practical efficiency. In our revised manuscript, we have build a comprehensive analysis in **Section 5.5 and Appendix C.4** to fully characterize the end-to-end cost of the HAT-VTR adpatation.
>
> **1. Per-round Latency and Breakdown (Table 8 & 9):**
> We measured the per-query inference latency on a commodity GPU (NVIDIA RTX 4090). As shown in Table 8, HAT-VTR requires 32.27ms per query, which remains highly competitive with the TCR baseline (26.37ms) and is only marginally slower than standard inference. To isolate the cost of our contributions, Table 9 provides a component-wise breakdown: the Hubness Suppression Memory (HSM) update incurs only 4.2ms (13.0% of total time), while the majority of the cost (65.7%) stems from the backward pass (gradient computation), which is inherent to all test-time adaptation methods.
>
> **2. Peak Memory Usage (Figure 11):**
> Regarding memory constraints, we analyzed the peak GPU memory usage during the online adaptation process. As illustrated in Figure 11(a) in Appendix C.4, the peak memory usage of HAT-VTR (~5665 MB) is only negligibly higher (approx. +17MB) than the TCR baseline. Furthermore, Figure 11(b) demonstrates that our core components are extremely lightweight: the HSM and Reliable Memory consume less than 0.4MB and 0.1MB respectively.
>
> **3. Wall-clock Time (Table 15):**
> We report the total wall-clock time to process the entire MSRVTT-1kA test set in Table 15 (Appendix C.4) across different hardware. On a consumer-grade RTX 4090, HAT-VTR completes the *v2t* task in 37.06s (vs. 36.38s for TCR). We also provide benchmarks on an RTX A6000, where HAT-VTR takes 65.41s (vs. 62.34s for TCR). This confirms that the overhead of our interactive loop is minimal in real-world deployment scenarios.  For the convenience, we list the content of Table 15 below:
>
> |  | RTX4090 |  | RTX A6000 |  |
> | --- | --- | --- | --- | --- |
> |  | v2t | t2v | v2t | t2v |
> | tent | 36.27s | 14.27s | 62.05s | 24.11s |
> | read | 36.73s | 14.85s | 62.72s | 24.51s |
> | sar | 63.09s | 16.54s | 113.27s | 28.61s |
> | eata | 36.58s | 14.58s | 62.6s | 24.44s |
> | tcr | 36.38s | 14.52s | 62.34s | 24.36s |
> | ours | 37.06s | 14.77s | 65.41s | 24.76s |
>
> **4. Amortized Cost over Sequence Length:**
> Regarding the sequence length, we clarify that in the standard Video-Text Retrieval (VTR) setting, input dimensions are generally fixed and do not vary dynamically during inference. The video input is typically standardized to a fixed number of sampled frames (e.g., 12 frames in our implementation), and the text input is bounded by the tokenizer's limit (e.g., 77 tokens for CLIPTokenizer). Consequently, the computational cost is dominated by the visual/text encoders and the gradient updates on the query encoder, rather than variations in sequence length. Therefore, the "amortized cost" is effectively constant per query, as reflected in our latency reports.

---

> ### Author Response · Authors · 2025-11-20
>
> > **[W2]** Comparisons to classical retrieval debiasing / hubness-reduction baselines are missing; adding such baselines would better isolate HSM’s contribution.
>
> Thanks for your insight regarding the isolation of our method's contributions. To address this, we have included **Table 14** in the revised paper, which compares HAT-VTR against representative "Training-based" (NeighborRetr) and "Training-free" (HSM+QBNorm) hubness reduction baselines.
>
> First, we wish to clarify that our primary contribution is identifying that spatio-temporal perturbations significantly amplify the hubness phenomenon in VTR—a specific pathology distinct from intrinsic hubness in clean data. Our proposed Hubness Suppression Memory (HSM) is designed to dynamically address this amplification in the online TTA setting. As shown in Table 14, while classical baselines do alleviate the issue—improving average *v2t* Recall@1 from 18.2% to 23.4% (NeighborRetr) and 24.8% (HSM+QBNorm)—our full HAT-VTR framework with X-Pool achieves a superior 30.4%.
>
> This demonstrates that HAT-VTR achieves the best results by combining test-time adaptation with hubness suppression, significantly outperforming methods that rely on suppression alone. Furthermore, our approach proves highly flexible; when applied to the X-Pool architecture, HAT-VTR achieves state-of-the-art performance, significantly outperforming static baselines and confirming its effectiveness as a plug-and-play solution.
>
> For the convenience, we list the content of Table 14 below:
>
> | v2t | type | gaussian | h264_compression | motion_blur | main_object_occlusion | style_transfer | event_insertion | Avg. |
> | --- | --- | --- | --- | --- | --- | --- | --- | --- |
> | CLIP4Clip_ori | - | 8.7 | 26.2 | 24.6 | 22.5 | 4.8 | 22.2 | 18.2 |
> | NeighborRetr | Training-based | 18.5 | 28.1 | 26.4 | 29.9 | 10.9 | 26.5 | 23.4 |
> | HSM+QBNorm | Training-free | 22.5 | 28.6 | 31.2 | 29.3 | 11.8 | 25.6 | 24.8 |
> | CLIP4Clip+ours | TTA-Optimization | 23.1 | 30.6 | 32.6 | 30.1 | 12.1 | 26.3 | 25.8 |
> | Xpool+ours | TTA-Optimization | 26.2 | 35.6 | 35.3 | 35.5 | 14.4 | 35.2 | 30.4 |
> | t2v | type | ocr | char_delete | synonym_replace | word_insert | formal | active | Avg. |
> | CLIP4Clip_ori | - | 21.5 | 11.2 | 38.9 | 39.0 | 40.9 | 41.8 | 32.2 |
> | NeighborRetr | Training-based | 23.8 | 12.8 | 40.2 | 40.8 | 45.4 | 46.2 | 34.9 |
> | QBNorm | Training-free | 23.6 | 12.1 | 40.4 | 39.8 | 42.7 | 42.8 | 33.6 |
> | CLIP4Clip+ours | TTA-Optimization | 24.5 | 12.8 | 40.7 | 41.6 | 43.6 | 43.7 | 34.5 |
> | Xpool+ours | TTA-Optimization | 26.9 | 14.7 | 44.8 | 43.8 | 48.7 | 48.1 | 37.8 |

---

> ### Author Response · Authors · 2025-11-20
>
> > **[W3]** The effect of online adaptation on in-distribution retrieval is unclear; include a no-shift control row to quantify any degradation and implement a switch to disable TTA when drift is not detected.
>
> Thanks for raising this consideration on in-distribution retrieval. Our primary motivation for Online Test-Time Adaptation (OTTA) is the reality that test distributions in deployment are typically unknown and non-stationary; if the test distribution were guaranteed to match the training data a priori, standard inference would indeed suffice without OTTA.  To verify our method's safety in such mixed settings, we conducted extensive experiments in **Table 17, 18**.
>
> Specifically, Table 18 simulates a realistic scenario where only a subset of queries are corrupted while the remainder are clean in-distribution samples. We observe that HAT-VTR maintains superior performance across all noise ratios. Even in the low-drift setting where only 20% of queries are perturbed, our method achieves the strongest baseline’s 36.6%. Similarly, Table 17 shows that our method consistently outperforms baselines even when perturbation severities fluctuate significantly within a batch. These results demonstrate that HAT-VTR does not catastrophically degrade performance on clean data and effectively handles mixed streams.
>
> Regarding the suggestion to implement an explicit switch to disable adaptation, we view the design of a dedicated hard gate for drift detection as a distinct Out-of-Distribution (OOD) research problem that lies parallel to our current focus on adaptation mechanisms. However, we agree this is a valuable direction for optimizing efficiency in future work.
>
> > **[W4]** Hyperparameter sensitivity is under-analyzed—queue size K, mixture weights/temperatures… and seed variance.
>
> Thank you for highlighting the importance of  stability analysis. In our revised paper, we have significantly expanded our ablation studies to systematically cover hyperparameter sensitivity, convergence diagnostics, and seed variance. To address concerns about parameter tuning, we provide detailed ablations for both our newly introduced parameters (Table A) and standard TTA parameters (Table B). As illustrated in **Figure 5 and Figure 7** in our paper, HAT-VTR maintains stable performance across a wide range of settings on both MSRVTT and ActivityNet datasets.
>
> - **Table A: Novel Hyperparameters introduced by HAT-VTR**
>
> | Hyperparameters | Description | Ablation Location |
> | --- | --- | --- |
> | $K$ | Size of Hubness Suppression Memory (HSM) queue | Fig. 7 (b) |
> | $\alpha, \beta$ | Temperature scaling factors for gallery/query in HSM | Fig. 5 (a), Tab. 12 |
> | $m$ | Balancing weight for hubness suppression (Eq. 6) | Fig. 5 (b) |
> - **Table B: Original TCR / Optimization Hyperparameters**
>
> | Hyperparameters | Description | Ablation Location |
> | --- | --- | --- |
> | $\tau$ | Softmax temperature for contrastive loss (Eq. 3) | Fig. 7 (a) |
> | $t$ | Temperature for uniformity loss (Eq. 7, 8) | Fig. 6 (a) |
> | Batch Size | Batch size for online adaptation | Fig. 6 (b) |
> | Learning Rate | Step size for optimizer | Fig. 6 (c) |
> | Reliable Memory Size | Capacity of the reliable memory bank | Fig. 7 (c) |
>
> Regarding optimization stability, we plotted the loss curves during TTA under various perturbation types (Gaussian, Style Transfer, OCR) in **Figure 8.** The loss values consistently decrease and converge rapidly within the adaptation steps, confirming that our multi-granular objectives lead to stable optimization without divergence. Finally, we evaluated the method's robustness to initialization by running experiments across 5 different random seeds (0, 42, 100, 200, 512). As reported in **Table 11**, the performance variance is minimal. Therefore, we fix the seed to 42 in all other experiments following X-Pool’s implementation. For the convenience, we list the Table 11 below:
>
> |  | MSRVTT |  | ActivityNet |  |
> | --- | --- | --- | --- | --- |
> | seed | v2t | t2v | v2t | t2v |
> | 42 | 25.8 | 34.5 | 21.4 | 32.3 |
> | 0 | 25.8 | 34.6 | 21.2 | 32.5 |
> | 100 | 25.9 | 34.7 | 21.3 | 32.3 |
> | 200 | 25.5 | 34.4 | 21.4 | 32.4 |
> | 512 | 25.6 | 34.4 | 21.4 | 32.3 |

---

> ### Author Response · Authors · 2025-11-20
>
> > **[Q1]** What is the per-batch adaptation time and memory for HAT-VTR vs. TCR on a single 3090/A10?
>
> Please first refer to our detailed efficiency analysis in Response to **W1**. Regarding the specific hardware inquiry, while we do not have immediate access to an RTX 3090 or A10, we conducted benchmarks on the RTX 4090 and RTX A6000, which serve as reliable proxies for high-end consumer and workstation-grade performance.
>
> In terms of latency, **Table 8** shows that on an RTX 4090, the per-query adaptation time for HAT-VTR is 32.27ms, compared to 26.37ms for TCR. This trend holds on the RTX A6000, where the total wall-clock time for the *v2t* task is 65.41s for HAT-VTR versus 62.34s for TCR, confirming that the computational overhead is minimal across different GPU architectures. Regarding memory, **Figure 11(a)** demonstrates that HAT-VTR is extremely efficient, with a peak memory usage of approximately 5665 MB, which is only ~17 MB higher than the TCR baseline.
>
> > **[Q2]** Does adaptation degrade R@K on clean test data? Any drift detector to gate adaptation?
>
> Please first refer to our **Response to W3** for a detailed discussion on this topic. We address the impact on "clean data" in two distinct contexts. First, in **Query-Gallery Shift (QGS)** scenarios—such as Cross-dataset (**Table 4**) or Zero-shot adaptation (**Table 5**)—where the test data is visually "clean" but distributionally distinct from the training source, HAT-VTR does not degrade performance. Instead, it adapts to the domain gap, significantly improving Recall. More results in this situation can be found in **Table 26-31 in Appendix**.
>
> Second, in the scenario where the test stream contains a mix of corrupted and in-distribution (clean) data, our experiments in **Table 18** demonstrate robustness. Even in a low-noise setting where only 20% of queries are corrupted (implying 80% are clean), HAT-VTR achieves a Recall@1 of 37.4%, surpassing the non-adapted CLIP4Clip baseline of 36.7%. Ideally, if the test data is guaranteed to follow the training distribution exactly, direct inference using the original frozen model is sufficient.
>
> Regarding the drift detector, we do not currently employ a hard switch to gate adaptation. While our noise-robust loss implicitly handles uncertainty, designing an explicit OOD detector lies outside the scope of this work, though we acknowledge it as a valuable direction for future research.

---

> ### Author Response · Authors · 2025-11-26
> **Looking forward to your reply**
>
> Dear Reviewer drGP,
>
> Thanks for your valuable feedback and insightful suggestions, which have helped us refine and clarify our work. We have carefully addressed all your raised concerns in our response.
>
> We would greatly appreciate it if you could provide further feedback. Your input is invaluable to ensuring the quality and clarity of our work.
>
> Kind Regards,
>
> Authors of Paper 7458

---

### Official Review · Reviewer_qdkY · 2025-11-01

**Soundness:** 3
**Presentation:** 3
**Contribution:** 3
**Rating:** 6
**Confidence:** 4

**Summary:**

This work studies the domain shift of the text-video retrieval by investigating the spatial-temporal axises. Accordingly, this work proposes a   new benchmark that performs video perturbation from low/mid/high-levels. Also a new method is proposed to enhance the test-time adaptation. Both the benchmark and the method admit the necessity of improving the generalization ability of text-video retrieval methods.

**Strengths:**

* Studying the generalization and robustness of the text-video retrieval is meaningful and urgent. The proposed method introduces a valuable benchmark. I'd like to support this work if the benchmark could be properly released to the academic communities.

* The proposed method studies different levels of the perturbation when devising the dataset, which is inspiring.

* The proposed benchmark and method are well-motivated.

**Weaknesses:**

* There might be some logical issues at presentation. This work starts from the query shift but operates on the video side, which is confusing. Are there any rationales that might be missing to bridge the two?

* It seems that there is no discussion on the query generalization method applied when devising the benchmark.

* It seems that the proposed benchmark is applied on very limited methods and datasets, which might lack generalizability.

* There lacks the discussion of the generaliability for existing foundation models on the proposed perturbations.

**Questions:**

* What might be the efficiency cost of the proposed method, such as the memory usage and the latency cost.

* Is that possible to provide some empirical evidence to show that the improvement on the hubness phenomenon.

---

> ### Author Response · Authors · 2025-11-20
>
> We thank the reviewer for their positive assessment of the importance and motivation of our work, and **we will fully open-source the MLVP benchmark together with all adaptation methods code** upon acceptance.
>
> > **[W1]** There might be some logical issues at presentation. This work starts from the query shift but operates on the video side, which is confusing. Are there any rationales that might be missing to bridge the two?
>
> Our work addresses video–text retrieval in both *v2t* and *t2v* modes, where query shift in practice often appears on the video side: user-uploaded videos are heavily affected by sensor noise, compression, and environmental factors, and we find **models are especially sensitive to such video-side perturbations.** In contrast, prior cross-modal robustness work has largely focused on image–text settings and on image/text perturbations, leaving video-specific degradations underexplored. This gap motivates MLVP as a dedicated multi-level video perturbation benchmark that concretely instantiates query shift in realistic video-side conditions.
>
> We also emphasize that **we do not ignore the text side**. In the *t2v* setting, we additionally evaluate robustness under 15 types of text perturbations, following the established perturbation recipes in MM-Robustness. These text-side experiments are reported in **Table 3** and **Appendix D.3**, and show that our framework consistently improves robustness under both video and text perturbations.
>
> > **[W2]** It seems that there is no discussion on the query generalization method applied when devising the benchmark.
>
> Thanks for your comment, but we think the term “query generalization method” is somewhat ambiguous here: MLVP is designed as a method-agnostic evaluation benchmark, and we do not apply any model-specific query generalization procedure when constructing it. Instead, MLVP realizes query shift through a multi-level video perturbation design, ranging from low-level degradations (e.g., noise, blur, compression) to higher-level temporal and semantic changes, explicitly crafted to mimic diverse real-world conditions that video queries may encounter. Beyond this benchmark construction, we further evaluate more challenging scenarios in our TTA experiments, including Query–Gallery Shift (Tables 4 and 5) and mixed perturbation settings (Appendix D.2), which demonstrate that **our framework remains effective under stronger and more heterogeneous query shifts**.
>
> We are happy to further discuss and clarify this point.

---

> ### Author Response · Authors · 2025-11-20
>
> > **[W3]** It seems that the proposed benchmark is applied on very limited methods and datasets, which might lack generalizability.
>
> Thanks for your comment. From the method side, we deliberately evaluate on two representative base models with different architectures and granularity—CLIP4Clip (coarse-grained) and X-Pool (fine-grained). Each base model is combined with six TTA methods (TENT, READ, SAR, EATA, TCR, and our HAT-VTR), **yielding 12 distinct method–adaptation configurations**. From the dataset side, our original submission already covers four widely used benchmarks (MSR-VTT, ActivityNet, LSMDC, MSVD), spanning both video–sentence and video–paragraph retrieval; this is on par with or stronger than recent VTR works such as CLIP4Clip, X-Pool, CLIP-VIP, NeighborRetr, and BIATVR (see the comparison table below).
>
> | Model | Adopted VTR Dataset | Total Number |
> | --- | --- | --- |
> | CLIP4Clip (Neurocomp 21) [1] | MSRVTT, MSVD, LSMDC, ActivityNet, Didemo | 5 |
> | X-Pool (CVPR22) [2] | MSRVTT, MSVD, LSMDC | 3 |
> | CLIP-VIP (ICLR23) [3] | MSRVTT, Didemo, LSMDC, ActivityNet | 4 |
> | NeighborRetr (CVPR25) [4] | MSRVTT, Didemo, MSVD, ActivityNet | 4 |
> | BIATVR (ICLR25) [5] | MSRVTT, MSVD, LSMDC, VATEX | 4 |
> | Ours (original) | MSRVTT, ActivityNet, LSMDC, MSVD | 4 |
> | Ours (revised) | MSRVTT, ActivityNet, LSMDC, MSVD, Didemo | 5 |
>
> To further strengthen the evidence, we have additionally included experiments on DiDeMo (Tables 24 and 25), where HAT-VTR consistently outperforms all compared TTA methods under both video and text perturbations for both base models. These results suggest that our benchmark and adaptation method generalize well across diverse architectures and datasets.
>
> For the convenience, we list the content of Table 24 and Table 25 below. These results all prove the efficacy of our HAT-VTR method.
>
> | Query Shift | Low-Level |  |  |  |  |  | Mid-Level |  |  | High-Level |  |  |  |
> | --- | --- | --- | --- | --- | --- | --- | --- | --- | --- | --- | --- | --- | --- |
> |  | Gauss. | Impul. | Fog | Snow | Elastic. | H264. | Motion | Defocus | Occlu. | Style | Event | Tempo. | Avg. |
> | CLIP4Clip | 6.08 | 6.67 | 21.61 | 12.15 | 11.16 | 32.87 | 22.21 | 5.88 | 13.65 | 4.88 | 1.69 | 30.18 | 14.09 |
> | Tent | 4.28 | 4.38 | 20.72 | 10.86 | 12.75 | 32.87 | 23.21 | 4.48 | 11.25 | 4.58 | 1.49 | 29.88 | 13.40 |
> | READ | 6.87 | 7.67 | 22.91 | 12.65 | 10.66 | 32.67 | 20.02 | 5.88 | 14.24 | 4.78 | 1.79 | 29.68 | 14.15 |
> | SAR | 4.58 | 4.78 | 22.31 | 11.85 | 12.55 | 33.17 | 23.01 | 5.08 | 13.25 | 5.08 | 1.79 | 30.18 | 13.97 |
> | EATA | 5.68 | 2.29 | 24.90 | 10.66 | 16.93 | 33.57 | 25.30 | 6.67 | 14.94 | 5.98 | 1.59 | 29.98 | 14.87 |
> | TCR | 13.15 | 14.54 | 27.29 | 18.73 | 18.03 | 33.76 | 23.80 | 10.76 | 16.33 | 6.77 | 2.79 | 29.18 | 17.93 |
> | Ours | 20.32 | 19.12 | 35.26 | 25.00 | 25.20 | 37.55 | 29.48 | 16.73 | 22.91 | 8.17 | 5.58 | 31.97 | 23.11 |
> | Xpool | 9.36 | 10.56 | 25.20 | 15.64 | 15.24 | 37.25 | 24.20 | 6.37 | 18.73 | 7.87 | 29.88 | 30.98 | 19.27 |
> | Tent | 8.67 | 9.16 | 23.51 | 15.64 | 16.73 | 36.95 | 25.80 | 5.98 | 15.24 | 7.57 | 30.38 | 30.98 | 18.88 |
> | READ | 10.16 | 10.76 | 27.19 | 15.84 | 14.34 | 36.85 | 22.91 | 6.57 | 20.32 | 7.97 | 29.58 | 31.08 | 19.46 |
> | SAR | 9.86 | 10.16 | 24.50 | 16.14 | 16.73 | 36.95 | 25.50 | 6.08 | 18.23 | 7.97 | 30.38 | 31.08 | 19.47 |
> | EATA | 13.75 | 9.96 | 26.59 | 12.75 | 19.42 | 37.75 | 26.89 | 7.47 | 21.02 | 6.47 | 31.47 | 31.08 | 20.39 |
> | TCR | 16.93 | 14.84 | 32.37 | 23.21 | 20.02 | 38.05 | 27.79 | 10.76 | 23.61 | 7.57 | 30.28 | 30.68 | 23.01 |
> | Ours | 20.32 | 22.21 | 37.75 | 27.69 | 30.78 | 39.74 | 33.17 | 17.53 | 26.39 | 11.55 | 33.96 | 33.37 | 27.87 |
>
> | Query Shift | Character-Level |  |  |  |  | Word-Level |  |  |  |  | Sentence-Level |  |  |  |  |  |
> | --- | --- | --- | --- | --- | --- | --- | --- | --- | --- | --- | --- | --- | --- | --- | --- | --- |
> |  | OCR | CI | CR | CS | CD | SR | WI | WS | WD | IP | Backtrans. | Formal | Casual | Passive | Active | Avg. |
> | CLIP4Clip | 28.39 | 19.72 | 20.82 | 21.12 | 21.51 | 34.56 | 36.35 | 35.26 | 35.76 | 37.75 | 27.79 | 34.46 | 35.76 | 34.46 | 34.46 | 30.54 |
> | Tent | 28.09 | 19.42 | 21.31 | 21.41 | 21.71 | 34.06 | 36.16 | 35.36 | 35.46 | 37.75 | 27.99 | 34.36 | 35.86 | 34.56 | 34.36 | 30.52 |
> | READ | 28.88 | 19.82 | 20.92 | 20.92 | 21.91 | 34.76 | 36.06 | 35.66 | 35.46 | 37.95 | 28.09 | 34.46 | 35.56 | 34.46 | 34.26 | 30.61 |
> | SAR | 28.19 | 19.52 | 21.22 | 21.41 | 21.91 | 34.26 | 36.16 | 35.26 | 35.66 | 37.75 | 27.89 | 34.46 | 35.56 | 34.76 | 34.06 | 30.54 |
> | EATA | 27.09 | 19.72 | 20.72 | 21.81 | 20.82 | 33.47 | 35.16 | 34.46 | 34.26 | 37.65 | 27.29 | 34.26 | 34.36 | 33.96 | 33.86 | 29.93 |
> | TCR | 27.79 | 18.43 | 21.02 | 22.61 | 20.62 | 33.76 | 35.26 | 35.06 | 34.56 | 38.25 | 27.19 | 34.26 | 34.66 | 33.96 | 33.67 | 30.07 |
> | Ours | 28.89 | 21.51 | 21.91 | 23.31 | 21.81 | 36.25 | 38.15 | 36.75 | 37.35 | 38.55 | 30.58 | 36.46 | 37.55 | 34.26 | 35.46 | 31.92 |

---

> ### Author Response · Authors · 2025-11-20
>
> > **[W4]** There lacks the discussion of the generaliability for existing foundation models on the proposed perturbations.
>
> A key goal of HAT-VTR is to serve as a plug-in adaptation framework that can be seamlessly combined with strong existing VTR models, many of which are foundation-model based. Beyond the two base models in the original submission, **we have therefore added experiments on four widely used multimodal foundation models**—CLIP-ViT-B/32, CLIP-ViT-B/16, BLIP-ViT-B/16, and LanguageBind[6] (a universal multimodal retriever)—to further demonstrate the robustness and general applicability of our approach under the proposed MLVP perturbations.
>
> As shown in the new Table 16 (or Table below), these models still suffer from substantial degradation under MLVP perturbations, indicating that large-scale pre-training alone does not guarantee robustness. When equipped with HAT-VTR, all four models obtain strong and consistent improvements across all perturbation types, with *Avg* scores increasing, and outperform all TTA baselines (TENT, READ, SAR, EATA, TCR). This demonstrates that **our method is model-agnostic and scales effectively to current foundation models.**
>
> | Query Shift | Low-Level |  |  |  |  |  | Mid-Level |  |  | High-Level |  |  |  |
> | --- | --- | --- | --- | --- | --- | --- | --- | --- | --- | --- | --- | --- | --- |
> |  | Gauss. | Impul. | Fog | Snow | Elastic. | H264. | Motion | Defocus | Occlu. | Style | Event | Tempo. | Avg. |
> | CLIP-ViT-B/32 | 7.1 | 4.9 | 16.1 | 10.6 | 7.9 | 14.2 | 11.8 | 3.6 | 8.7 | 4.3 | 16.7 | 22.8 | 10.7 |
> | Tent | 5.5 | 4.3 | 15.3 | 9.3 | 7.4 | 14.4 | 14.5 | 3.6 | 8.5 | 3.2 | 17.4 | 22.8 | 10.5 |
> | READ | 7.4 | 5.0 | 17.0 | 11.1 | 7.8 | 14.7 | 9.9 | 3.4 | 8.7 | 4.4 | 15.5 | 22.1 | 10.6 |
> | SAR | 6.4 | 4.7 | 16.0 | 9.9 | 8.3 | 14.6 | 14.2 | 3.9 | 8.8 | 3.6 | 17.4 | 22.9 | 10.9 |
> | EATA | 4.1 | 2.6 | 17.1 | 3.2 | 1.0 | 15.1 | 15.6 | 2.7 | 10.1 | 1.8 | 18.5 | 23.1 | 9.6 |
> | TCR | 12.0 | 9.3 | 21.8 | 13.3 | 14.2 | 17.1 | 17.5 | 6.6 | 12.5 | 6.3 | 18.7 | 23.8 | 14.4 |
> | Ours | 18.4 | 15.9 | 29.7 | 22.7 | 24.2 | 25.0 | 26.1 | 13.1 | 22.6 | 7.6 | 23.4 | 27.4 | 21.3 |
> | CLIP-ViT-B/16 | 5.5 | 6.8 | 19.9 | 15.4 | 3.7 | 15.6 | 14.0 | 4.0 | 12.5 | 3.2 | 17.8 | 25.8 | 12.0 |
> | Tent | 3.1 | 4.9 | 19.9 | 15.3 | 2.9 | 14.3 | 15.2 | 3.2 | 11.4 | 2.7 | 18.1 | 26.0 | 11.4 |
> | READ | 6.2 | 7.3 | 19.8 | 15.7 | 4.2 | 15.5 | 12.9 | 4.0 | 12.2 | 4.0 | 15.9 | 24.6 | 11.9 |
> | SAR | 4.7 | 6.2 | 20.2 | 15.6 | 3.3 | 14.5 | 15.1 | 3.6 | 12.4 | 2.8 | 18.0 | 25.8 | 11.9 |
> | EATA | 4.9 | 3.7 | 21.1 | 15.5 | 1.0 | 15.7 | 16.7 | 1.5 | 13.6 | 1.1 | 18.3 | 26.7 | 11.7 |
> | TCR | 11.6 | 11.0 | 26.5 | 19.0 | 12.5 | 18.1 | 19.0 | 8.1 | 14.2 | 5.0 | 17.3 | 24.8 | 15.6 |
> | Ours | 17.4 | 18.0 | 32.6 | 26.1 | 19.3 | 26.6 | 26.4 | 15.4 | 26.4 | 9.4 | 25.6 | 30.6 | 22.8 |
> | BLIP-ViT-B/16 | 6.7 | 7.4 | 20.6 | 16.3 | 4.5 | 17.2 | 15.1 | 4.7 | 12.9 | 4.5 | 18.4 | 26.4 | 12.9 |
> | Tent | 3.6 | 5.2 | 20.3 | 16.2 | 3.7 | 15.6 | 16.1 | 4.1 | 12.8 | 4.1 | 18.8 | 26.4 | 12.2 |
> | READ | 6.9 | 7.8 | 20.1 | 16.6 | 5.0 | 17.0 | 15.1 | 4.8 | 12.8 | 4.6 | 17.9 | 25.8 | 12.9 |
> | SAR | 5.3 | 7.0 | 21.1 | 16.6 | 4.1 | 16.6 | 16.3 | 4.5 | 13.1 | 3.9 | 19.2 | 26.8 | 12.9 |
> | EATA | 5.1 | 3.9 | 23.2 | 16.2 | 1.5 | 16.2 | 17.6 | 2.2 | 14.3 | 1.4 | 18.4 | 26.9 | 12.2 |
> | TCR | 11.9 | 12.5 | 28.1 | 19.7 | 13.0 | 18.8 | 20.2 | 8.5 | 15.8 | 5.3 | 18.8 | 25.2 | 16.5 |
> | Ours | 17.9 | 19.2 | 34.0 | 26.5 | 20.2 | 27.6 | 26.7 | 15.3 | 28.2 | 10.0 | 26.1 | 31.5 | 23.6 |
> | LanguageBind | 14.4 | 15.0 | 32.9 | 23.1 | 11.2 | 27.6 | 27.3 | 5.5 | 27.5 | 7.1 | 23.2 | 32.6 | 20.6 |
> | Tent | 12.2 | 11.1 | 32.0 | 22.3 | 8.5 | 27.5 | 27.6 | 3.3 | 28.0 | 3.7 | 24.2 | 32.6 | 19.4 |
> | READ | 15.3 | 15.8 | 33.4 | 23.0 | 11.8 | 27.8 | 27.4 | 6.5 | 26.8 | 7.7 | 23.2 | 33.0 | 21.0 |
> | SAR | 14.1 | 14.1 | 33.1 | 23.7 | 9.7 | 27.7 | 27.4 | 4.2 | 28.0 | 4.9 | 24.1 | 32.9 | 20.3 |
> | EATA | 16.5 | 7.1 | 35.2 | 16.5 | 4.5 | 29.4 | 27.6 | 0.4 | 30.8 | 4.3 | 22.5 | 32.9 | 19.0 |
> | TCR | 18.4 | 19.5 | 32.3 | 27.1 | 15.5 | 29.1 | 28.5 | 8.0 | 28.0 | 9.9 | 22.5 | 31.8 | 22.6 |
> | Ours | 24.7 | 23.9 | 39.3 | 31.9 | 24.8 | 32.5 | 32.3 | 15.1 | 33.0 | 12.5 | 28.9 | 33.3 | 27.7 |

---

> ### Author Response · Authors · 2025-11-20
>
> > **[Q1]** What might be the efficiency cost of the proposed method, such as the memory usage and the latency cost.
>
> The main overhead of HAT-VTR comes from the standard backward pass used by all TTA methods, while our hubness-specific components are lightweight. On an RTX 4090 with batch size 16 (**Table 8**), the per-query runtime of HAT-VTR is 32.27 ms, comparable to strong TTA baselines such as TCR (26.37 ms) and far lower than SAR (53.41 ms). The component-wise breakdown in **Table 9** shows that the backward pass accounts for 65.7% of the total runtime, whereas the HSM contributes only 4.2 ms (13.0%) and the Reliable Memory module 2.45 ms (7.6%). At the end-to-end level, **Table 15** reports the wall-clock time on the full MSRVTT-1kA test set: on an RTX 4090, our method takes 37.06 s vs. 36.38 s for TCR in the v2t setting (14.77 s vs. 14.52 s for t2v), and shows similarly small gaps on an RTX A6000, indicating only a minor increase in total latency.
>
> For memory, **Appendix C.4** provides a detailed analysis. As shown in Fig. 11(a), the peak GPU memory usage of HAT-VTR is only about 17 MB higher than TCR. **Fig. 11(b)** further shows that HSM and Reliable Memory together consume less than 0.5 MB (≈0.4 MB and 0.1 MB respectively), and their usage scales linearly with the queue size rather than the gallery size, making the overhead negligible in large-scale retrieval. Overall, HAT-VTR achieves substantial robustness gains with only a small increase in runtime and a negligible memory cost, which makes it practical for real-world video–text retrieval deployments.
>
> > **[Q2]** Is that possible to provide some empirical evidence to show that the improvement on the hubness phenomenon.
>
> Our paper provides both qualitative and quantitative evidence that HAT-VTR explicitly alleviates hubness. First, **Fig. 2** analyzes the *k-occurrence* distribution: under Gaussian perturbations, the distribution for CLIP4Clip becomes highly heavy-tailed, indicating severe hubness, while HAT-VTR restores a much more balanced k-occurrence histogram than both the corrupted model and the TCR baseline. Second, **Fig. 9** visualizes similarity matrices for 250 v2t samples: compared with the original model and “TCR + Gaussian”, our method yields a much cleaner diagonal and strongly suppressed off-diagonal activations, showing that hubs are no longer dominating the similarity space.
>
> Finally, following NeighborRetr[7], we additionally include in the revised paper several hubness-related metrics (skewness, truncated skewness, etc.) in **Table 13** of our paper. Across different perturbations (e.g., Gaussian, motion blur, temporal scrambling, OCR, back-translation), HAT-VTR consistently achieves the lowest hubness scores (e.g., skew: 9.09→5.07→0.97 for CLIP4Clip/TCR/Ours under Gaussian), quantitatively confirming that our approach substantially mitigates the hubness phenomenon.
>
> For the convenience, we list the Table 13 of our paper below
>
> |  | gaussian |  |  | motion_blur |  |  | temporal_scrambling |  |  | ocr |  |  | backtrans |  |  |
> | --- | --- | --- | --- | --- | --- | --- | --- | --- | --- | --- | --- | --- | --- | --- | --- |
> | Metrics | CLIP4Clip4 | TCR | Ours | CLIP4Clip4 | TCR | Ours | CLIP4Clip4 | TCR | Ours | CLIP4Clip4 | TCR | Ours | CLIP4Clip4 | TCR | Ours |
> | skew | 9.09 | 5.07 | 0.97 | 16.27 | 5.25 | 0.86 | 2.19 | 1.8 | 1.15 | 1.95 | 1.84 | 0.39 | 1.36 | 1.19 | 1.12 |
> | trunc | 5.05 | 2.12 | 0.63 | 2.22 | 1.55 | 0.6 | 1.11 | 1.18 | 0.53 | 1.19 | 1.04 | 0.34 | 0.72 | 0.8 | 0.5 |
> | atkinson | 0.6 | 0.27 | 0.05 | 0.23 | 0.15 | 0.05 | 0.08 | 0.1 | 0.06 | 0.12 | 0.11 | 0.05 | 0.08 | 0.07 | 0.06 |
> | robin | 0.64 | 0.41 | 0.18 | 0.38 | 0.31 | 0.18 | 0.38 | 0.31 | 0.18 | 0.27 | 0.26 | 0.18 | 0.22 | 0.21 | 0.19 |
> | anti | 0.19 | 0.04 | 0 | 0.02 | 0 | 0 | 0.02 | 0.004 | 0 | 0 | 0 | 0 | 0 | 0 | 0 |
> | hub | 10.25 | 6.58 | 0.94 | 6.11 | 4.54 | 1.07 | 2.78 | 3.32 | 1.24 | 3.74 | 3.16 | 0.59 | 2.63 | 1.88 | 1.15 |

---

> ### Author Response · Authors · 2025-11-20
>
> ### Reference
>
> [1] Luo, Huaishao, et al. "Clip4clip: An empirical study of clip for end to end video clip retrieval." arXiv preprint arXiv:2104.08860 (2021).
>
> [2] Gorti, Satya Krishna, et al. "X-pool: Cross-modal language-video attention for text-video retrieval." *Proceedings of the IEEE/CVF conference on computer vision and pattern recognition*. 2022.
>
> [3] Xue, Hongwei, et al. "Clip-vip: Adapting pre-trained image-text model to video-language representation alignment." arXiv preprint arXiv:2209.06430 (2022).
>
> [4] Lin, Zengrong, et al. "NeighborRetr: Balancing Hub Centrality in Cross-Modal Retrieval." Proceedings of the Computer Vision and Pattern Recognition Conference. 2025.
>
> [5] Bai, Zechen, et al. "Bridging information asymmetry in text-video retrieval: A data-centric approach." arXiv preprint arXiv:2408.07249 (2024).
>
> [6] Zhu, Bin, et al. "Languagebind: Extending video-language pretraining to n-modality by language-based semantic alignment." arXiv preprint arXiv:2310.01852 (2023).
>
> [7] Lin, Zengrong, et al. "NeighborRetr: Balancing Hub Centrality in Cross-Modal Retrieval." *Proceedings of the Computer Vision and Pattern Recognition Conference*. 2025.

---

> ### Author Response · Authors · 2025-11-26
> **Looking forward to your reply**
>
> Dear Reviewer qdkY,
>
> We appreciate the time you took to review our work. Following your constructive comments, we have posted a detailed response and uploaded a revised version of our paper to clarify the points you mentioned.
>
> As the discussion period progresses, we would be very grateful to hear your thoughts on our revisions. We hope that our updates have successfully addressed your concerns.
>
> Kind Regards,
>
> Authors of Paper 7458

---

### Official Review · Reviewer_2w2B · 2025-11-01

**Soundness:** 3
**Presentation:** 4
**Contribution:** 3
**Rating:** 6
**Confidence:** 5

**Summary:**

Modern Video-Text Retrieval (VTR) models demonstrate strong performance on standard benchmarks but are highly vulnerable to real-world distribution shifts in query data, which cause a significant performance drop and amplify the "hubness" phenomenon where a few gallery items become dominant hubs for an disproportionate number of queries. To systematically evaluate this robustness issue, this paper first introduces a comprehensive benchmark with 12 types of video perturbations across five severity levels. In response, the authors propose HAT-VTR, a test-time adaptation framework designed to directly mitigate hubness. This framework leverages a Hubness Suppression Memory to refine similarity scores and a Multi-Granularity Loss to enhance temporal feature consistency. Extensive experiments show that HAT-VTR significantly improves robustness, consistently outperforming prior methods across diverse query shift scenarios and thereby increasing model reliability for practical applications. Overall, the paper presents an interesting and valuable contribution, and the reviewer expresses a willingness to raise their score pending satisfactory responses to their questions and concerns

**Strengths:**

1.   The authors have clearly identified and articulated a critical issue in video-text retrieval: the problem of overcoming the query gap between training data and real-world application scenarios. They provide a fairly detailed introduction to related work and effectively contrast their approach with TCR techniques from image-text retrieval, highlighting the distinctions of applying such technology in the video domain.

 2.    The authors' idea of designing a Hubness Suppression Memory module to identify and mitigate the influence of hub points in the embedding space is both intuitive and sound.

3.    The authors present their work through a well-structured and clearly organized writing flow, which enhances the readability and coherence of the paper.

**Weaknesses:**

1.     While addressing the query gap between training data and real-world scenarios is indeed a critical issue in video-text retrieval, the authors have overlooked relevant works in their related work section. For instance, approaches like FreestyleRet, which attempt to construct multi-style queries in a data-driven manner to address this task, represent an alternative direction beyond Test-Time Adaptation and should be discussed for a more comprehensive literature review.

 2.   Although the authors have constructed a robustness benchmark with three levels of perturbations for the video modality, it would be interesting to explore whether similar robustness constructions can be applied from the perspective of the text modality. Specifically, I am curious if the authors have considered or could incorporate various types of noise and perturbations into the textual queries to further enhance model resilience.

3.    While the authors clearly explain the weak transferability of TCR (Test-Time Contamination Remediation) techniques to video domains, it is notable that their triplet-based loss functions are inherited directly from the TCR framework. A more in-depth analysis would be beneficial to clarify which specific improvements or modifications in the TCR components contribute most significantly to the performance gains observed in the video-text retrieval task.

**Questions:**

None

---

> ### Author Response · Authors · 2025-11-20
>
> > **[W1]** While addressing the query … the authors have overlooked relevant works in their related work section. For instance, approaches like FreestyleRet … discussed for a more comprehensive literature review.
>
>
> We thank the reviewer for pointing out this relevant line of work[1][2][3]. FreestyleRet introduces a style-diversified query-based image retrieval setting and a dataset that contains multiple query styles (text, sketch, art, low-resolution). It learns to handle these known styles via style-space construction and style-init prompt tuning on top of frozen visual encoders.  This family of methods explicitly trains on curated multi-style datasets to improve robustness to predefined style variations.
>
> Our work, in contrast, focuses on **online Test-Time Adaptation for video–text retrieval under unforeseen query shifts**. We do not assume access to a style-annotated multi-style dataset, nor do we re-train the model on such data. Instead, our method is trained only on clean data and adapts on-the-fly using the unlabeled test stream, targeting complex spatio-temporal perturbations in videos (and various textual corruptions) that are not enumerated as a fixed set of styles. Conceptually, data-driven style robustness (e.g., FreestyleRet) expands the *training distribution* with manually curated styles, while our HAT-VTR framework operates in a stricter setting where only test-time adaptation to unseen query shifts is allowed.
>
> We agree that these directions are complementary and have clarified this in the revised manuscript. Specifically, in the Related Work section we added a new paragraph “**Data-driven Style Robustness**” , where we discuss FreestyleRet and related approaches, emphasize the difference between style-diversified training vs. online Test-TTA.
>
> > **[W2]** Although the authors have constructed … for the video modality, it would be interesting to explore the text modality …into the textual queries to further enhance model resilience.
>
> Thank you for your suggestion to consider robustness from the text side. In fact, **our experiments already adopt a comprehensive text perturbation suite.** Specifically, we use the text perturbation set introduced in MM_Robustness[4], which covers character-, word-, and sentence-level corruptions (e.g., OCR-like character noise, word insert or delete at the word level, and back translation at the sentence level). This benchmark is widely used and provides a relatively complete coverage of realistic textual noise.
>
> Thus, our method is evaluated on **both video-side and text-side robustness**: we intentionally adopt a standard, widely used perturbation protocol for text, while contributing a new three-level perturbation design on the video side, where such a systematic robustness benchmark was previously missing. This combination allows us to more thoroughly validate our conclusions about test-time robustness and hubness across modalities.

---

> ### Author Response · Authors · 2025-11-20
>
> > **[W3]** It is notable that their triplet-based loss functions are inherited … A more in-depth analysis would be beneficial to clarify which specific improvements or modifications in the TCR components…
>
>
> Thanks for you question. To clarify, TCR itself does not use a triplet loss; instead, it employs distribution-level regularizers on global image–text embeddings. We believe the concern is mainly about our **cross-modal loss**. Indeed, Eq. 8 follows the global cross-modal regularizer introduced in TCR and is used in our framework as the global-level constraint. On top of this, Eq. 9 introduces a new frame-level term that aligns video–text covariance and explicitly models temporal structure, which is absent in TCR.
>
> Regarding which components contribute most, **Table 7** shows that adding the global cross-modal term on top of the uniformity losses improves Avg. R@1 from 29.0 to 29.8, and the full model with both global and frame-level terms plus the noise-aware loss further reaches 30.1. **Figure 5(c)** visually confirms that our multi-granular learning moves the query distribution closer to the gallery distribution. We additionally quantify this by measuring the modality gap (distance between the centers of query and gallery embeddings) on MSRVTT:
>
> | Modality gap | v2t | t2v |
> | --- | --- | --- |
> | w/o MGCM | 7.96 | 1.58 |
> | HAT-VTR (full) | **6.13** | **1.07** |
>
> Thus, the proposed multi-granular cross-modal loss is a key factor behind the observed gains, by effectively reducing the modality gap while avoiding collapse. In fact, TCR is not the first to discuss the notion of a modality gap; **this concept has been explicitly analyzed and exploited in other cross-modal works**, such as Mind the Gap [5] and MAPLE [6] .
>
> Reference
>
> [1] Li, Hao, et al. "Freestyleret: retrieving images from style-diversified queries." *European Conference on Computer Vision*. Cham: Springer Nature Switzerland, 2024.
>
> [2]Jia, Yanhao, et al. "Uni-Retrieval: A Multi-Style Retrieval Framework for STEM's Education." *arXiv preprint arXiv:2502.05863* (2025).
>
> [3] Wu, Xinyi, et al. "From query to explanation: Uni-rag for multi-modal retrieval-augmented learning in stem." *arXiv preprint arXiv:2507.03868* (2025).
>
> [4] Qiu, Jielin, et al. "Benchmarking robustness of multimodal image-text models under distribution shift." *arXiv preprint arXiv:2212.08044* (2022).
>
> [5] Liang, Victor Weixin, et al. *Mind the gap: Understanding the modality gap in multi-modal contrastive representation learning.* NeurIPS 35, 2022.
>
> [6] Zhao, Pengfei, et al. *Guiding Cross-Modal Representations with MLLM Priors via Preference Alignment.* arXiv:2506.06970, 2025.

---

> ### Author Response · Authors · 2025-11-26
> **Looking forward to your reply**
>
> Dear Reviewer 2w2B,
>
> Thank you for your review and the positive suggestions. We have submitted our response and updated the paper to reflect your feedback.
>
> We would appreciate it if you could take a moment to review our changes. We look forward to your feedback and hope that our response satisfactorily resolves your concerns.
>
> Kind Regards,
>
> Authors of Paper 7458

---

### Author Response · Authors · 2025-11-20
**Change Logs of Our Paper**

We thank all reviewers for their time, careful reading, and constructive feedback on our submission. We are encouraged that the reviewers consistently found **the problem formulation and motivation clear**, **recognized the value** of the proposed MLVP benchmark and the HAT-VTR test-time adaptation framework, including the Hubness Suppression Memory and multi-granular losses, and **appreciated the strong empirical validation** and overall soundness of the work.

Below we summarize the main revisions made to address the reviewers’ concerns and further improve the paper.

---

**1. Related Work (in response to Reviewer 2w2B - W1)**

- **[Add]** We add a discussion on data-driven style robustness methods (e.g., FreestyleRet) in Section 2 (Related Work) to provide a more comprehensive literature review and better situate our TTA-based approach within the broader robustness literature.

---

**2. Method Clarification (in response to Reviewer tpJo - W1, Q1, Q2)**

- **[Modify]** We expand Section 4.2 to clarify the motivation and mechanisms of the Hubness Suppression Memory (HSM). We explicitly detail its two complementary roles: (1) ensuring stable TTA learning (*Hubness-aware Target Selection*) and (2) improving final retrieval accuracy (*Posterior Similarity Reranking*).
- **[Modify]** For improved clarity, we rename “Hubness-aware Candidate Selection” to “Hubness-aware Target Selection” throughout the paper.

---

**3. Expanded Experimental Results (in response to Reviewer qdkY)**

- **[Add]** To further extend the variety of datasets in our evaluation, we add a new dataset, **DiDeMo**. The new **Table 24** and **Table 25** report comprehensive v2t and t2v results for all compared TTA methods on this dataset.
- **[Add]** To further demonstrate the generalizability of our method (Reviewer qdkY - W4), we add **Table 16**. This new table compares all TTA methods across four different foundation models: CLIP-B/32, CLIP-B/16, BLIP-B/16, and LanguageBind.

---

**4. Hubness Analysis & Baselines (in response to Reviewer qdkY - Q2 & drGP - W2)**

- **[Add]** We now provide empirical evidence for hubness reduction in the new **Table 13**. This table reports results across six standard hubness metrics (e.g., skewness, hub occurrence), quantitatively showing that HAT-VTR effectively alleviates the hubness phenomenon.
- **[Add]** To more directly connect our diagnosis of hubness to the classical hubness-suppression literature and to better validate MLVP as a benchmark for studying hubness in VTR, we add **Table 14**, where we newly adapt representative hubness-suppression methods (NeighborRetr, QBNorm), which are not standard VTR baselines, to our setting and compare them with HAT-VTR on MLVP.

---

**5. Efficiency Analysis (in response to Reviewer drGP - W1 & tpJo - W3)**

- **[Modify]** We have moved the primary efficiency analysis (per-query latency and component breakdown) from the appendix to the main paper (**Table 8** and **Table 9**) for better visibility.
- **[Add]** We add a more detailed end-to-end cost analysis. **Figure 11** now reports the peak GPU memory usage of TTA models, and **Table 15** reports the total wall-clock time to process the entire MSRVTT test set on two different GPUs.

---

**6. Hyperparameter & Stability Analysis (in response to Reviewer drGP - W4 & tpJo - W2)**

- **[Modify]** We expand our hyperparameter sensitivity analysis. **Figure 5(b)** and **Figure 7** now show the impact of key hyperparameters on both MSRVTT and ActivityNet datasets, demonstrating stability across benchmarks.
- **[Add]** We add **Figure 8** to show the loss convergence curves during adaptation.
- **[Add]** We add **Table 11** to analyze the model’s stability across different random seeds and **Table 12** to provide detailed sensitivity data for hyperparameters $\alpha$ and $\beta$ on two datasets.

---

### Meta-Review · Area_Chair_VaQT · 2026-01-07

**Summary:**

The reviewers consistently recognized the importance of studying Query Shift in Video-Text Retrieval (VTR), an area often neglected in favor of standard in-distribution benchmarks. The paper was praised for its "well-motivated" approach and the introduction of the MLVP benchmark, which provides a comprehensive suite of 12 video perturbations. The core of the discussion centered on: (i) Technical Soundness: Understanding how the Hubness Suppression Memory (HSM) and multi-granular losses actually mitigate the "hubness" phenomenon. (ii) Generalizability: Whether the method holds up across diverse foundation models and additional datasets like DiDeMo. (iii) Efficiency: The practical latency and memory overhead of performing test-time adaptation (TTA) in a retrieval pipeline.

**Reviewer Concerns:**

Addressed by Rebuttal:

- Hubness Quantification (qdkY, drGP): The authors provided new empirical evidence using skewness and k-occurrence metrics, proving that HAT-VTR effectively "flattens" the embedding space compared to the TCR baseline.

- Generalizability & Foundation Models (qdkY): The authors added experiments using DiDeMo and four major foundation models (CLIP B/32, B/16, BLIP, LanguageBind). HAT-VTR showed consistent improvements, confirming it is model-agnostic.

- Component Contribution/Mechanics (tpJo): Reviewer tpJo’s concern regarding the "Candidate Selection" module was resolved by clarifying its role in training stability (Target Selection) versus the Final Reranking step. The nomenclature change and Figure 10 (RM accuracy) were decisive.

- Efficiency & Latency (tpJo, drGP): The authors moved detailed efficiency tables to the main text, showing a negligible latency increase (~32ms per query) and minimal memory usage (+17MB), making the TTA loop practical for real-world use.

- Related Work (2w2B): The distinction between "Data-driven Style Robustness" (training-time) and "Online TTA" (test-time) was clearly articulated and integrated into the manuscript.

Outstanding Part:

- Explicit OOD Gating (drGP, tpJo): While the reviewers suggested a "switch" to disable TTA on clean data, the authors argued that OOD detection is a separate research problem. Given that HAT-VTR shows no degradation on clean data (Table 18), the AC agrees that a hard gate is not a requirement for this specific contribution.

**Reviewer Scores:**

Reviewer 2w2B (Initial 6): The reviewer was already leaning positive and explicitly stated they would raise their score if responses were satisfactory. The thorough clarification on style-based retrieval work should secure this.

Reviewer qdkY (Initial 6): This reviewer requested "empirical evidence" for hubness and "discussion on foundation models." The authors provided a massive amount of new data addressing these exact points, likely moving this into a strong accept.

Reviewer tpJo (Initial 4): Most of this reviewer's concerns were based on a misunderstanding of the "Candidate Selection" module and hyperparameter sensitivity. The authors’ detailed FIFO queue explanation and stability charts (Fig 5, 7, 8) effectively nullify the initial reasons for the 4.

Reviewer drGP (Initial 4): The author think it's an AI-generated review due to the hallucinations (irrelevant metrics). After going through the review and rebuttal, the AC think there is insufficient evidence to support this allegation. And the author has addressed the major issue in the rebuttal.

---

### Decision · Program_Chairs · 2026-01-26

Accept (Poster)